# Preserved central model for faster bidirectional compression in distributed settings

**Constantin Philippenko**     **Aymeric Dieuleveut**
CMAP, École Polytechnique, Institut Polytechnique de Paris
`[fistname].[lastname]@polytechnique.edu`

## Abstract

We develop a new approach to tackle communication constraints in a distributed learning problem with a central server. We propose and analyze a new algorithm that performs bidirectional compression and achieves the same convergence rate as algorithms using only uplink (from the local workers to the central server) compression. To obtain this improvement, we design MCM, an algorithm such that the downlink compression *only impacts local models*, while the global model is preserved. As a result, and contrary to previous works, the gradients on local servers are computed on *perturbed models*. Consequently, convergence proofs are more challenging and require a precise control of this perturbation. To ensure it, MCM additionally combines model compression with a memory mechanism. This analysis opens new doors, e.g. incorporating worker dependent randomized-models and partial participation.

## 1 Introduction

Large scale distributed machine learning is widely used in many modern applications [1, 6, 28]. The training is distributed over a potentially large number $N$ of workers that communicate either with a central server [see 17, 22, on federated learning], or using peer-to-peer communication [9, 34, 32].

In this work, we consider a setting using a central server that aggregates updates from remote nodes. Formally, we have a number of features $d \in \mathbb{N}^*$, and a convex cost function $F : \mathbb{R}^d \to \mathbb{R}$. We want to solve the following distributed convex optimization problem using stochastic gradient algorithms [25, 5]: $\min_{w \in \mathbb{R}^d} F(w)$ with $F(w) = \frac{1}{N} \sum_{i=1}^N F_i(w)$, where $(F_i)_{i=1}^N$ is a *local* risk function (empirical risk or expected risk in a streaming framework). This applies to both instances of *distributed* and *federated* learning.

An important issue of those frameworks is the high communication cost between the workers and the central server [16, Sec. 3.5]. This cost is a concern from several points of view. First, exchanging information can be the bottleneck in terms of speed. Second, the data consumption and the bandwidth usage of training large distributed models can be problematic; and furthermore, the energetic and environmental impact of those exchanges is a growing concern. Over the last few years, new algorithms were introduced, compressing messages in the *upload communications* (i.e., from remote devices to the central server) in order to reduce the size of those exchanges [29, 3, 36, 2, 35, 31, 30, 23, 18]. More recently, a new trend has emerged to also compress the *downlink communication*: this is *bidirectional compression*.

The necessity for bidirectional compression can depend on the situation. For example, a single uplink compression could be sufficient in *asymmetric* regimes in which broadcasting a message to $N$ workers ("one to $N$") is faster than aggregating the information coming from each node ("$N$ to one"). However, in other regimes, e.g. with few machines, where the bottleneck is the transfer time of a heavy model (up to several GB in modern Deep Learning architectures) the downlink communication cannot be disregarded, as the upload and download speed are of the same order [24].

35th Conference on Neural Information Processing Systems (NeurIPS 2021).

Furthermore, in a situation in which participants have to systematically download an update (e.g., on their smartphones) to participate in the training, participants would prefer to receive a small size update (compressed) rather than a heavier one. To encompass all situations, we consider algorithms for which the information exchanged is compressed in both directions.

To perform downlink communication, existing bidirectional algorithms [33, 38, 26, 19, 24, 14, 37, 11] first aggregate all the information they have received, compress them and then carry out the broadcast. Both the main "global" model and the "local" ones perform the *same* update with this compressed information. Consequently, the model hold on the central server and the one used on the local workers (to query the gradient oracle) are identical. However, this means that the model on the central server has been artificially *degraded*: instead of using all the information it has received, it is updated with the compressed information.

Here, we focus on *preserving* (instead of *degrading*) the central model: the update made on its side does not depend on the downlink compression. This implies that the local models are *different* from the central model. The local gradients are thus measured on a *"perturbed model"* (or *"perturbed iterate"*): such an approach requires a more involved analysis and the algorithm must be carefully designed to control the deviation between the local and global models [21]. For example, algorithms directly compressing the model or the update would simply not converge.

We propose MCM - *Model Compression with Memory* - a new algorithm that 1) preserves the central model, and 2) uses a memory scheme to reduce the variance of the local model. We prove that the convergence of this method is similar to the one of algorithms using only unidirectional compression.

**Potential Impact.** Proposing an analysis that handles perturbed iterates is the key to unlock three major challenges of distributed learning run with bidirectionally compressed gradients. First, we show that it is possible to improve the convergence rate by sending *different **randomized** models* to the different workers, this is Rand-MCM. Secondly, this analysis also paves the way to deal with partially participating machines: the adaptation of Rand-MCM to this framework is straightforward; while adapting existing algorithms [26] to partial participation is not practical. Thirdly, this framework is also promising in terms of business applications, e.g., in the situation of learning with privacy guarantees and *with a trusted central server*. We detail those three possible extensions in Subsection 4.1.

**Broader impact.** This work is aligned with a global effort to make the usage of large scale Federated Learning sustainable by minimizing its environmental impact. Though the impact of such algorithms is expected to be positive, at least on environmental concerns, cautiousness is still required, as a rebound effect may be observed [12]: having energetically cheaper and faster algorithms may result in an increase of such applications, annihilating the gain made by algorithmic progress.

**Contributions.** We make the following contributions:
1. We propose a new algorithm MCM, combining a memory process to the "preserved" update. To convey the key steps of the proof, we also introduce an auxiliary hypothetical algorithm, Ghost.
2. For those algorithms, we carefully control the variance of the local models w.r.t. the global one. We provide a *contraction equation* involving the control on the local model's variance and show that MCM achieves the same rate of convergence as single compression in strongly-convex, convex and non-convex regimes. We give a comparisons of MCM's rates with existing algorithms in Table 2.
3. We propose a variant, Rand-MCM incorporating diversity into models shared with the local workers and show that it improves convergence for quadratic functions.

This is the first algorithm for double compression to focus on a **preserved central model**. We underline, both theoretically and in practice, that we get the same asymptotic convergence rate for simple and double compression - which is a major improvement. Our approach is one of the first to allow for worker dependent model, and to naturally adapt to worker dependent compression levels.

The rest of the paper is organized as follows: in Section 2 we present the problem statement and introduce MCM and Rand-MCM. Theoretical results on these algorithms are successively presented in Sections 3 and 4. Finally, we present experiments supporting the theory in Section 5.

## 2   Problem statement

We consider the minimization problem described in Section 1. In the convex case, we assume there exists an optimal parameter $w_*$, and denote $F_* = F(w_*)$. We use $\|\cdot\|$ to denote the Euclidean norm. To solve this problem, we rely on a stochastic gradient descent (SGD) algorithm. A stochastic gradient $g^i_{k+1}$ is provided at iteration $k$ in $\mathbb{N}$ to the device $i$ in $[\![1, N]\!]$. This gradient oracle can be

Table 1: Features of the main existing algorithms performing compression. $e_k^i$ (resp. $E_k$) denotes the use of error-feedback at uplink (resp. downlink). $h_k^i$ (resp. $H_k$) denotes the use of a memory at uplink (resp. downlink). Note that `Dist-EF-SGD` is identical to `Double-Squeeze` but has been developed simultaneously and independently.

| | Compr. | $e_k^i$ | $h_k^i$ | $E_k$ | $H_k$ | Rand. | update point |
|---|---|---|---|---|---|---|---|
| `Qsgd` [3] | one-way | | | | | | |
| `ECQ-sgd` [36] | one-way | ✓ | | | | | |
| `Diana` [23] | one-way | | ✓ | | | | |
| `Dore` [19] | two-way | | ✓ | ✓ | | | degraded |
| `Double-Squeeze` [33], `Dist-EF-SGD` [38] | two-way | ✓ | | ✓ | | | degraded |
| `Artemis` [24] | two-way | | ✓ | | | | degraded |
| `MCM` | two-way | | ✓ | | ✓ | | non-degraded |
| `Rand-MCM` | two-way | | ✓ | | ✓ | ✓ | non-degraded |

computed on a mini-batch of size $b$. This function is then evaluated at point $w_k$. In the classical centralized framework (without compression), for a learning rate $\gamma$, SGD corresponds to:

$$w_{k+1} = w_k - \gamma \frac{1}{N} \sum_{i=1}^{N} g_{k+1}^i(w_k) . \tag{1}$$

We now describe the framework used for compression.

## 2.1 Bidirectional compression framework

Bidirectional compression consists in compressing communications in both directions between the central server and remote devices. We use two different compression operators, respectively $\mathcal{C}_{\text{up}}$ and $\mathcal{C}_{\text{dwn}}$ to compress the message in each direction. Roughly speaking, the update in eq. (1) becomes:

$$w_{k+1} = w_k - \gamma \mathcal{C}_{\text{dwn}} \left( \frac{1}{N} \sum_{i=1}^{N} \mathcal{C}_{\text{up}}(g_{k+1}^i(w_k)) \right) .$$

However, this approach has a major drawback. The central server receives and aggregates information $\frac{1}{N} \sum_{i=1}^{N} \mathcal{C}_{\text{up}}(g_{k+1}^i(w_k))$. But in order to be able to broadcast it back, it compresses it, *before* applying the update. We refer to this strategy as the "degraded update" approach. Its major advantage is simplicity, and it was used in all previous papers performing double compression. Yet, it appears to be a waste of valuable information. In this paper, we update the global model $w_{k+1}$ independently of the downlink compression:

$$\begin{cases} w_{k+1} = w_k - \gamma \frac{1}{N} \sum_{i=1}^{N} \mathcal{C}_{\text{up}} \left( g_{k+1}^i(\hat{w}_k) \right) . \\ \hat{w}_{k+1} = C_{\text{dwn}}(w_{k+1}) \end{cases} \tag{2}$$

However, bluntly compressing $w_{k+1}$ in eq. (2) hinders convergence, thus the second part of the update needs to be refined by adding a memory mechanism. **We now describe both communication stages of the real `MCM`, which is entirely defined by the following uplink and downlink equations.**

**Downlink**
$$\begin{cases} \Omega_{k+1} = w_{k+1} - H_k , \\ \widehat{w}_{k+1} = H_k + \mathcal{C}_{\text{dwn}}(\Omega_{k+1}) \\ H_{k+1} = H_k + \alpha_{\text{dwn}} \mathcal{C}_{\text{dwn}}(\Omega_{k+1}). \end{cases}$$

**Uplink**
$$\begin{cases} \forall i \in [\![1, N]\!], \Delta_k^i = g_{k+1}^i(\widehat{w}_k) - h_k^i \\ w_{k+1} = w_k - \frac{\gamma}{N} \sum_{i=1}^{N} \mathcal{C}_{\text{up}}(\Delta_k^i) + h_k^i \\ h_{k+1}^i = h_k^i + \alpha_{\text{up}} \mathcal{C}_{\text{up}}(\Delta_k^i). \end{cases} \tag{3}$$

**Downlink Communication.** We introduce a *downlink memory term* $(H_k)_k$, which is available on both workers and central server. The difference $\Omega_{k+1}$ between the model and this memory is compressed and exchanged, then the local model is reconstructed from this information. The memory is then updated as defined on left part of eq. (3), with a learning rate $\alpha_{\text{dwn}}$.

Introducing this memory mechanism is crucial to control the variance of the local model $\widehat{w}_{k+1}$. To the best of our knowledge `MCM` is the first algorithm that uses such a memory mechanism for downlink compression. This mechanism was introduced by Mishchenko et al. [23] for the uplink compression but with the other purpose of mitigating the impact of heterogeneity, while we use it here to avoid divergence of the local model's variance.

**Uplink Communication.** The motivation to introduce an uplink memory term $h_k^i$ for each device $i \in [\![1, N]\!]$ is different, and better understood. Indeed, for the uplink direction, this mechanism is only necessary (and then crucial) to handle heterogeneous workers [i.e., with different data distributions, see e.g. 24]. Here, the difference $\Delta_k^i$ between the stochastic gradient $g_{k+1}^i$ at the local model $\widehat{w}_k$ (as defined in eq. (3)) and the memory term is compressed and exchanged. The memory is then updated as defined on right part of eq. (3) with a rate $\alpha_{\mathrm{dwn}}$.

**Remark 1** (Rate $\alpha_{\mathrm{dwn}}$)**.** *It is necessary to use $\alpha_{\mathrm{dwn}} < 1$. Otherwise, the compression noise tends to propagate and is amplified, because of the multiplicative nature of the compression. In Figure 1 we compare* MCM, *with 3 other strategies: compressing only the update, compressing $w_k - \widehat{w}_{k-1}$, (i.e., $\alpha_{\mathrm{dwn}} = 1$), and compressing the model (i.e., $H_k = 0$), showing that only* MCM *converges.*

**Remark 2** (Memory vs Error Feedback)**.** *Error feedback is another technique, introduced by Seide et al. [29]. In the context of double compression, it has been shown to improve convergence for a restrictive class of* contracting *compression operators (which are generally biased) by Zheng et al. [38], Tang et al. [33]. However, we note several differences to our approach. (1) For unbiased operators - as considered in* Dore, *it did not lead to any theoretical improvement [Remark 2 in Sec. 4.1., 19]. (2) Moreover, only a fraction (namely $(1 + \omega_{\mathrm{dwn}})^{-1}$) of the "error" $w_{k+1} - \hat{w}_{k+1}$ can be preserved in the EF term (see line 18 in algo 1 in Liu et al.). It is thus impossible to recover the central preserved model as a function of the degraded model and the EF term. (3) [38] consider a biased operator and the same compression level for uplink and downlink compression. They also rely on stronger assumptions on the gradient (uniformly bounded) and only tackle the homogeneous case.*

In Table 1 we summarize the main algorithms for compression in distributed training. As downlink communication can be more efficient than uplink, we consider distinct operators $\mathcal{C}_{\mathrm{dwn}}, \mathcal{C}_{\mathrm{up}}$ and allow the corresponding compressions levels to be distinct: those quantities are defined in Assumption 1.

**Assumption 1.** *There exists constants $\omega_{\mathrm{up}}, \omega_{\mathrm{dwn}} \in \mathbb{R}_+^*$, such that the compression operators $\mathcal{C}_{\mathrm{up}}$ and $\mathcal{C}_{\mathrm{dwn}}$ satisfy the two following properties for all $w$ in $\mathbb{R}^d$: $\mathbb{E}[\mathcal{C}_{\mathrm{up/dwn}}(w)] = w$, and $\mathbb{E}[\|\mathcal{C}_{\mathrm{up/dwn}}(w) - w\|^2] \leq \omega_{\mathrm{up/dwn}}\|w\|^2$. The higher is $\omega$, the more aggressive the compression is.*

We only consider unbiased operators, that encompass sparsification, quantization and sketching. References and a discussion on those operators, and possible extensions of our results to biased operators are provided in Appendix A.1.

**Remark 3** (Related work on Perturbed iterate analysis)**.** *The theory of perturbed iterate analysis was introduced by Mania et al. [21] to deal with asynchronous SGD. More recently, it was used by Stich and Karimireddy [30], Gorbunov et al. [11] to analyze the convergence of algorithms with uplink compressions, error feedback and asynchrony. Using gradients at randomly perturbed points can also be seen as a form of randomized smoothing [27], a point we discuss in Appendix A.2.*

### 2.2 The randomization mechanism, `Rand-MCM`

In this subsection, we describe the key feature introduced in `Rand-MCM`: *randomization*. It consists in performing an independent compression for each device instead of performing a single one for all of them. As a consequence, each worker holds a different model centered around the global one. This introduces some supplementary randomness that stabilizes the algorithm. Formally, we will consider $N$ mutually independent compression operators $\mathcal{C}_{\mathrm{dwn},i}$ instead of a single one $\mathcal{C}_{\mathrm{dwn}}$, and the central server will send to the device $i$ at iteration $k + 1$ the compression of the difference between its model and the local memory on worker $i$: $\mathcal{C}_{\mathrm{dwn},i}(w_{k+1} - H_k^i)$. The tradeoffs associated with this modification are discussed in Section 4.

The pseudocode of `Rand-MCM` is given in Algorithm 1 in Appendix A. It incorporates all components described above: 1) the bidirectional compression, 2) the model update using the non-degraded point, 3) the two memories, 4) the up and down compression operators, 5) the randomization mechanism.

## 3 Assumptions and Theoretical analysis

We make standard assumptions on $F : \mathbb{R}^d \to \mathbb{R}$. We first assume that the loss function $F$ is smooth.

**Assumption 2** (Smoothness)**.** *$F$ is twice continuously differentiable, and is $L$-smooth, that is for all vectors $w_1, w_2$ in $\mathbb{R}^d$: $\|\nabla F(w_1) - \nabla F(w_2)\| \leq L\|w_1 - w_2\|$.*

Results in Section 3 are provided in a convex, strongly-convex and non-convex setting.

**Assumption 3** (Strong convexity)**.** *$F$ is $\mu$-strongly convex (or convex if $\mu = 0$), that is for all vectors $w_1, w_2$ in $\mathbb{R}^d$: $F(w_2) \geq F(w_1) + (w_2 - w_1)^T \nabla F(w_1) + \frac{\mu}{2}\|w_2 - w_1\|_2^2$.*

Next, we present the assumption on the stochastic gradients.

**Assumption 4** (Noise over stochastic gradients computation)**.** *The noise over stochastic gradients for a mini-batch of size b, is uniformly bounded: there exists a constant $\sigma \in \mathbb{R}_+$, such that for all k in $\mathbb{N}$, for all i in $[\![1, N]\!]$ and for all w in $\mathbb{R}^d$ we have: $E[\|g_k^i(w) - \nabla F(w)\|^2] \le \sigma^2/b$.*

We here provide guarantees of convergence for MCM. MCM incorporates an uplink memory term, designed to handle heterogeneous workers. To highlight our main contributions, that concerns the downlink compression, we present the results in the homogeneous setting, that is with $F_i = F_j$ and $\alpha_{\text{up}} = 0$. Similar results (almost identical, up to constant numerical factors) in to the heterogeneous setting are described in Appendix G. Experiments are also performed on heterogeneous workers. We provide here convergence results in the strongly-convex, then convex case.

**Notations and settings.** For $k$ in $\mathbb{N}$, we denote $\Upsilon_k = \|w_k - H_{k-1}\|^2$, and define $V_k = \mathbb{E}[\|w_k - w_*\|^2] + 32\gamma L\omega_{\text{dwn}}^2 \mathbb{E}[\Upsilon_k]$, which serves as Lyapunov function. $V_k$ is composed of two terms: the first one controls the quadratic distance to the optimal model, and the second controls the variance of the local models $\hat{w}_k$. For both theorems, we choose $\alpha_{\text{dwn}} = (8\omega_{\text{dwn}})^{-1}$. We denote $\Phi(\gamma) := (1 + \omega_{\text{up}})(1 + 64\gamma L\omega_{\text{dwn}}^2)$.

**Limit learning rate:** There exists a maximal learning rate to ensure convergence. More specifically, we define $\gamma_{\max} := \min(\gamma_{\max}^{\text{up}}, \gamma_{\max}^{\text{dwn}}, \gamma_{\max}^{\Upsilon})$, where $\gamma_{\max}^{\text{up}} := (2L(1 + \omega_{\text{up}}/N))^{-1}$ corresponds to the classical constraint on the learning rate in the unidirectional regime [see 23, 24], $\gamma_{\max}^{\text{dwn}} := (8L\omega_{\text{dwn}})^{-1}$ is a similar constraint coming from the downlink compression, and $\gamma_{\max}^{\Upsilon} := (8\sqrt{2}L\omega_{\text{dwn}}\sqrt{8\omega_{\text{dwn}} + \omega_{\text{up}}/N})^{-1}$ is a combined constraint that arises when controlling the variance term $\Upsilon$.[1] Overall, this constraints are weaker than in the "degraded" framework [19, 24], in which $\gamma_{\max}^{\text{Dore}} \le (8L(1 + \omega_{\text{dwn}})(1 + \omega_{\text{up}}/N))^{-1}$. Especially, in the regime in which $\omega_{\text{up,dwn}} \to \infty$ and $\omega_{\text{dwn}} \simeq \omega_{\text{up}} \simeq: \omega$, the maximal learning rate for MCM is $(L\omega^{3/2})^{-1}$, while it is $(L\omega^2)^{-1}$ in [19, 24]. Our $\gamma_{\max}$ is thus larger by a factor $\sqrt{\omega}$, see Table 2. We define $\widetilde{L}$ such that $\gamma_{\max} = (2\widetilde{L})^{-1}$.

**Theorem 1** (Convergence of MCM in the homogeneous and strongly-convex case)**.** *Under Assumptions 1 to 4 with $\mu > 0$, for k in $\mathbb{N}$, for any sequence $(\gamma_k)_{k \ge 0} \le \gamma_{\max}$ we have:*

$$V_k \le (1 - \gamma_k\mu)V_{k-1} - \gamma_k\mathbb{E}[F(\widehat{w}_{k-1}) - F(w_*)] + \frac{\gamma_k^2\sigma^2\Phi(\gamma_k)}{Nb}, \tag{4}$$

*Consequently, (1) if $\sigma^2 = 0$ (noiseless case), for $\gamma_k \equiv \gamma_{\max}$ we recover a linear convergence rate: $\mathbb{E}[\|w_k - w_*\|^2] \le (1 - \gamma_{\max}\mu)^k V_0$; (2) if $\sigma^2 > 0$, taking for all K in $\mathbb{N}$, $\gamma_K = 2/(\mu(K+1) + \widetilde{L})$, for the weighted Polyak-Ruppert average $\bar{w}_K = \sum_{k=1}^K \lambda_k w_{k-1} / \sum_{k=1}^K \lambda_k$, with $\lambda_k := (\gamma_{k-1})^{-1}$,*

$$\mathbb{E}[F(\bar{w}_K) - F(w_*)] \le \frac{\mu + 2\widetilde{L}}{4\mu K^2}\|w_0 - w_*\|^2 + \frac{4\sigma^2(1 + \omega_{\text{up}})}{\mu K N b}\left(1 + \frac{64L\omega_{\text{dwn}}^2}{\mu K}\ln(\mu K + \widetilde{L})\right). \tag{5}$$

**Limit Variance (Equation (4)).** For a constant $\gamma$, the variance term (i.e., term proportional to $\sigma^2$) in Equation (4) is upper bounded by $\frac{\gamma^2\sigma^2}{Nb}(1 + \omega_{\text{up}})(1 + 64\gamma L\omega_{\text{dwn}}^2)$. The impact of the downlink compression is attenuated by a factor $\gamma$. As $\gamma$ decreases, this makes the limit variance similar to the one of Diana, i.e., without downlink compression [23, Eq. 16 in Th. 2] and much lower than the variance for previous algorithms using double compression for which the variance scales quadratically with the compression constants as $\gamma^2\sigma^2(1 + \omega_{\text{up}})(1 + \omega_{\text{dwn}})/N$: (1) for Dore, see Corollary 1 in Liu et al. [19] (who indicate $(1 - \rho)^{-1} \ge (1 + \omega_{\text{up}}/N)(1 + \omega_{\text{dwn}})$), (2) for Artemis see Table 2 and Th. 3 point 2 in [24], (3) for [11], see Theorem I.1. (with $\gamma D_1' \propto \gamma^2\sigma^2(1 + \omega_{\text{up}})(1 + \omega_{\text{dwn}})/N$).

Bound 5 has a quadratic dependence on $\omega_{\text{dwn}}$, but the corresponding term is divided by an extra factor $K$, the number of iterations. For example in experiments, for *w8a* using quantization with $s = 2^0$, we have $\omega_{\text{dwn}} \simeq 17$, and after only 50 epoch with a batch size $b = 12$, we have $K \simeq 2500$. Hence, the term $\omega^2/K$ is vanishing through iterations and we asymptotically recover a rate of convergence equivalent to algorithms using unidirectional compression.

---

[1] The dependency in $\omega^{3/2}$ is similar to the one obtained by Horváth et al. [15] in unidirectional compression in the non-convex case (Theorem 4).

**Convergence and complexity:** With a decaying sequence of steps, we obtain a convergence rate scaling as $O(K^{-1})$ in Equation (5), without dependency on the $\omega_{\mathrm{dwn}}$ in the dominating term, which only appears in faster decaying terms scaling as $K^{-2}$. The iteration complexity (i.e., number of iterations to achieve $\epsilon$ expected error) is thus at first order $O_{\epsilon \to 0}(\frac{\sigma^2(1+\omega_{\mathrm{up}})}{\mu \epsilon N b})$. Again, this matches the complexity of Diana [15, see Theorem 1 and Corollary 1] and is smaller by a factor $1 + \omega_{\mathrm{dwn}}$ than the one of `Artemis`, `Dore`, `DIANAsr-DQ` (see Corollary I.1. in [11]). Next, we give a convergence result in the convex case.

**Theorem 2** (Convergence of `MCM`, convex case)**.** *Under Assumptions 1 to 4 with $\mu = 0$. For all $k > 0$, for any $\gamma \leq \gamma_{\max}$, we have, for $\bar{w}_k = \frac{1}{k}\sum_{i=0}^{k-1} w_i$,*

$$\gamma \mathbb{E}\left[F(w_{k-1}) - F(w_*)\right] \leq V_{k-1} - V_k + \frac{\gamma^2 \sigma^2 \Phi(\gamma)}{Nb} \implies \mathbb{E}[F(\bar{w}_k) - F_*] \leq \frac{V_0}{\gamma k} + \frac{\gamma \sigma^2 \Phi(\gamma)}{Nb}. \quad (6)$$

*Consequently, for $K$ in $\mathbb{N}$ large enough, a step-size $\gamma = \sqrt{\frac{\|w_0 - w_*\|^2 Nb}{(1+\omega_{\mathrm{up}})\sigma^2 K}}$, we have:*

$$\mathbb{E}[F(\bar{w}_K) - F_*] \leq 2\sqrt{\frac{\|w_0 - w_*\|^2 (1 + \omega_{\mathrm{up}})\sigma^2}{NbK}} + O(K^{-1}). \quad (7)$$

*Moreover if $\sigma^2 = 0$ (noiseless case), we recover a faster convergence: $\mathbb{E}[F(\bar{w}_K) - F_*] = O(K^{-1})$.*

**Limit Variance (Eq. (6)).** The variance term is identical to the strongly-convex case.

**Convergence and complexity (Equation (7)).** The downlink compression constant only appears in the second-order term, scaling as $1/K$. In other words, the convergence rate is equivalent to the convergence rate of Diana, in the non-strongly-convex. As $K$ increases, this complexity scales as $\frac{(1+\omega_{\mathrm{up}})}{n\epsilon^2}$ independently of the downlink compression. Again, for previous algorithms with double compression the complexity is at least $O\left(\frac{(1+\omega_{\mathrm{up}})(1+\omega_{\mathrm{dwn}})}{n\epsilon^2}\right)$ (see Corollary I.2 in [11]).

**Control of the variance of the local model.**

We here present the backbone Lemma of `MCM`'s proof. It allows to control the variance of the local model $\mathbb{E}[\|\hat{w}_k - w_k\|^2 \,|w_k]$ (which is upper-bounded by $\omega_{\mathrm{dwn}}\mathbb{E}[\|\Upsilon_k\|^2 \,|w_k]$) and to build the Lyapunov function defined in Theorems 1 and 2.

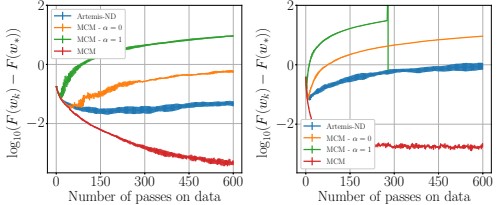

This result highlights the impact of the downlink memory term. Without memory, i.e., with $\alpha_{\mathrm{dwn}} = 0$, the variance of the local model $\|\hat{w}_k - w_k\|^2$ increases with the number of iterations. On the other hand, if $\alpha_{\mathrm{dwn}}$ is too large (close to 1), this variance diverges. This behavior is illustrated on two real datasets on Figure 1. This phenomenon is similar to the divergence observed in frameworks involving error feedback, when the compression operator is not contractive.

Figure 1: Comparing `MCM` on two datasets with three other algorithms using a non-degraded update, $\gamma = 1/L$. `Artemis-ND` stands for `Artemis` with a non-degraded update.

**Theorem 3.** *Consider the `MCM` update as in eq. (2). Under Assumptions 1, 2 and 4 with $\mu = 0$, if $\gamma \leq (8\omega_{\mathrm{dwn}}L)^{-1}$ and $\alpha \leq (4\omega_{\mathrm{dwn}})-1$, then for all $k$ in $\mathbb{N}$:*

$$\mathbb{E}\left[\Upsilon_k\right] \leq \left(1 - \frac{\alpha_{\mathrm{dwn}}}{2}\right) \mathbb{E}\left[\Upsilon_{k-1}\right] + 2\gamma^2 \left(\frac{1}{\alpha_{\mathrm{dwn}}} + \frac{\omega_{\mathrm{up}}}{N}\right) \mathbb{E}\left[\|\nabla F(\hat{w}_{k-1})\|^2\right] + \frac{2\gamma^2 \sigma^2(1 + \omega_{\mathrm{up}})}{Nb}.$$

This bound provides a recursive control on $\Upsilon_k$. Beyond the $(1 - \alpha_{\mathrm{dwn}})$ contraction, the bound comprises the squared-norm of the gradient at the previous perturbed iterate, and a noise term.

**Summary of rates.** In Table 2, we summarize the rates and complexities, and maximal learning rate for Diana, Artemis, Dore and MCM. For simplicity, we ignore absolute constants, and provide asymptotic values for large $\omega_{\mathrm{up}}$, $\omega_{\mathrm{dwn}}$, and complexities for $\epsilon \to 0$.

**Proof in the heterogeneous case.** To extend Theorems 1 to 3 in the heterogeneous setting for a convex objective (Appendix G), we assume that there exists a constant $B$ in $\mathbb{R}_+$, s.t.:

Table 2: Summary of rates on the initial condition, limit variance, asympt. complexities and $\gamma_{\mathrm{max}}$.

| Problem | | Diana | Artemis, Dore | MCM, Rand-MCM |
|---|---|---|---|---|
| | $L\gamma_{\mathrm{max}} \propto$ | $1/(1+\omega_{\mathrm{up}})$ | $1/(1+\omega_{\mathrm{up}})(1+\omega_{\mathrm{dwn}})$ | $1/(1+\omega_{\mathrm{dwn}})\sqrt{1+\omega_{\mathrm{up}}} \wedge 1/(1+\omega_{\mathrm{up}})$ |
| | Lim. var. $\propto \gamma^2\sigma^2/n\times$ | $(1+\omega_{\mathrm{up}})$ | $(1+\omega_{\mathrm{up}})(1+\omega_{\mathrm{dwn}})$ | $(1+\omega_{\mathrm{up}})(1+\gamma L\omega_{\mathrm{dwn}}^2)$ |
| Str.-convex | Rate on init. cond. (SC) | $(1-\gamma\mu)^k$ | $(1-\gamma\mu)^k$ | $(1-\gamma\mu)^k$ |
| | Complexity | $(1+\omega_{\mathrm{up}})/\mu\epsilon N$ | $(1+\omega_{\mathrm{dwn}})(1+\omega_{\mathrm{up}})/\mu\epsilon N$ | $(1+\omega_{\mathrm{up}})/\mu\epsilon N$ |
| Convex | Complexity | $(\omega_{\mathrm{up}}+1)/\epsilon^2$ | $(1+\omega_{\mathrm{up}})(1+\omega_{\mathrm{dwn}})/\epsilon^2$ | $(\omega_{\mathrm{up}}+1)/\epsilon^2$ |

$\frac{1}{N}\sum_{i=0}^{N}\|\nabla F_i(w_*)\|^2 = B^2$. We further define $\Xi_k = \frac{1}{N^2}\sum_{i=1}^{N}\|h_k^i - \nabla F_i(w_*)\|^2$, where for all $i$ in $[\![1, N]\!]$. This term is recursively controled [23, 24] and combined into the Lyapunov function.

**Proofs.** To convey the best understanding of the theorems and the spirit of the proof, we introduce a `Ghost` algorithm (impossible to implement) in Appendix D.1. A sketch of the proof describes the main steps in the case of `Ghost`, those steps are similar for `MCM`. Fundamentally, our proof relies on a tight analysis, related to perturbed iterate analysis [21]. Proofs of Theorems 1 to 3 are given in Appendix E. Th. S11 in Appendix E.4 ensures convergence for a non-convex $F$. Note that the proof for non-convex follows a different approach than the one in Theorems 1 and 2.

As mentioned in the introduction, our analysis of perturbed iterate in the context of double compression opens new directions: in particular, it opens the door to handling a different model for each worker. In the next section, we detail those possibilities, and provide theoretical guarantees for `Rand-MCM`, the variant of `MCM` in which instead of sending the same model to all workers, the compression noises are mutually *independent*.

**Remark 4** (Communication budget). *How to split a given communication budget between uplink and downlink to optimize the convergence is an open question which is intrinsically related to the situation. Indeed it depends on many factors like the selected operators of compression, the upload/downlink speed or the number of participating workers at each iteration. However, our approach provides some insights on this question. Because asymptotically the impact of double compression is marginal, for a fixed budget, Theorem 2 suggests to strongly compress on the downlink direction (which leads to a large $\omega_{dwn}$), but to perform a weaker compression in the uplink direction.*

## 4 Extension to `Rand-MCM`

### 4.1 Communication and convergence trade-offs

In `Rand-MCM`, we leverage the fact that the compressions used for each worker need not to be identical. On the contrary, it is possible to consider *independent* compressions. By doing so, we reduce the impact of the downlink compression.

The relevance of such a modification depends on the framework: while the convergence rate will be improved, the computational time can be slightly increased. Indeed, $N$ compressions need to be computed instead of one: however, this computational time is typically not a bottleneck w.r.t. the communication time. A more important aspect is the communication cost. While the size of each message will remain identical, a different message needs to be sent to each worker. That is, we go from a "one to $N$" configuration to $N$ "one to one" communications. While this is a drawback, it is not an issue when the bandwidth/transfer time are the bottlenecks, as `Rand-MCM` will result in a better convergence with almost no cost. Furthermore, we argue that handling worker dependent models is essential for several major applications. `Rand-MCM` can directly be adapted to those frameworks.

**1. Worker dependent compression.** A first simple situation is the case in which workers are allowed to choose the size (or equivalently the compression level) of their updates.

**2. Partial participation (PP).** Similarly, having $N$ different messages to send to each worker may be unavoidable in the case of *partial participation* of the workers. This is a key feature in Federated Learning frameworks [22]. In the classical distributed framework (without downlink constraints) it is easy to deal with it, as each available worker just queries the global model to compute its gradient on it [see for example 14]. On the other hand, for bidirectional compression, to ensure that all the local models match the central model, the adaptation to partial participation relies on a *synchronization step*. During this step, each worker that has not participated in the last $S$ steps receives the last $S$ corresponding messages as long as it costs less to send this sequence than a full uncompressed model. This is described in the description of the adaptation to partial participation in [24], in the remark preceding Eq. (20) in [26] and by Tang et al. [33, v2 on arxiv for the distributed case], who use

a buffer. On the contrary, `Rand-MCM` naturally handles a different model, memory and update per worker. The adaptation to partial participation is thus straightforward. Though theoretical results are out of the scope of this paper, we provide experiments on PP in Appendix B.1.1 and fig. 4.

One drawback is the necessity to store the $N$ memories $(H_k^i)_{i \in [N]}$ instead of one, which results in an additional memory cost. To circumvent this issue we propose two independent solutions. 1) Keep and use a single memory $\bar{H}_k = N^{-1} \sum_{i=1}^{N} H_k^i$ (as suggested in [24]). It is then necessary to periodically reset the local memories $H_k^i$ on all workers to the averaged value $\bar{H}_k$ (rarely enough not to impact the communication budget). This is illustrated in fig. 4. 2) Use `Rand-MCM` with an arbitrary number of groups $G \ll N$ of workers. In each group $\mathcal{G}_g$, $g \in [G]$, all workers share the same memory $(H_k^g)$ and receive the same update $\mathcal{C}_{\mathrm{dwn},g}(w_{k+1} - H_k^g)$. We call this algorithm `Rand-MCM-G`.

**Remark 5** (Protecting the global model from honest-but-curious clients). *Another business advantage of `MCM` and `Rand-MCM` is that providing degraded models to the participants can be used to guarantee privacy, or to ensure the workers participate in good faith, and not only to obtain the model. This issue of detecting ill-intentioned clients (free-riders) that want to obtain the model without actually contributing has been studied by Fraboni et al. [10].*

### 4.2 Theoretical results

In this Section, we provide two main theoretical results for `Rand-MCM`. First Theorem 4 ensures that the theoretical guarantees are at least as good for `Rand-MCM` as for `MCM`. Then, in Theorem 5, we provide convergence result for both `MCM` and `Rand-MCM` in the case of quadratic functions.

**Theorem 4.** *Theorems 1 to 3 are valid for `Rand-MCM` and `Rand-MCM-G`.*

The improvement in `Rand-MCM` comes from the fact that we are ultimately averaging the gradients at several random points, reducing the variance coming from this aspect. The goal is obviously to reduce the impact of $\omega_{\mathrm{dwn}}$. Keeping in mind that the dominating term in the rate is independent of $\omega_{\mathrm{dwn}}$, *we can thus only expect to reduce the second-order term*. Next, the uplink compression noise increases with the variance of the randomized model, which will not be directly reduced by `Rand-MCM`. As a consequence, we only expect the improvement to be visible in the part of the second-order term that does not depend on $\omega_{\mathrm{up}}$ (that is, the effect would be the most significant if $\omega_{\mathrm{up}}$ is small or 0).

This intuition is corroborated by the following result, in which we show that the convergence is improved when adding the randomization process for a quadratic function. Extending the proof beyond quadratic functions is possible, though it requires an assumption on third or higher order derivatives of $F$ (e.g., using self-concordance [4]) to control of $\mathbb{E}\left[ ||\nabla F(\widehat{w}_{k-1}) - \mathbb{E}[\nabla F(\widehat{w}_{k-1})]||^2 \mid w_{k-1} \right]$.

**Theorem 5** (Convergence in the quadratic case). *Under Assumptions 1 to 4 with $\mu = 0$, if the function is quadratic, after running $K > 0$ iterations, for any $\gamma \leq \gamma_{\max}$, and we have*

$$\mathbb{E}[F(\bar{w}_K) - F_*] \leq \frac{V_0}{\gamma K} + \frac{\gamma \sigma^2 \Phi^{\mathrm{Rd}}(\gamma)}{Nb} \, ,$$

*with $\Phi^{\mathrm{Rd}}(\gamma) = (1 + \omega_{\mathrm{up}}) \left( 1 + \frac{4\gamma^2 L^2 \omega_{\mathrm{dwn}}}{K} \left( \frac{1}{\mathbf{C}} + \frac{\omega_{\mathrm{up}}}{N} \right) \right)$ and $\mathbf{C} = N$ for `Rand-MCM`, $\mathbf{C} = G$ `Rand-MCM-G`, and $\mathbf{C} = 1$ for `MCM`.*

This result is derived in Appendix F. We can make the following comments: (1) The convergence rate for quadratic functions is slightly better than for smooth functions. More specifically, the right hand term in $\Phi$ is multiplied by an additional $\gamma \left( \frac{1}{\mathbf{C}} + \frac{\omega_{\mathrm{up}}}{N} \right)$ (w.r.t. Theorem 2), which is decaying at the same rate as $\gamma$. Besides, the proof for `Rand-MCM` is substantially modified, as $\mathbb{E}[\nabla F(\widehat{w}_{k-1})]$ is an unbiased estimator of $\nabla F(w_{k-1})$. (2) Moreover, the randomization in `Rand-MCM` (resp. `Rand-MCM-G`) further reduces by a factor $N$ (resp. $G$) this term. Depending on the relative sizes of $\omega_{\mathrm{up}}$ and $N$, this can lead to a significant improvement up to a factor of $N$. In practice the impact of `Rand-MCM` is noticeable, as illustrated in the following experiments.

## 5 Experiments

In this section, we illustrate the validity of the theoretical results given in the previous section on both synthetic and real datasets, on (1) least-squares linear regression (LSR), (2) logistic regression (LR), and (3) non-convex deep learning. We compare `MCM` with classical algorithms used in distributed settings: `Diana`, `Artemis`, `Dore` and of course the simplest setting - `SGD`, which is the baseline.

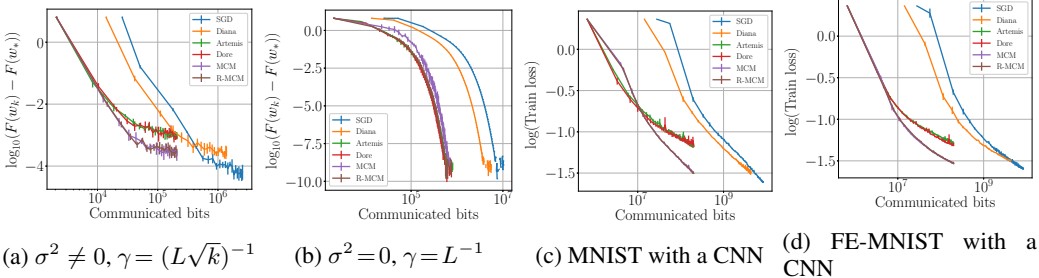

(a) $\sigma^2 \neq 0, \gamma = (L\sqrt{k})^{-1}$  (b) $\sigma^2 = 0, \gamma = L^{-1}$  (c) MNIST with a CNN  (d) FE-MNIST with a CNN

Figure 2: Convergence on neural networks.

In these experiments, we provide results on the log of the excess loss $F(w_k) - F_*$, averaged on 5 runs (resp. 2) in convex settings (resp. deep learning), with errors bars displayed on each figure (but not in the "zoom square"), corresponding to the standard deviation of $\log_{10}(F(w_k) - F_*)$. On Figure 3, the X-axis is respectively the number of iterations and the number of bits exchanged.

Each experiment has been run with $N = 20$ workers using stochastic scalar quantization [3], w.r.t. 2-norm. To maximize compression, we always quantize on a single level ($s = 2^0$), unless for PP ($s = 2^1$) and neural network (the value of $s$ depends on the dataset).

We used 9 different datasets.

- One toy dataset devoted to linear regression in an homogeneous setting. This toy dataset allows to illustrate MCM properties in a simple framework, and in particular to ilustrate that when $\sigma^2 = 0$, we recover a linear convergence[2], see Figure 2b.
- Five datasets commonly used in convex optimization (a9a, quantum, phishing, superconduct and w8a); see Table S1 for more details. Experiments were conducted with heterogeneous workers obtained by clustering (using *TSNE* [20]) the input points.
- Four dataset in a non-convex settings (CIFAR10, Fashion-MNIST, FE-MNIST, MNIST); see Table S2 for more details.

All experiments are performed without any tuning of the algorithms, (e.g., with the same learning rate for all algorithms and without reducing it after a certain number of epochs). Indeed, our goal is to show that our method achieves a performance close to the unidirectional-compression framework (Diana), while performing an important downlink compression. More details about experiments can be found in Appendix B.

On Figure 3, we display the excess loss for quantum and a9a w.r.t. the number of iteration and number of communicated bits. The plots of phising, superconduct and w8a are not provided but can be found on our github repository. We only report their excess loss after 450 iterations in Table 3.

Table 3: MCM- convex experiments, $b$ is the batch size

| Excess loss after 450 epochs | SGD | Diana | MCM | Dore | Ref |
|---|---|---|---|---|---|
| a9a ($b = 50$) | $-3.5$ | $-2.7$ | $-2.7$ | $-1.8$ | [8] |
| quantum ($b = 400$) | $-3.4$ | $-3.2$ | $-3.2$ | $-2.6$ | [7] |
| phishing ($b = 50$) | $-3.7$ | $-3.5$ | $-3.4$ | $-2.7$ | [8] |
| superconduct ($b = 50$) | $-1.6$ | $-1.6$ | $-1.55$ | $-1.45$ | [13] |
| w8a ($b = 12$) | $-3.5$ | $-3.0$ | $-2.5$ | $-1.75$ | [8] |
| Compression | no | uni-dir | bi-dir | bi-dir | |

**Saturation level.** All experiments are performed with a *constant learning rate* $\gamma$ to observe the bias (initial reduction) and the variance (saturation level) independently. Stochastic gradient descent results in a fast convergence during the first iterations, and then reaches a saturation at a given level proportional to $\sigma^2$. Theorem 2 states that the variance of MCM is proportional to $\omega_{\text{up}}$, this is experimentally observed on Tables 3 and 4 and figs. 2 and 3: MCM meets Diana while Artemis and Dore saturate at a higher level (scaling as $\omega_{\text{up}} \times \omega_{\text{dwn}}$). These trade-offs are preserved with optimized learning rates.

---

[2]Even stronger, we show in experiments that we recover a linear rate if we have $\sigma_* = 0$ (the noise over stochastic gradient computation at the optimum point $w_\star$).

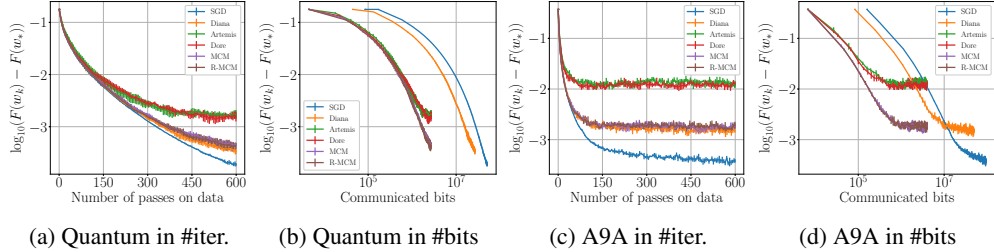

| (a) Quantum in #iter. | (b) Quantum in #bits | (c) A9A in #iter. | (d) A9A in #bits |

Figure 3: Experiments on real dataset with $\gamma = 1/L$, quantization with $s = 1$, LSR (a,b), LR (c,d).

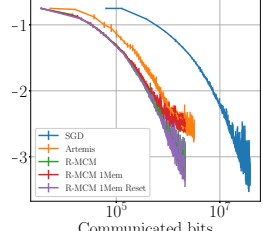

**Linear convergence when $\sigma^2 = 0$.** The six algorithms present a linear convergence when $\sigma^2 = 0$. This is illustrated by Figure 2b: we ran experiments with a full gradient descent. Note that in these settings MCM has a slightly worse performance than other methods; however, this slow-down is compensated by `Rand-MCM`.

**Impact of randomization.** The impact of randomization is noticeable on Figures 2b and S5b. Randomization helps to stabilise convergence of it reduces the variance of the runs and when $\sigma^2 = 0$, it performs identically to `SGD`. Figure 4 illustrates the impact of using a single memory, instead of $N$, to alleviate the memory cost in the PP setting (Subsection 4.1), with or without periodic reset. Without reset, performance are slightly degraded, but with it, we recover previous results.

Figure 4: `Rand-MCM` (PP) on quantum with a *single memory* ($s = 2$).

**Deep learning.** Table 4 and figs. 2c and 2d illustrate experiments with neural networks, details on dataset settings and networks architecture are given in Appendix B.2. Again, MCM meets `Diana` rates as stated by Theorem S11 (theorem in the non-convex case).

Table 4: Accuracy and train loss in non-convex experiments, detailed settings can be found in Table S2.

|  | Algorithm | MNIST | Fashion MNIST | FE-MNIST | CIFAR-10 |
|---|---|---|---|---|---|
| Accuracy after | SGD: | 99.0% | 92.4% | 99.0% | 69.1% |
| 300 epochs | Diana: | 98.9% | 92.4% | 98.9% | 64.0% |
|  | MCM: | 98.8% | 90.6% | 98.9% | 63.5% |
|  | Artemis: | 97.9% | 86.7% | 98.3% | 54.8% |
|  | Dore: | 97.9% | 87.9% | 98.5% | 56.3% |
| Train loss after | SGD: | 0.025 | 0.093 | 0.026 | 0.909 |
| 300 epochs | Diana: | 0.034 | 0.141 | 0.031 | 1.047 |
|  | MCM: | 0.033 | 0.209 | 0.030 | 1.096 |
|  | Artemis: | 0.075 | 0.332 | 0.052 | 1.342 |
|  | Dore: | 0.072 | 0.300 | 0.048 | 1.292 |

Overall, these experiments show the benefits of MCM and `Rand-MCM`, that reach the saturation level of `Diana` while exchanging at 10x to 100x fewer bits. More experiments with partial participation for `Rand-MCM` are given in Appendix B.1.1. All the code is provided on our github repository.

## 6 Conclusion

In this work, we propose a new algorithm to perform bidirectional compression while achieving the convergence rate of algorithms using compression in a single direction. One of the main application of this framework is Federated Learning. With MCM we stress the importance of not degrading the global model. In addition, we add the concept of randomization which allows to reduce the variance associated with the downlink compression. The analysis of MCM is challenging as the algorithm involves perturbed iterates. Proposing such an analysis is the key to unlocking numerous challenges in distributed learning, e.g., proposing practical algorithms for partial participation, incorporating privacy-preserving schemes *after* the global update is performed, dealing with local steps, etc. This approach could also be pivotal in non-smooth frameworks, as it can be considered as a weak form of randomized smoothing.

## Acknowledgments

We would like to thank Richard Vidal, Laeticia Kameni from Accenture Labs (Sophia Antipolis, France) and Eric Moulines from École Polytechnique for insightful discussions. This research was supported by the *SCAI: Statistics and Computation for AI* ANR Chair of research and teaching in artificial intelligence, by *Hi!Paris*, and by *Accenture Labs* (Sophia Antipolis, France).

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
