_{\mathrm{dwn}} = (8\omega_{\mathrm{dwn}})^{-1}$. We denote $\Phi(\gamma) := (1 + \omega_{\mathrm{up}})\left(1 + 64\gamma L\omega_{\mathrm{dwn}}^2\right)$.

**Limit learning rate:** There exists a maximal learning rate to ensure convergence. More specifically, we define $\gamma_{\max} := \min(\gamma_{\max}^{\mathrm{up}}, \gamma_{\max}^{\mathrm{dwn}}, \gamma_{\max}^{\Upsilon})$, where $\gamma_{\max}^{\mathrm{up}} := (2L(1 + \omega_{\mathrm{up}}/N))^{-1}$ corresponds to the classical constraint on the learning rate in the unidirectional regime [see 34, 36], $\gamma_{\max}^{\mathrm{dwn}} := (8L\omega_{\mathrm{dwn}})^{-1}$ is a similar constraint coming from the downlink compression, and $\gamma_{\max}^{\Upsilon} := \left(8\sqrt{2}L\omega_{\mathrm{dwn}}\sqrt{8\omega_{\mathrm{dwn}} + \omega_{\mathrm{up}}/N}\right)^{-1}$ is a combined constraint that arises when controlling the variance term $\Upsilon$.[1] Overall, this constraints are weaker than in the "degraded" framework [29, 36], in which $\gamma_{\max}^{\mathrm{Dore}} \leq \left(8L(1 + \omega_{\mathrm{dwn}})(1 + \omega_{\mathrm{up}}/N)\right)^{-1}$. Especially, in the regime in which $\omega_{\mathrm{up,dwn}} \to \infty$ and $\omega_{\mathrm{dwn}} \simeq \omega_{\mathrm{

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

# Supplementary material

In this appendix, we provide additional details about our work. First, in Appendix A we give complementary references on operators of compression and on perturbed iterate analysis. We also give the pseudo-code of `Rand-MCM`. Secondly, in Appendix B we enlarge figures provided in Section 5 and complete them with experiments on partial participation and with a comparison between `MCM` and other algorithms using non-degraded updates. The next sections are all devoted to theoretical results. In Appendix C we detail some technical results required to demonstrate Theorems 1 to 5, in Appendix D we highlight the key stages of the demonstration in the easier case of `Ghost`, in Appendix E we completely prove the given guarantees of convergence in three regimes: convex, strongly-convex and non-convex. In Appendix F we show the benefit of `Rand-MCM` compared to `MCM` in the context of quadratic functions. In Appendix G we adapt the proof to the heterogeneous scenario. And finally, in Appendix H we answer to the Neurips checklist.

# Contents

## A  Complementary discussions and references

We give the pseudo-code of `Rand-MCM` in Algorithm 1. It summarizes the algorithm's description given in Section 1.

### A.1  Compression Operators

In this section, we give additional details on compression operators (see Assumption 1).

Operators of compression can be biased or unbiased and they may have drastically different impacts on convergence. For instance, if the operator is not contracting, algorithms with error-feedback may diverge. Horváth and Richtárik [17] propose a method to unbiase a biased operator and a general study of biased operator has been carried out by Beznosikov et al. [6]. But in this work, as stated by Assumption 1, we consider only unbiased operators: for instance s-quantization.

The choice of the operator of compression is crucial when compressing data. Operators of compression may be classified into three mains categories: 1) sparsification [43, 19, 22, 4, 22, 32] 2) quantization [41, 53, 3, 18, 48] and 3) sketching [20, 27].

**Possible Extensions** Our analysis could be extended to biased uplink operators, following similar lines of proof as [6].

The extension for the downlink operator seems more difficult as our analysis

---

**Algorithm 1** Pseudocode of `Rand-MCM`

**Input:** Mini-batch size $b$, learning rates $\alpha_{\mathrm{up}}, \alpha_{\mathrm{dwn}}, \gamma > 0$, initial model $w_0 \in \mathbb{R}^d$ (on all devices), operators $\mathcal{C}_{\mathrm{up}}$ and $\mathcal{C}_{\mathrm{dwn}}$, $S = [\![1, N]\!]$ the set of devices.
**Init.:** Memories: $\forall i \in S, h_0^i = g_1^i(w_0)$ and $H_{-1}^i = w_0$
**Output:** Model $w_K$
**for** $k = 1, 2, \ldots, K$ **do**
  **for** each device $i = 1, 2, 3, \ldots, N$ **do**
    Receive $\widehat{\Omega}_{k-1}^i$, and set: $w_{k-1}^i = \widehat{\Omega}_{k-1}^i + H_{k-2}^i$
    Compute $g_k^i(w_{k-1}^i)$ (with mini-batch)
    Update down memory: $H_{k-1}^i = H_{k-2}^i + \alpha_{\mathrm{dwn}}\widehat{\Omega}_{k-1}^i$
    Up compr.: $\widehat{\Delta}_{k-1}^i = \mathcal{C}_{\mathrm{up}}(g_k^i(w_{k-1}^i) - h_{k-1}^i)$
    **Update uplink memory**: $h_k^i = h_{k-1}^i + \alpha_{\mathrm{up}}\widehat{\Delta}_{k-1}^i$

    Send $\widehat{\Delta}_{k-1}^i$ to central server
  **end for**
  Receive $(\widehat{\Delta}_{k-1}^i)_{i=1}^N$ from all remote servers
  Compute $\widehat{g}_k = \frac{1}{N}\sum_{i=1}^N \widehat{\Delta}_{k-1}^i + h_{k-1}^i$
  Update up memory: $\forall i \in S, h_k^i = h_{k-1}^i + \alpha_{\mathrm{up}}\widehat{\Delta}_{k-1}^i$

  Non-degraded update: $w_k = w_{k-1} - \gamma\widehat{g}_k$
  Down compr.: $\forall i \in S, \widehat{\Omega}_k^i = \mathcal{C}_{\mathrm{dwn},i}(w_k - H_{k-1}^i)$
  **Update downlink memory:** $H_k^i = H_{k-1}^i + \alpha_{\mathrm{dwn}}\widehat{\Omega}_k^i$

  Send $(\widehat{\Omega}_k^i)_{i=1}^N$ to all remote servers
**end for**

---

relies on numerous occurrences on the fact that the expectation of $\widehat{w}_{k-1}$ knowing $w_{k-1}$ is $w_{k-1}$.

### A.2 Relation to Randomized Smoothing

Our approach can also be related to randomized smoothing. Formally, $\nabla F(\widehat{w}_{k-1})$ can be considered as an unbiased gradient of the smoothed function $F_\rho$ at point $w_{k-1}$, with $F_\rho : w \mapsto \mathbb{E}[F(w + \widehat{w}_{k-1} - w_{k-1})]$. Then $\mathbb{E}\langle \nabla F(\widehat{w}_{k-1}), w_{k-1} - w_* \rangle = \mathbb{E}\langle \nabla F_\rho(w_{k-1}), w_{k-1} - w_* \rangle$. One key aspect is that the condition number $\mu_\rho / L_\rho$ of $F_\rho$ is always larger (better) than the one for $F$. However, the minimum of $F_\rho$ is different and moving, thus the proof techniques from Randomized smoothing are not adapted to a varying noise which distribution is unknown. Providing a theoretical result that quantifies the smoothing impact of `MCM` is an interesting open direction.

Randomized smoothing has been applied to non-smooth problems by Duchi et al. [12]. The aim is to transform a non-smooth function into a smooth function, before computing the gradient. This is achieved by adding a Gaussian noise to the point where the gradient is computed. This mechanism has been applied by Scaman et al. [39] to convex problems. We consider in this work a randomized version of compression: at iteration $k$ in $\mathbb{N}$ each worker $i$ in $[\![1, N]\!]$ receives a noisy estimate $\widehat{w}_k^i$ of the global model $w_k$ kept on central server. Thus, we compute the local gradient at a perturbed point $w_k + \delta_k^i$. Unlike the randomization process as defined by Duchi et al. [12], the noise here is not chosen to improve the function's regularity but results from the compression.

## B Experiments

In this section we provide additional details about our experiments. We first give the settings of our experiments in Tables S1 and S2. Next, we describe the numerical results obtained on our 9 datasets. Thirdly, we add some explanation concerning the wall clock time. Finally, we provide an estimation of the carbon footprint required by this paper.

We use the same operator of compression for uplink and downlink, thus we consider that $\omega_{\mathrm{up}} = \omega_{\mathrm{dwn}}$. In addition, we choose $\alpha_{\mathrm{up}} = \alpha_{\mathrm{dwn}} = \dfrac{1}{2(1 + \omega_{\mathrm{up/dwn}})}$.

**Convex settings** are given in Table S1. We obtain non-i.i.d. data distributions by computing a TSNE representation [defined in 30] followed by a clustering. Experiments have been performed with 600 epochs. Apart from the case of partial participation, we use quantization [defined in 3] with $s = 2^0$.

Table S1: Settings of experiment in the convex mode.

| Settings | a9a | quantum | phishing | superconduct | w8a |
|---|---|---|---|---|---|
| references | [10] | [9] | [10] | [16] | [10] |
| model | LR | LR | LSR | LR | LR |
| dimension $d$ | 124 | 66 | 69 | 82 | 301 |
| training dataset size | $32,561$ | $50,000$ | $11,055$ | $21,200$ | $49,749$ |
| batch size $b$ | 50 | 400 | 50 | 50 | 12 |
| compression rate $s$ | $2^0$ (*i.e.* two levels) | | | | |
| norm quantization | $\|\cdot\|_2$ | | | | |
| momentum $m$ | no momentum | | | | |
| step size $\gamma$ | $1/L$ | | | | |

**Deep-learning settings** are provided in Table S2. All experiments have been performed with 300 epochs

Table S2: Settings of experiments in the non-convex mode.

| Settings | MNIST | Fashion-MNIST | FE-MNIST | CIFAR10 |
|---|---|---|---|---|
| references | [26] | [50] | [8] | [24] |
| model | CNN | Fashion CNN | CNN | LeNet |
| trainable parameters $d$ | $20 \times 10^3$ | $400 \times 10^3$ | $20 \times 10^3$ | $62 \times 10^3$ |
| training dataset size | $60,000$ | $60,000$ | $805,263$ | $60,000$ |
| compression rate $s$ | $2^2$ | $2^2$ | $2^2$ | $2^4$ |
| momentum $m$ | 0 | 0 | 0 | 0.9 |
| norm quantization | $\|\cdot\|_2$ | | | |
| batch size $b$ | 128 | | | |
| step size $\gamma$ | 0.1 | | | |
| loss | Cross Entropy | | | |

## B.1 Convex settings

In this section, we provide the plot of excess loss for the toy dataset, for quantum and for a9a datasets. For results on superconduct, phishing and w8a, see our github repository. For these last three datasets, we give only the excess loss w.r.t. number of iteration in the basic settings of full participation on Figure S5. We detail experiments in the PP settings in Appendix B.1.1. At the left side (resp. right side) we display the result w.r.t. the number of iterations (resp. number of communicated bits).

We provide results on the log of the excess loss $F(w_k) - F_*$, with error bars displayed on each figure, corresponding to the standard deviation of $\log_{10}(F(w_k) - F_*)$. Figures S1b, S2b, S3 and S4 correspond to Figures 2a, 2b and 3 given in Section 5. Additionally, we provide results for the synthetic dataset (Figures 2a and 2b) w.r.t to the number of iterations in Figure S1 (stochastic gradient) and Figure S2 (full batch gradient). As predicted by Theorem 2, when $\sigma = 0$, we observe a linear convergence.

On Figure S6, we present a9a, quantum and w8a with a different operator of compression than in all other experiments. We use random unbiased sparsification: each coordinate has a likelihood $p = 0.1$ to be selected.

### B.1.1 Experiments on partial participation

In this subsection, we run the experiments in a setting where only *half of devices* (independently picked at each iteration) are available at each iteration, thus simulating a setting of partial participation.

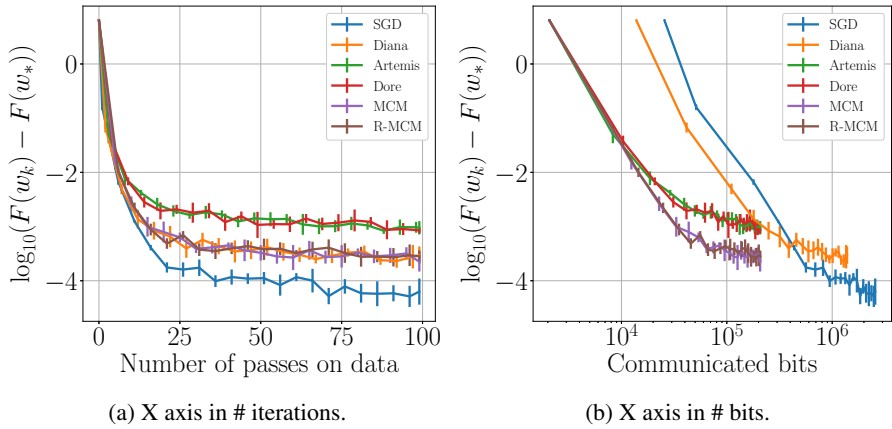

(a) X axis in # iterations.

(b) X axis in # bits.

Figure S1: Least-square regression, toy dataset: $\gamma = (L\sqrt{k})^{-1}$, $\sigma \neq 0$.

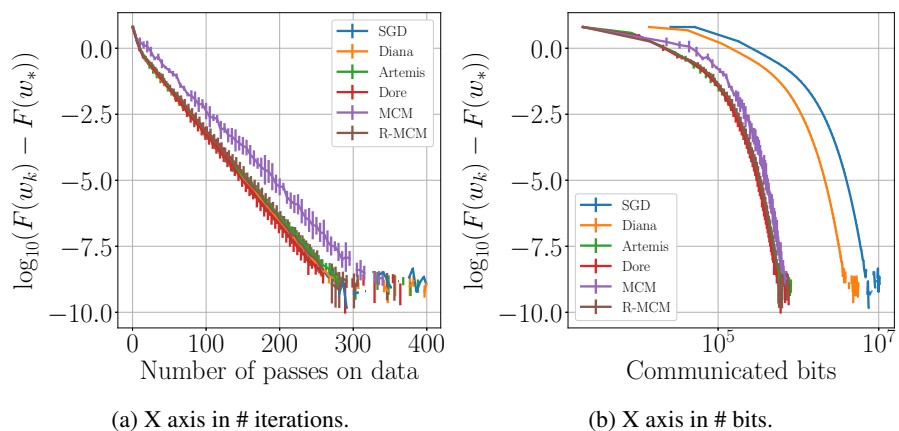

(a) X axis in # iterations.

(b) X axis in # bits.

Figure S2: Least-square regression, toy dataset: $\gamma = 1/L$, $\sigma_*^2 = 0$.

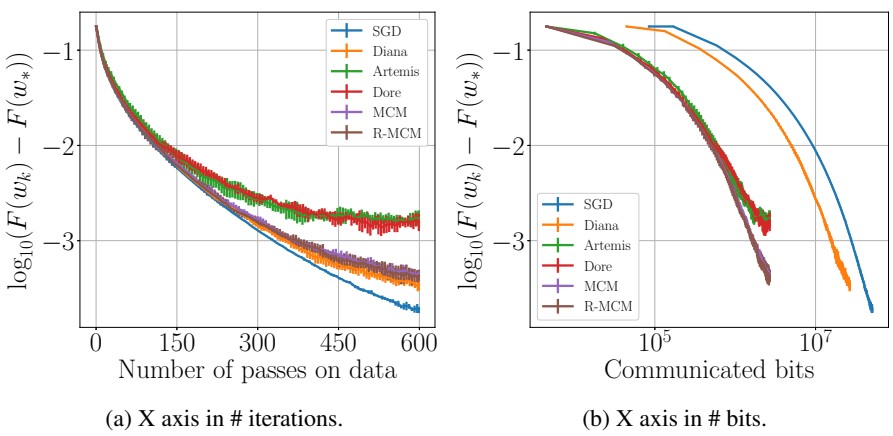

(a) X axis in # iterations.

(b) X axis in # bits.

Figure S3: quantum with $b = 400$, $\gamma = 1/L$.

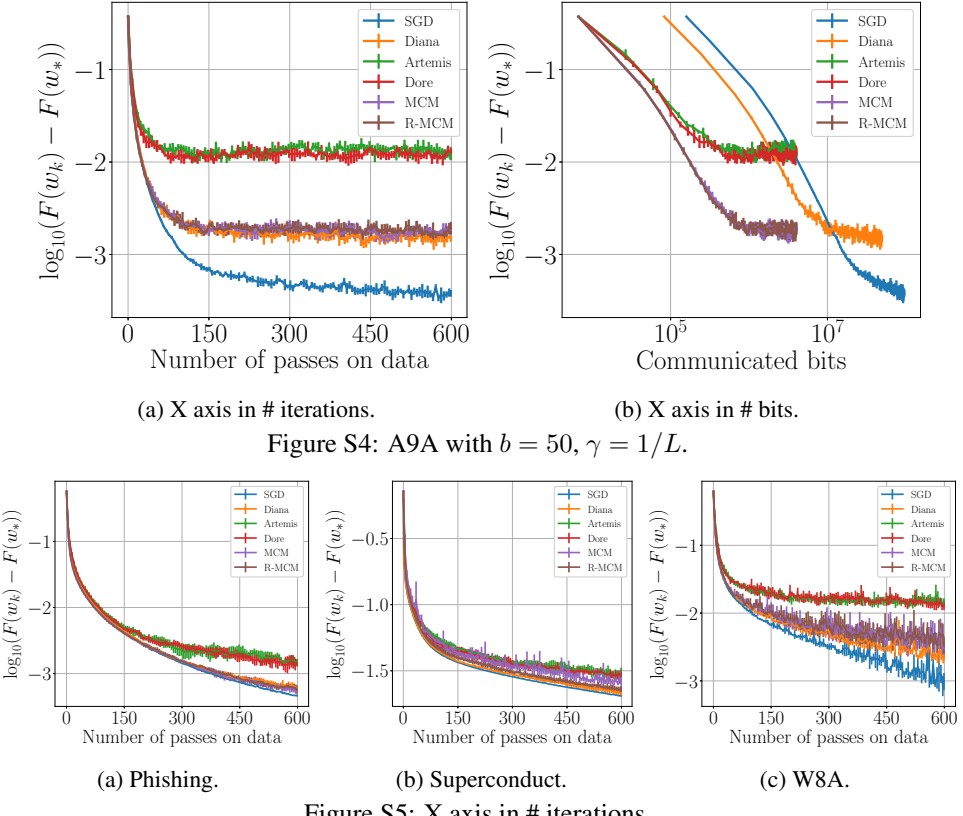

(a) X axis in # iterations.  (b) X axis in # bits.

Figure S4: A9A with $b = 50$, $\gamma = 1/L$.

(a) Phishing.  (b) Superconduct.  (c) W8A.

Figure S5: X axis in # iterations.

Figures S7 and S8 present the results for respectively quantum and A9A. For these experiments, we used a 2-quantization compression. We do not plot `MCM` on these figures because in a context of partial participation, `Rand-MCM` is the natural thing to do. Indeed in this context, we must hold a memory for each worker, and thus the compressed vector sent to each worker is unique.

We observe that partial participation leads to an increase of the variance for all algorithms. Furthermore, we can observe on both Figures S7b and S8b that `Rand-MCM` outperforms `Artemis` and `Dore` not only in term of convergence but also in term of communication cost. This is because `Rand-MCM` does not require the synchronization step, at which any active nodes receive any update it has missed. This saves a few communication rounds. In these settings, the level of saturation of `SGD`, `Diana` and `Rand-MCM` seems to be almost identical, this fact stresses again the benefit of our designed algorithm.

Additionally, we present on Figures S9 and S10 the impact of only using a single averaged downlink memory term instead of $N$ distinct memories. More details about update equations are given in Equation (S1). We display three versions of `Rand-MCM` that we compare to the SGD-baseline and to `Artemis`:

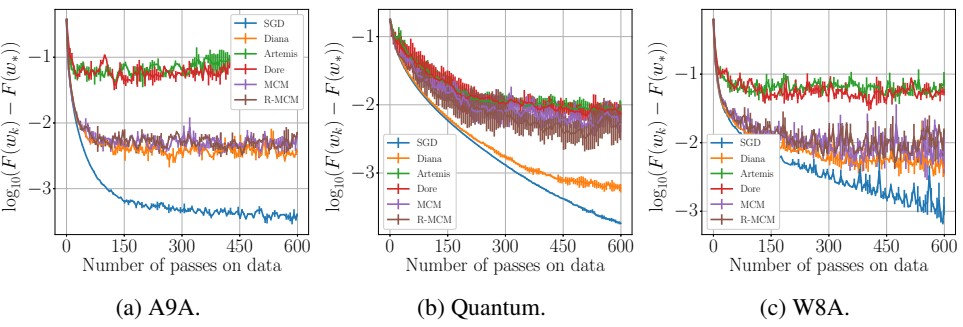

(a) A9A.  (b) Quantum.  (c) W8A.

Figure S6: X axis in # iterations using random sparsification with $p = 0.1$.

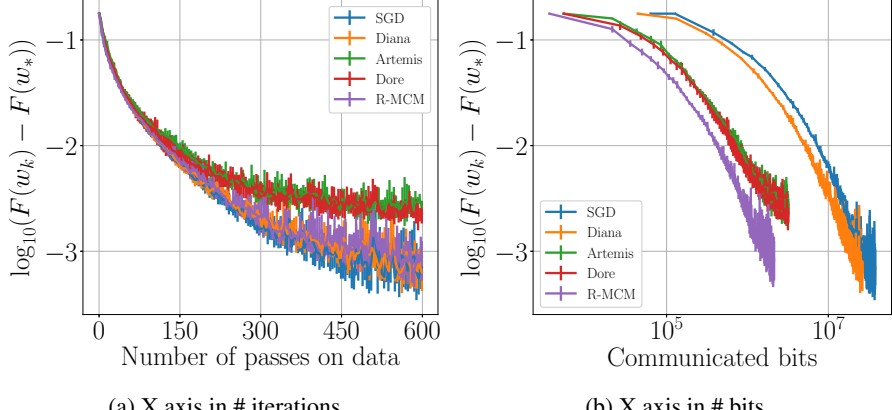

(a) X axis in # iterations.

(b) X axis in # bits.

Figure S7: quantum with $b = 400$, $\gamma = 1/L$ and a 2-quantization. Only half of the devices are participating at each round.

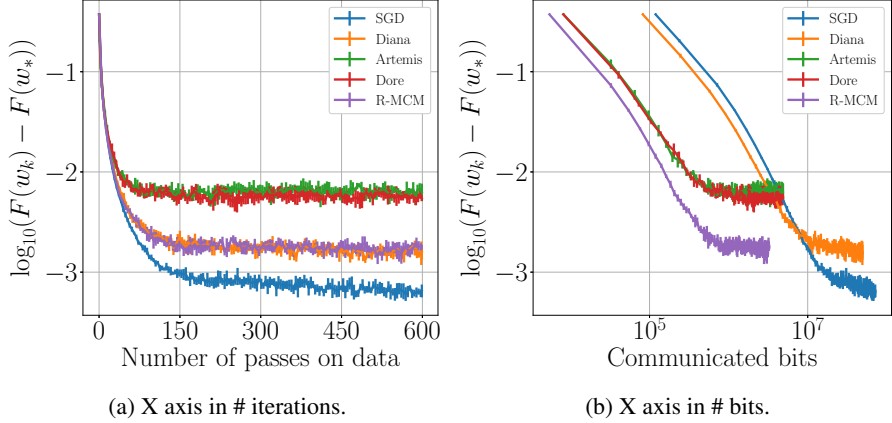

(a) X axis in # iterations.

(b) X axis in # bits.

Figure S8: A9A with $b = 50$, $\gamma = 1/L$ and a 2-quantization. Only half of the devices are participating at each round.

1. The standard `Rand-MCM`, using $N$ downlink memories,

2. `Rand-MCM` with a single memory, without any periodically reset.

3. `Rand-MCM` with both a single memory and a reset of the downlink memory every $4\sqrt{d}$ iterations, where $d$ is the dimension of the optimization problem. This allows to limit the increase of communicated bits. Indeed as we use quantization with $s = 1$, each communication costs $32 \times \sqrt{d} \log(d)$ bits instead of $32 \times d$. Because every $4\sqrt{d}$ iterations we send the uncompressed downlink memory term, there is an additional cost of $\frac{32d}{4\sqrt{d}}$. At the end, the memory reset leads to send $32 \times \sqrt{d}(\log(d) + 1/4)$ bits by iterations instead of $32 \times \sqrt{d} \log(d)$ bits for `Rand-MCM`(without reset). The increase is thus marginal.

For sake of clarity, we present below the two versions of `Rand-MCM`. In the first version, the central server holds $N$ memories that exactly correspond to those kept on the $N$ remote devices. In the second version, the central server holds a single memory $\bar{H}_k = \frac{1}{N} \sum_{i=1}^{N} H_k^i$ and each worker $i$ holds there own memory $H_k^i$.

$N$ **memories**

$$\begin{cases} \Omega_{k+1}^i = w_{k+1} - H_k^i, \\ \widehat{w}_{k+1}^i = H_k^i + \mathcal{C}_{\text{dwn},i}(\Omega_{k+1}^i) \\ H_{k+1}^i = H_k^i + \alpha_{\text{dwn}} \mathcal{C}_{\text{dwn},i}(\Omega_{k+1}^i). \end{cases}$$

1 **memories**

$$\begin{cases} \Omega_{k+1} = w_{k+1} - \bar{H}_k, \\ \widehat{w}_{k+1}^i = H_k^i + \mathcal{C}_{\text{dwn},i}(\Omega_{k+1}) \\ H_{k+1}^i = H_k^i + \alpha_{\text{dwn}} \mathcal{C}_{\text{dwn},i}(\Omega_{k+1}) \\ \bar{H}_{k+1} = \bar{H}_k + \frac{\alpha_{\text{dwn}}}{N} \sum_{i=1}^{N} \mathcal{C}_{\text{dwn},i}(\Omega_{k+1}). \end{cases}$$

(S1)

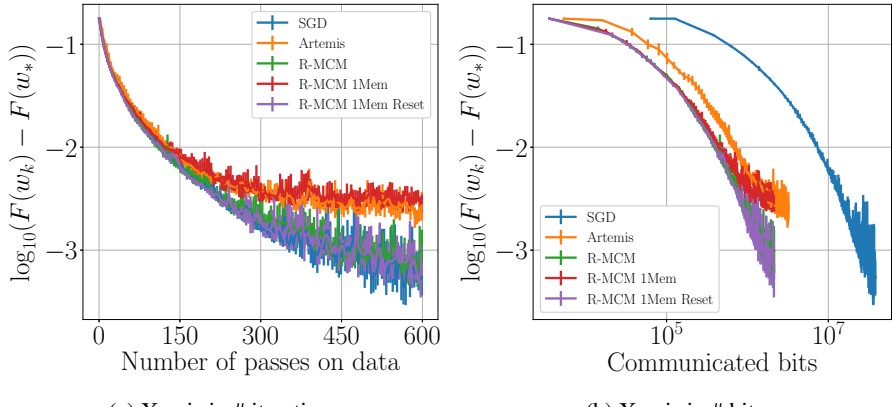

(a) X axis in # iterations.

(b) X axis in # bits.

Figure S9: quantum with $b = 400$, $\gamma = 1/L$ and a 2-quantization. Only half of the devices are participating at each round.

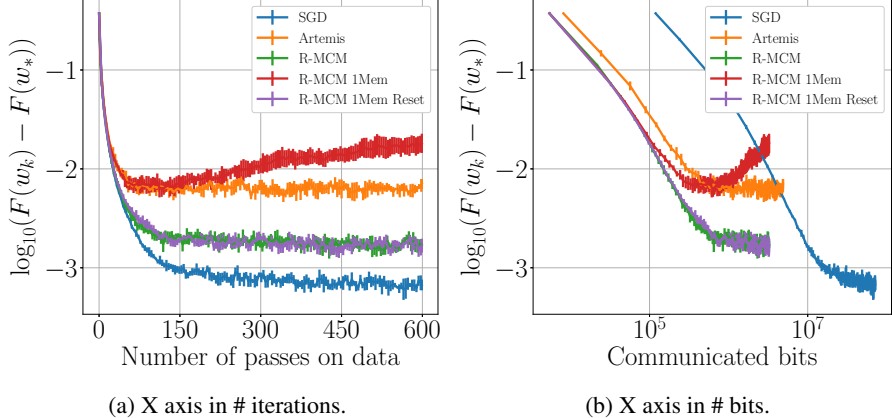

(a) X axis in # iterations.

(b) X axis in # bits.

Figure S10: A9A with $b = 50$, $\gamma = 1/L$ and a 2-quantization. Only half of the devices are participating at each round.

In this experiments, it is noticeable that using single-downlink-memory-`Rand-MCM` without periodic reset makes the algorithms saturate at a high level with an important variance. But as soon as we introduce the reset, we recover previous rates.

### B.1.2   Comparing `MCM` with other algorithm using non-degraded update

The aim of this section is to show the importance to set $\alpha < 1$, for this purpose we compare `MCM` with three other algorithms:

1. `Artemis` with a non-degraded update i.e. unlike the version proposed by Philippenko and Dieuleveut [36], we do not update the global model with the compression sent to all remote nodes. *It means that we compress only the update that has already been performed on the global server.* It corresponds to:

$$\begin{cases} \forall i \in [\![1, N]\!], \Delta_k^i = g_{k+1}^i(\widehat{w}_k) - h_k^i \\ w_{k+1} = w_k - \frac{\gamma}{N} \sum_{i=1}^N \mathcal{C}_{\text{up}}(\Delta_k^i) + h_k^i \\ \widehat{w}_{k+1} = \widehat{w}_k - \gamma \mathcal{C}_{\text{dwn}}\left(\frac{1}{N} \sum_{i=1}^N \mathcal{C}_{\text{up}}(\Delta_k^i) + h_k^i\right) \\ h_{k+1}^i = h_k^i + \alpha_{\text{up}} \mathcal{C}_{\text{up}}(\Delta_k^i). \end{cases}$$

2. `MCM` with $\alpha = 0$, *thus without memory.*

3. `MCM` with $\alpha = 1$, in other words, for $k$ in $\mathbb{N}^*$ it corresponds to the case $H_{k+1} = \widehat{w}_{k+1}$. Indeed by definition we have $H_{k+1} = H_k + \alpha \widehat{\Omega}_{k+1}$, and furthermore, when we rebuild the compressed model on remote device, we have: $\widehat{w}_{k+1} = \widehat{\Omega}_{k+1} + H_k$. *In this case, we use the compressed model as memory.*

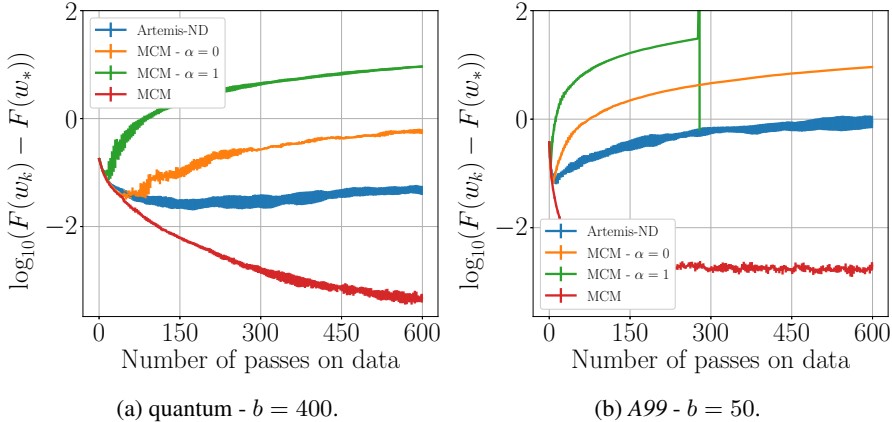

(a) quantum - $b = 400$.    (b) $A99$ - $b = 50$.

Figure S11: Comparing MCM with three other algorithms using a non-degraded update, $\gamma = 1/L$. Artemis-ND stands for Artemis with a non-degraded update.

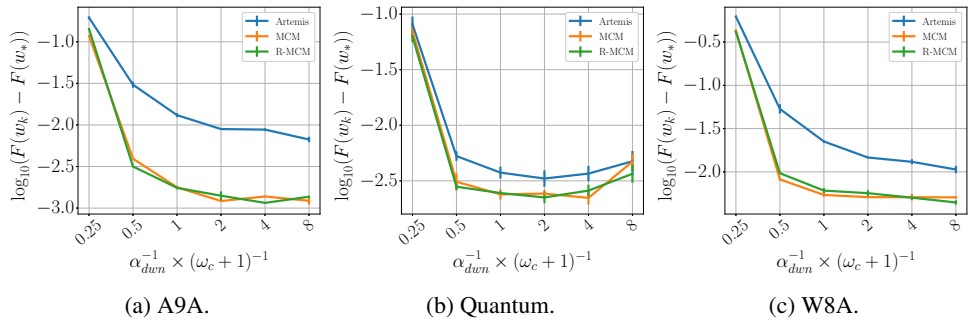

(a) A9A.    (b) Quantum.    (c) W8A.

Figure S12: On X axis is displayed different values of $\dfrac{1}{\alpha(\omega_{\mathrm{dwn}} + 1)}$. On Y axis is given the excess loss after 250 epochs. In all other experiments, we choose $\alpha_{\mathrm{dwn}} = \dfrac{1}{2(\omega_{\mathrm{dwn}} + 1)} (= \alpha_{\mathrm{up}})$.

Figures S11a and S11b clearly show the superiority of MCM over the three other variants. Some conclusions can be drawn from the observation of these figures.

- MCM without downlink memory (orange curve, $\alpha = 0$) does not converge. As stressed in Subsection 2.1, this mechanism is crucial to control the variance of the local model $w_{k+1}$, for $k$ in $\mathbb{N}$.

- Intuitively, while it appears reasonable to consider as memory the model that has been compressed at the previous step, experiments (green curves) show that this is not the case in practice and that $\alpha$ must be small enough to ensure convergence. This is the *noise explosion* phenomenon that was mentioned earlier in the paper.

- Compressing only the update gives reasonable results (blue curve). However, the convergence saturates at a higher level than for MCM.

### B.1.3 Impact of the learning rate $\alpha$

On Figure S12, we plot the value of the excess loss obtained after 250 epochs w.r.t. to the value of $\frac{1}{2(1+\omega_{\mathrm{up/dwn}})}$. We observe that if $\alpha$ is too big, MCM converges slowly; but after reaching a threshold, the value of $\alpha$ does not impact anymore the rate of convergence. This confirms theory that suggests to use the largest possible $\alpha_{dwn}$ but smaller than a given value. The condition $\alpha_{\mathrm{dwn}} \leq \frac{1}{4(\omega_{\mathrm{dwn}}+1)}$ results from the proofs of Theorems S8 and S14. But because the constant 4 is partially an artifact of the proof, in experiments we used $\alpha_{\mathrm{dwn}} = \frac{1}{2(\omega_{\mathrm{dwn}}+1)}$ as in [36] (see condition S19 in Theorem S7), and this choice is confirmed by Figure S12.

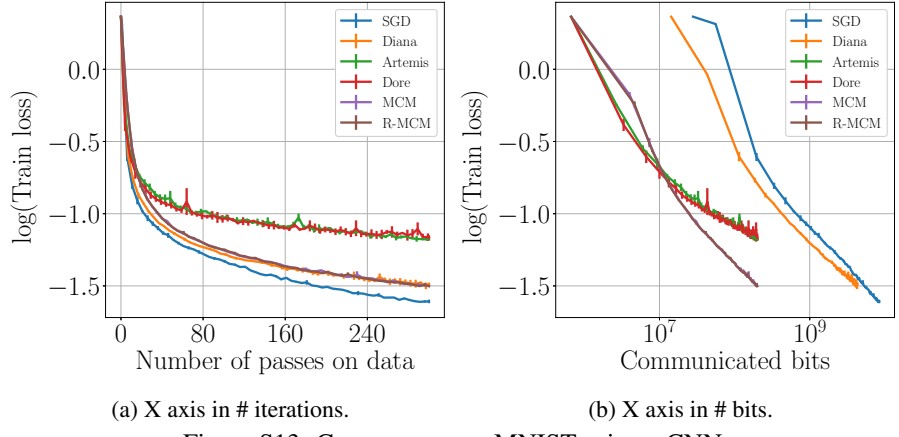

(a) X axis in # iterations.       (b) X axis in # bits.

Figure S13: Convergence on MNIST using a CNN.

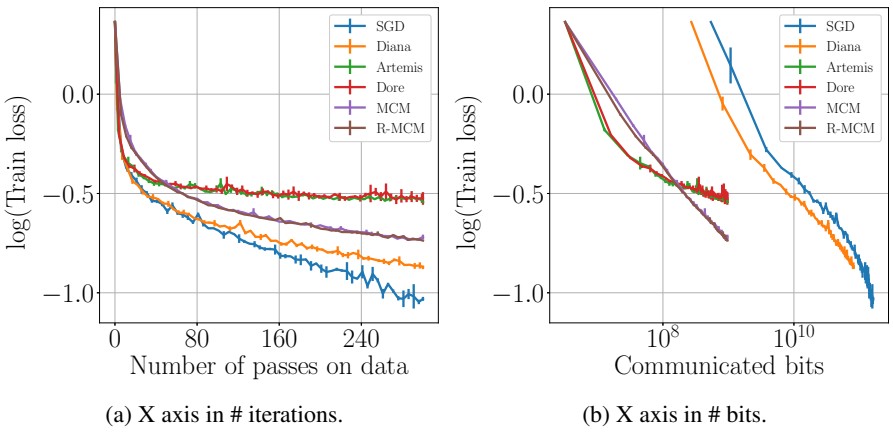

(a) X axis in # iterations.       (b) X axis in # bits.

Figure S14: Convergence on Fashion-MNIST.

## B.2 Experiments in deep learning

In this section, we show the robustness of `MCM` in high dimension using more complex data and applying the algorithm to non-convex problems (see Theorem S11 for a guarantee of convergence in this scenario). We carried out experiments on MNIST/FE-MNIST/Fashion-MNIST using a CNN (Figures S13 to S15), and on CIFAR using the LeNet model (Figure S16). We plot the logarithm of the train loss w.r.t the number of iterations and the number of communicated bits. The accuracy has been given in Section 5, see Table 4. Settings of the experiments can be found in Table S2, all experiments are averaged over 2 runs.

As for experiments in convex case, `MCM` presents identical rates of convergence than `Diana` but with a small shift that makes `Artemis` better during the first iterations.

## B.3 Wall clock time

We verified in our experiments that the downlink compression of $w_k - H_{k-1}$ on the central server does not lead to a noticeable overhead w.r.t. gradients computation and communications. Here, as experiments are performed in a *simulated environment* there is no communication cost. In Table S3 we report the computation time when training on FE-MNIST, this allows to highlight that compression only marginally increases the computation cost.

## B.4 Hardware and Carbon footprint

As part as a community effort to report the carbon footprint of experiments, we describe in this subsection the hardware used and the total computation time.

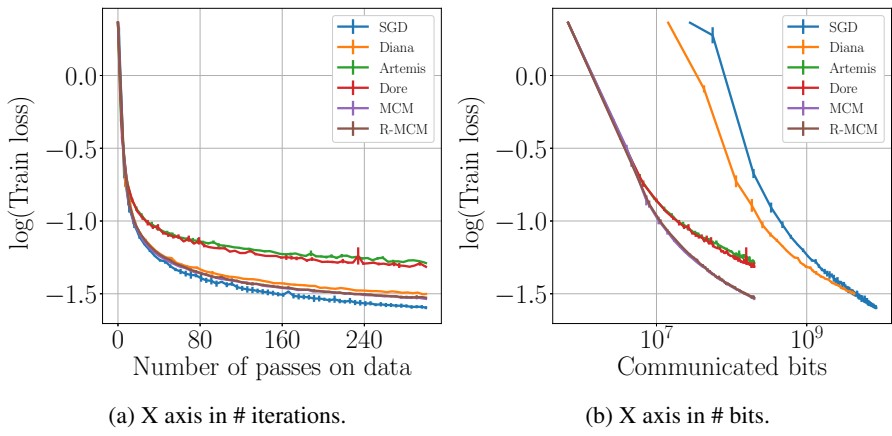

(a) X axis in # iterations.  (b) X axis in # bits.

Figure S15: Convergence on FE-MNIST.

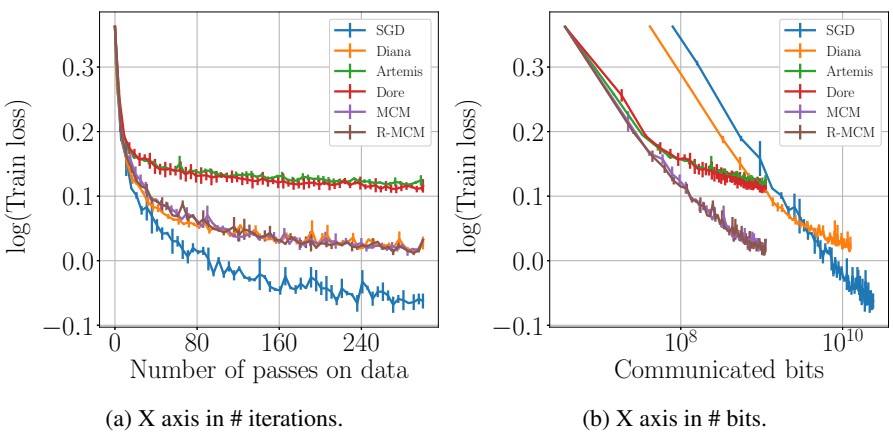

(a) X axis in # iterations.  (b) X axis in # bits.

Figure S16: Convergence on CIFAR10.

Table S3: Wall clock time on FE-MNIST with $b = 128$ and $s = 2^2$.

| Compression regime | Computation time for 150 epoch |
|---|---|
| No compression (SGD) | 15421s |
| Compression on uplink | 16773s, ratio: 1.08 |
| Compression on uplink and downlink | 16769s, ratio: 1.08 |

We have two kind of experiments : for deep learning models we ran experiments on a GPU, and for linear/logistic regression on a CPU. We used an Intel(R) Xeon(R) CPU E5-2667 processor with 16 cores; and we used an Nvidia Tesla V100 GPU with 4 nodes.

To generate all figures in this paper, our code ran (if run in a sequential mode) for 150 hours on a CPU. In overall, we consider that the whole paper writing process required (code development, debugging, exploring settings ...) at least 6000 hours end to end on the CPU. The carbon emissions caused by this work were subsequently evaluated with the `Green Algorithm`, built by Lannelongue et al. [25]. It estimates our computations to generate around 100kg of $CO_2$, requiring 2.5MWh. To compare, this corresponds to about 570km by car.

On the GPU, experiments require to be ran for around 140 hours (if run in a sequential mode). In overall, we consider that the full paper writing process required at least 2800 hours end to end on the GPU. The `Green Algorithm` estimates our computations to generate 220kg of $CO_2$, requiring 5.7MWh. To compare, this corresponds to about $1,270$km by car.

# C  Technical results

In this section, we provide some technical results required by our demonstration. In Appendix C.1 we recall classical inequalities and in Appendix C.2 we present two preliminary lemmas.

In Appendices C to E, for ease of notation we denote, for $k$ in $\mathbb{N}^*$, $\widetilde{g}_k = \frac{1}{N} \sum_{i=1}^{N} \widehat{g}_k^i(\widehat{w}_{k-1})$. Furthermore we use the convention $\nabla F(w_{-1}) = 0$.

## C.1  Basic inequalities

In this subsection, we recall some very classical inequalities, for all $a, b \in \mathbb{R}^d$, $\beta > 0$ we have:

$$\langle\, a,\, b\,\rangle \leq \frac{\|a\|^2}{2\beta} + \frac{\beta \|b\|^2}{2}\,, \tag{S2}$$

$$\|a + b\|^2 \leq (1 + \frac{1}{\beta}) \|a\|^2 + (1 + \beta) \|a\|^2\,, \tag{S3}$$

$$\|a + b\|^2 \leq 2 \left(\|a\|^2 + \|b\|^2\right)\,, \tag{S4}$$

$$|\langle\, a,\, b\,\rangle| \leq \|a\| \cdot \|b\| \qquad \text{(Cauchy-Schwarz inequality)}\,, \tag{S5}$$

$$\langle\, a,\, b\,\rangle \leq \frac{1}{2} \left(\|a\|^2 + \|b\|^2 - \|a - b\|^2\right) \qquad \text{(Polarization identity)}. \tag{S6}$$

Below, we recall Jensen's inequality.

**Jensen inequality**  Let a probability space $(\Omega, \mathcal{A}, \mathbf{P})$ with $\Omega$ a sample space, $\mathcal{A}$ an event space, and $\mathbf{P}$ a probability measure. Suppose that $X : \Omega \longrightarrow \mathbb{R}^d$ is a random variable, then for any convex function $f : \mathbb{R}^d \longrightarrow \mathbb{R}$ we have:

$$f\left(\mathbb{E}(X)\right) \leq \mathbb{E}f(X)\,. \tag{S7}$$

The next lemma will be used several times in the proofs.

**Lemma S1.** *Let a probability space $(\Omega, \mathcal{A}, \mathbf{P})$ with $\Omega$ a sample space, $\mathcal{A}$ an event space, $\mathbf{P}$ a probability measure, and $\mathcal{F}$ a $\sigma-$algebra. For any $a \in \mathbb{R}^d$ and for any random vector in $\mathbb{R}^d$ we have:*

$$\mathbb{E}\left[\|X - \mathbb{E}X\|^2\right] \leq \mathbb{E}\left[\|X - a\|^2\right]$$

*indeed $\mathbb{E}[X] = \arg\min_{a \in \mathbb{R}^d} \mathbb{E}\left[\|X - a\|^2\right]$. Similarly, for any random vector $Y$ in $\mathbb{R}^d$ which is $\mathcal{F}$-measurable, we have:*

$$\mathbb{E}\left[\|X - \mathbb{E}\left[X \mid \mathcal{F}\right]\|^2 \,\Big|\, \mathcal{F}\right] \leq \mathbb{E}\left[\|X - Y\|^2 \,\Big|\, \mathcal{F}\right]\,.$$

**Assumption 5** (Cocoercivity). *We suppose that for all $k$ in $\mathbb{N}$, stochastic gradients functions $(g_k^i)_{i\in[\![1,N]\!]}$ are L-cocoercive in quadratic mean. That is, for $k$ in $\mathbb{N}$, $i$ in $[\![1, N]\!]$ and for all vectors $w_1, w_2$ in $\mathbb{R}^d$, we have:*

$$\mathbb{E}[\|g_k^i(w_1) - g_k^i(w_2)\|^2] \leq L\,\langle\, \nabla F_i(w_1) - \nabla F_i(w_2),\, w_1 - w_2\,\rangle\,.$$

*This assumption is stronger than supposing convexity and L-smoothness of F.*

The final proposition of this subsection presents two inequalities used in our demonstrations when invoking convexity or strong-convexity. They follow from Assumption 3 and can be found in [35].

**Proposition S1.** *If a function $F$ is convex, then it satisfies for all $w$ in $\mathbb{R}^d$:*

$$\langle\, \nabla F(x),\, w - w_*\,\rangle \geq \frac{1}{2}(F(w) - F(w_*)) + \frac{1}{2L} \|\nabla F(w)\|^2\,. \tag{S8}$$

*If a function $F$ is strongly-convex, then it satisfies for all $w$ in $\mathbb{R}^d$:*

$$\langle\, \nabla F(x),\, w - w_*\,\rangle \geq \frac{1}{2}(F(w) - F(w_*)) + \frac{1}{2} \left(\mu \|w - w_*\|^2 + \frac{1}{L} \|\nabla F(w)\|^2\right)\,. \tag{S9}$$

## C.2 Two lemmas

In this subsection, we give two lemmas required to prove the convergences of `Ghost`[3], `MCM` and `Rand-MCM`.

The first lemma will be used to show that `MCM` indeed satisfies Theorem 3. The proof is straightforward from the definition of $w_k$ and $H_{k-1}$.

**Lemma S2** (Expectation of $w_k - H_{k-1}$). *For any $k$ in $\mathbb{N}^*$, the expectation of $(w_k - H_{k-1})$ conditionally to $w_{k-1}$ can be decomposed as follows:*

$$\mathbb{E}\left[w_k - H_{k-1} \mid w_{k-1}\right] = (1 - \alpha_{\mathrm{dwn}})(w_{k-1} - H_{k-2}) - \gamma\mathbb{E}\left[\nabla F(\widehat{w}_{k-1}) \mid w_{k-1}\right].$$

*Proof.* Let $k$ in $\mathbb{N}^*$, by definition and with Assumption 1:

$$\mathbb{E}\left[w_k - H_{k-1} \mid w_{k-1}\right] = \mathbb{E}\left[w_{k-1} - \gamma\widehat{g}_k(\widehat{w}_{k-1}) - (H_{k-2} + \alpha_{\mathrm{dwn}}\mathcal{C}(w_{k-1} - H_{k-2})) \mid w_{k-1}\right]$$
$$= (w_{k-1} - H_{k-2}) - \alpha_{\mathrm{dwn}}\mathbb{E}\left[\mathcal{C}(w_{k-1} - H_{k-2}) \mid w_{k-1}\right] - \gamma\mathbb{E}\left[\widetilde{g}_k \mid w_{k-1}\right],$$

from which the result follows.

$\square$

The following lemma provides a control of the impact of the uplink compression. It decomposes the squared-norm of stochastic gradients into two terms: 1) the true gradient 2) the variance of the stochastic gradient $\sigma^2$.

**Lemma S3** (Squared-norm of stochastic gradients). *For any $k$ in $\mathbb{N}^*$, the second moment and variance of the compressed gradients can be bounded a.s.:*

$$\mathbb{E}\left[\|\widetilde{g}_k\|^2 \;\Big|\; \widehat{w}_{k-1}\right] \leq \left(1 + \frac{\omega_{\mathrm{up}}}{N}\right)\|\nabla F(\widehat{w}_{k-1})\|^2 + \frac{\sigma^2(1 + \omega_{\mathrm{up}})}{Nb},$$

$$\mathbb{E}\left[\|\widetilde{g}_k - \nabla F(\widehat{w}_{k-1})\|^2 \;\Big|\; \widehat{w}_{k-1}\right] \leq \frac{\omega_{\mathrm{up}}}{N}\|\nabla F(\widehat{w}_{k-1})\|^2 + \frac{\sigma^2(1 + \omega_{\mathrm{up}})}{Nb}.$$

*Interpretation:*

- *If $\omega_{\mathrm{up}} = 0$ (i.e. no up compression), the variance corresponds to a mini-batch.*

- *If $\sigma = 0$ and $N = 1$ (i.e. full batch descent with a single device), it becomes:*
  $\mathbb{E}\left[\|\mathcal{C}(\nabla F(w_{k-1})) - \nabla F(w_{k-1})\|^2\right] \leq \omega_{\mathrm{up}}\|\nabla F(w_{k-1})\|^2$ *which is consistent with Assumption 1.*

*Proof.* Let $k$ in $\mathbb{N}^*$, then $\mathbb{E}\left[\|\widetilde{g}_k\|^2 \;\Big|\; \widehat{w}_{k-1}\right] = \|\nabla F(\widehat{w}_{k-1})\|^2 + \mathbb{E}\left[\|\widetilde{g}_k - \nabla F(\widehat{w}_{k-1})\|^2 \;\Big|\; \widehat{w}_{k-1}\right]$.

Secondly:

$$\mathbb{E}\left[\|\widetilde{g}_k - \nabla F(\widehat{w}_{k-1})\|^2 \;\Big|\; \widehat{w}_{k-1}\right]$$
$$= \mathbb{E}\left[\left\|\frac{1}{N}\sum_{i=1}^{N}\left(\widehat{g}_k^i(\widehat{w}_{k-1}) - \nabla F(\widehat{w}_{k-1})\right)\right\|^2 \;\Bigg|\; \widehat{w}_{k-1}\right]$$
$$= \mathbb{E}\left[\left\|\frac{1}{N}\sum_{i=1}^{N}\left(\widehat{g}_k^i(\widehat{w}_{k-1}) - g_k^i(\widehat{w}_{k-1}) + g_k^i(\widehat{w}_{k-1}) - \nabla F(\widehat{w}_{k-1})\right)\right\|^2 \;\Bigg|\; \widehat{w}_{k-1}\right]$$
$$= \mathbb{E}\left[\left\|\frac{1}{N}\sum_{i=1}^{N}\left(\widehat{g}_k^i(\widehat{w}_{k-1}) - g_k^i(\widehat{w}_{k-1})\right)\right\|^2 \;\Bigg|\; \widehat{w}_{k-1}\right]$$
$$+ \mathbb{E}\left[\left\|\frac{1}{N}\sum_{i=1}^{N}\left(g_k^i(\widehat{w}_{k-1}) - \nabla F(\widehat{w}_{k-1})\right)\right\|^2 \;\Bigg|\; \widehat{w}_{k-1}\right],$$

---

[3]`Ghost` is defined in Appendix D.1.

the inner product being null.

Next expanding the squared norm again, and because the two sums of inner products are null as the stochastic oracle and uplink compressions are independent:

$$\mathbb{E}\left[\|\widetilde{g}_k - \nabla F(\widehat{w}_{k-1})\|^2 \,\Big|\, \widehat{w}_{k-1}\right] = \frac{1}{N^2}\sum_{i=1}^{N}\mathbb{E}\left[\left\|\widehat{g}_k^i(\widehat{w}_{k-1}) - g_k^i(\widehat{w}_{k-1})\right\|^2 \,\Big|\, \widehat{w}_{k-1}\right]$$

$$+ \frac{1}{N^2}\sum_{i=1}^{N}\mathbb{E}\left[\left\|g_k^i(\widehat{w}_{k-1}) - \nabla F(\widehat{w}_{k-1})\right\|^2 \,\Big|\, \widehat{w}_{k-1}\right].$$

Then, for any $i$ in $[\![1, N]\!]$ as $\mathbb{E}\left[\left\|\widehat{g}_k^i(\widehat{w}_{k-1}) - g_k^i(\widehat{w}_{k-1})\right\|^2 \,\Big|\, \widehat{w}_{k-1}\right] = \mathbb{E}\left[\mathbb{E}\left[\left\|\widehat{g}_k^i(\widehat{w}_{k-1}) - g_k^i(\widehat{w}_{k-1})\right\|^2 \,\Big|\, g_k^i\right] \,\Big|\, \widehat{w}_{k-1}\right]$, and using Assumption 1 we have:

$$\mathbb{E}\left[\|\widetilde{g}_k - \nabla F(\widehat{w}_{k-1})\|^2 \,\Big|\, \widehat{w}_{k-1}\right] = \frac{\omega_{\mathrm{up}}}{N^2}\sum_{i=1}^{N}\mathbb{E}\left[\left\|g_k^i(\widehat{w}_{k-1})\right\|^2 \,\Big|\, \widehat{w}_{k-1}\right]$$

$$+ \frac{1}{N^2}\sum_{i=1}^{N}\mathbb{E}\left[\left\|g_k^i(\widehat{w}_{k-1}) - \nabla F(\widehat{w}_{k-1})\right\|^2 \,\Big|\, \widehat{w}_{k-1}\right].$$

Furthermore $\mathbb{E}\left[\left\|g_k^i(\widehat{w}_{k-1})\right\|^2 \,\Big|\, \widehat{w}_{k-1}\right] = \mathbb{E}\left[\left\|g_k^i(\widehat{w}_{k-1}) - \nabla F(\widehat{w}_{k-1})\right\|^2 \,\Big|\, \widehat{w}_{k-1}\right] + \|\nabla F(\widehat{w}_{k-1})\|^2$, and using Assumption 4:

$$\mathbb{E}\left[\|\widetilde{g}_k - \nabla F(\widehat{w}_{k-1})\|^2 \,\Big|\, \widehat{w}_{k-1}\right] = \frac{\omega_{\mathrm{up}}}{N}\|\nabla F(\widehat{w}_{k-1})\|^2 + \frac{\sigma^2(1+\omega_{\mathrm{up}})}{Nb},$$

from which we derive the two inequalities of the lemma.

$\square$

## D  The Ghost algorithm

### D.1  Motivation, definition of Ghost and proof sketch

In this section, to convey the best understanding of the theorems and the spirit of the proof, we define a *ghost* algorithm (that is impossible to implement in practice). Ghost is introduced only to get some intuition of the theoretical insight.

**Definition 1** (Ghost algorithm). *The Ghost algorithm is defined as follows, for $k \in \mathbb{N}$, for all $i \in [\![1, N]\!]$ we have:*

$$w_{k+1} = w_k - \gamma\frac{1}{N}\sum_{i=1}^{N}\widehat{g}_{k+1}^i(\widehat{w}_k) \quad and \quad \widehat{w}_{k+1} = w_k - \gamma\mathcal{C}_{\mathrm{dwn}}\left(\frac{1}{N}\sum_{i=1}^{N}\widehat{g}_{k+1}^i(\widehat{w}_k)\right). \quad \text{(S10)}$$

*While the global model is unchanged (1st line), the local model $\widehat{w}_{k+1}$ (2nd line) is updated using the global model $w_k$ at the previous step, which is not available locally.*

In the following, we give the main results for Ghost and complete them with a sketch of proof. Demonstrations are all in the next subsection.

The following Proposition, provides the control of the variance of the local model for Ghost.

**Proposition S2.** *Consider the Ghost update in eq. (S10), under Assumptions 1, 2 and 4, for all $k$ in $\mathbb{N}$ with the convention $\nabla F(w_{-1}) = 0$:*

$$\mathbb{E}\left[\|w_k - \widehat{w}_k\|^2 \,\Big|\, \widehat{w}_{k-1}\right] \le \gamma^2\omega_{\mathrm{dwn}}\left(1 + \frac{\omega_{\mathrm{up}}}{N}\right)\|\nabla F(\widehat{w}_{k-1})\|^2 + \frac{\gamma^2\omega_{\mathrm{dwn}}(1+\omega_{\mathrm{up}})\sigma^2}{Nb}.$$

*Proof.* The proof of Proposition S2 is straightforward using Definition 1. Let $k$ in $\mathbb{N}$, by Definition 1 we have:

$$\|w_k - \widehat{w}_k\|^2 = \left\| \left( w_{k-1} - \gamma \mathcal{C}_{\text{dwn}} \left( \frac{1}{N} \sum_{i=1}^{N} \widehat{g}_k^i(\widehat{w}_{k-1}) \right) \right) - \left( w_{k-1} - \gamma \frac{1}{N} \sum_{i=1}^{N} \widehat{g}_k^i(\widehat{w}_{k-1}) \right) \right\|^2$$

$$= \gamma^2 \left\| \mathcal{C}_{\text{dwn}} \left( \frac{1}{N} \sum_{i=1}^{N} \widehat{g}_k^i(\widehat{w}_{k-1}) \right) - \frac{1}{N} \sum_{i=1}^{N} \widehat{g}_k^i(\widehat{w}_{k-1}) \right\|^2 .$$

Taking expectation w.r.t. down compression, as $\frac{1}{N} \sum_{i=1}^{N} \widehat{g}_k^i(\widehat{w}_{k-1})$ is $w_k$-measurable:

$$\mathbb{E}\left[ \|w_k - \widehat{w}_k\|^2 \mid w_k \right] = \gamma^2 \omega_{\text{dwn}} \mathbb{E}\left[ \left\| \frac{1}{N} \sum_{i=1}^{N} \widehat{g}_k^i(\widehat{w}_{k-1}) \right\|^2 \mid w_k \right] = \gamma^2 \omega_{\text{dwn}} \|\widetilde{g}_k\|^2 ,$$

and Lemma S3 gives the upper bound $\mathbb{E}\left[ \|\widetilde{g}_k\|^2 \mid \widehat{w}_{k-1} \right]$. $\qquad\square$

The takeaway from this Proposition is that we are able to bound the variance of the local model by an affine function of the squared norm of the *previous* stochastic gradients $\nabla F(\widehat{w}_{k-1})$. For `Ghost` only the previous gradient is involved, while for `MCM`, we obtain an additional recursive process.

To obtain the convergence, we then follow the classical approach [31], expanding $\mathbb{E} \|w_k - w_*\|^2$ as $\mathbb{E} \|w_{k-1} - w_*\|^2 - 2\gamma \mathbb{E} \langle \nabla F(\widehat{w}_{k-1}), w_{k-1} - w_* \rangle + \gamma^2 \mathbb{E}\left[ \|\widehat{g}_k(\widehat{w}_{k-1})\|^2 \right]$. The critical aspect is that the inner product does not directly result in a contraction, as the support point of the gradient differs from $w_{k-1}$. Using the fact that $\mathbb{E}[\widehat{w}_{k-1} \mid w_{k-1}] = w_{k-1}$, we further decompose it as

$$-2\gamma \mathbb{E} \langle \nabla F(\widehat{w}_{k-1}), \widehat{w}_{k-1} - w_* \rangle + 2\gamma \mathbb{E} \langle \nabla F(\widehat{w}_{k-1}) - \nabla F(w_{k-1}), w_{k-1} - \widehat{w}_{k-1} \rangle . \quad \text{(S11)}$$

The first part of eq. (S11), corresponds to a "strong contraction": by (strong-)convexity, we can upper bound it by $-2\gamma(\mu \|\widehat{w}_{k-1} - w_*\|^2 + F(\widehat{w}_{k-1}) - F_*)$, which is on average larger than $-2\gamma(\mu \|w_{k-1} - w_*\|^2 + F(w_{k-1}) - F_*)$ (Jensen's inequality). Moreover, as the function is smooth and convex, it can also be upper bounded by $-2\gamma \|\nabla F(\widehat{w}_{k-1})\|^2 / L$. This is a crucial term: we "gain" something of the order of a squared norm of the gradient at $\widehat{w}_{k-1}$, which will *in fine* compensate the variance of the local model. The second part of eq. (S11), corresponds to a positive residual term, proportional to the variance of the compressed model, that can be controlled thanks to Proposition S2 (at $w_{k-1}$!). Putting things together, we get, in the convex case ($\mu = 0$):

**Theorem S6** (Contraction for `Ghost`, convex case). *Under Assumptions 1 to 4, with $\mu = 0$, if $\gamma L(1 + \omega_{\text{up}}/N) \leq \frac{1}{2}$.*

$$\mathbb{E}\|w_k - w_*\|^2 \leq \mathbb{E}\|w_{k-1} - w_*\|^2 - \gamma \mathbb{E}(F(w_{k-1}) - F_*) - \frac{\gamma}{2L} \mathbb{E}\left[ \|\nabla F(\widehat{w}_{k-1})\|^2 \right]$$

$$+ 2\gamma^3 \omega_{\text{dwn}} L \left( 1 + \frac{\omega_{\text{up}}}{N} \right) \mathbb{E} \|\nabla F(\widehat{w}_{k-2})\|^2 + \gamma^2 \frac{(1 + \omega_{\text{up}})\sigma^2}{Nb} (1 + 2\gamma L \omega_{\text{dwn}}) .$$

We can make the following observations:

1. At step $k$, the residual can be upper bounded by a constant times squared norm of the gradient at point $\widehat{w}_{k-2}$. When using recursively this upper bound, if $2\gamma^3 \omega_{\text{dwn}} L(1 + \omega_{\text{up}}/N) \leq \gamma/(2L)$, then these terms cancel out. This is equivalent to $2\gamma L \sqrt{\omega_{\text{dwn}}(1 + \omega_{\text{up}}/N)} \leq 1$. It is natural to chose $\gamma \leq 1/(2L \max(1 + \omega_{\text{up}}/N, 1 + \omega_{\text{dwn}}))$.
2. The bound is in fact proved conditionally to $w_{k-1}$, recursive conditioning is required to propagate the inequality. We carefully handle conditioning in the proofs.

### D.2 Convergence of `Ghost`, complete proof

In this subsection, we provide the complete proof of convergence for `Ghost`. Thus in the following demonstration, we give the key concepts required to later prove the convergence of `MCM`.

> **Theorem S7** (Convergence of `Ghost`, convex case). *Under Assumptions 1 to 4 with $\mu = 0$ (convex case), for all $k$ in $\mathbb{N}$, defining $V_k := \mathbb{E}\left[w_k - w_*\right] + \frac{\gamma}{2L}\mathbb{E}\left[\|\nabla F(\widehat{w}_{k-1})\|^2\right] + 2\gamma L \mathbb{E}\left[\|\widehat{w}_k - w_k\|^2\right]$, we have:*
>
> $$V_k \leq V_{k-1} - \gamma\mathbb{E}\left[F(w_{k-1}) - F(w_*)\right] + \frac{\gamma^2\sigma^2\Phi^{\mathcal{G}}(\gamma)}{Nb},$$
>
> *with $\Phi^{\mathcal{G}}(\gamma) := (1 + \omega_{\text{up}})(1 + 2\gamma L\omega_{\text{dwn}})$.*

**Remark 6.** *This result is similar to eq. (6) but with a different function $\Phi^{\mathcal{G}}$ that has a weaker dependency on $\omega_{\text{dwn}}$.*

*Proof.* Let $k$ in $\mathbb{N}^*$, by definition:

$$\|w_k - w_*\|^2 \leq \|w_{k-1} - w_*\|^2 - 2\gamma\left\langle \widetilde{g}_k,\ w_{k-1} - w_*\right\rangle + \gamma^2\|\widetilde{g}_k\|^2 .$$

Next, we expend the inner product as following:

$$\|w_k - w_*\|^2 \leq \|w_{k-1} - w_*\|^2 - 2\gamma\left\langle \widetilde{g}_k,\ \widehat{w}_{k-1} - w_*\right\rangle - 2\gamma\left\langle \widetilde{g}_k,\ w_{k-1} - \widehat{w}_{k-1}\right\rangle + \gamma^2\|\widetilde{g}_k\|^2 .$$

Taking expectation conditionally to $w_{k-1}$, and using $\mathbb{E}\left[\widetilde{g}_k \mid w_{k-1}\right] = \mathbb{E}\left[\mathbb{E}\left[\widetilde{g}_k \mid \widehat{w}_{k-1}\right] \mid w_{k-1}\right] = \mathbb{E}\left[\nabla F(\widehat{w}_{k-1}) \mid w_{k-1}\right]$, we obtain:

$$\mathbb{E}\left[\|w_k - w_*\|^2 \;\Big|\; w_{k-1}\right] \leq \|w_{k-1} - w_*\|^2 - \mathbb{E}\left[2\gamma\left\langle \nabla F(\widehat{w}_{k-1}),\ \widehat{w}_{k-1} - w_*\right\rangle \mid w_{k-1}\right]$$
$$- 2\gamma\mathbb{E}\left[\left\langle \nabla F(\widehat{w}_{k-1}),\ w_{k-1} - \widehat{w}_{k-1}\right\rangle \mid w_{k-1}\right]$$
$$+ \gamma^2\mathbb{E}\left[\|\widetilde{g}_k\|^2 \;\Big|\; w_{k-1}\right] .$$

Then invoking Lemma S3 to upper bound the squared norm of the stochastic gradients, and noticing that $\mathbb{E}\left[\left\langle \nabla F(w_{k-1}),\ \widehat{w}_{k-1} - w_{k-1}\right\rangle \mid w_{k-1}\right] = 0$ leads to:

$$\mathbb{E}\left[\|w_k - w_*\|^2 \;\Big|\; w_{k-1}\right] \leq \|w_{k-1} - w_*\|^2 - 2\gamma\mathbb{E}\left[\left\langle \nabla F(\widehat{w}_{k-1}),\ \widehat{w}_{k-1} - w_*\right\rangle \mid w_{k-1}\right]$$
$$- 2\gamma\mathbb{E}\left[\left\langle \nabla F(\widehat{w}_{k-1}) - \nabla F(w_{k-1}),\ w_{k-1} - \widehat{w}_{k-1}\right\rangle \mid w_{k-1}\right]$$
$$\text{(S12)}$$
$$+ \gamma^2\left(\left(1 + \frac{\omega_{\text{up}}}{Nb}\right)\mathbb{E}\left[\|\nabla F(\widehat{w}_{k-1})\|^2 \;\Big|\; w_{k-1}\right] + \frac{\sigma^2\left(1 + \omega_{\text{up}}\right)}{Nb}\right) .$$

In the upper inequality:

1. the term $\mathbb{E}\left[\left\langle \nabla F(\widehat{w}_{k-1}),\ \widehat{w}_{k-1} - w_*\right\rangle \mid w_{k-1}\right]$ allows the "strong contraction"

2. the terms $\mathbb{E}\left[\left\langle \nabla F(\widehat{w}_{k-1}) - \nabla F(w_{k-1}),\ w_{k-1} - \widehat{w}_{k-1}\right\rangle \mid w_{k-1}\right]$ and $\mathbb{E}\left[\|\nabla F(\widehat{w}_{k-1})\|^2 \;\Big|\; w_{k-1}\right]$ are two positives terms that we treat as residuals.

3. the last term $\sigma^2\left(1 + \omega_{\text{up}}\right)/(Nb)$ is due to the stochastic noise.

Now using Cauchy-Schwarz inequality (eq. (S5)) and smoothness:

$$- \mathbb{E}\left[2\gamma\left\langle \nabla F(\widehat{w}_{k-1}) - \nabla F(w_{k-1}),\ w_{k-1} - \widehat{w}_{k-1}\right\rangle \mid w_{k-1}\right]$$
$$= 2\gamma\mathbb{E}\left[\left\langle \nabla F(\widehat{w}_{k-1}) - \nabla F(w_{k-1}),\ \widehat{w}_{k-1} - w_{k-1}\right\rangle \mid w_{k-1}\right]$$
$$\leq 2\gamma L\mathbb{E}\left[\|\widehat{w}_{k-1} - w_{k-1}\|^2 \;\Big|\; w_{k-1}\right] ,$$

and thus:

$$\mathbb{E}\left[\|w_k - w_*\|^2 \;\Big|\; w_{k-1}\right] \leq \|w_{k-1} - w_*\|^2 - 2\gamma\mathbb{E}\left[\left\langle \nabla F(\widehat{w}_{k-1}),\ \widehat{w}_{k-1} - w_*\right\rangle \mid w_{k-1}\right]$$
$$+ 2\gamma L\mathbb{E}\left[\|\widehat{w}_{k-1} - w_{k-1}\|^2 \;\Big|\; w_{k-1}\right]$$
$$+ \gamma^2\left(1 + \frac{\omega_{\text{up}}}{N}\right)\mathbb{E}\left[\|\nabla F(\widehat{w}_{k-1})\|^2 \;\Big|\; w_{k-1}\right] + \frac{\gamma^2\sigma^2(1 + \omega_{\text{up}})}{Nb} .$$
$$\text{(S13)}$$

Now, using convexity with Proposition S1:

$$\mathbb{E}\left[\|w_k - w_*\|^2 \;\Big|\; w_{k-1}\right] \leq \|w_{k-1} - w_*\|^2$$

$$- \gamma \mathbb{E}\left[\left(F(\widehat{w}_{k-1}) - F(w_*) + \frac{1}{L}\|\nabla F(\widehat{w}_{k-1})\|^2\right) \;\Big|\; w_{k-1}\right]$$

$$+ 2\gamma L \mathbb{E}\left[\|\widehat{w}_{k-1} - w_{k-1}\|^2 \;\Big|\; w_{k-1}\right]$$

$$+ \gamma^2\left(1 + \frac{\omega_{\mathrm{up}}}{N}\right)\mathbb{E}\left[\|\nabla F(\widehat{w}_{k-1})\|^2 \;\Big|\; w_{k-1}\right] + \frac{\gamma^2\sigma^2(1+\omega_{\mathrm{up}})}{Nb}.$$

Taking the full expectation (without conditioning over any random vectors), and because invoking Jensen inequality (S7) leads to $\mathbb{E}\left[F(\widehat{w}_{k-1})\right] \geq \mathbb{E}\left[F(w_{k-1})\right]$, we finally obtain this intermediate result:

$$\mathbb{E}\left[\|w_k - w_*\|^2\right] \leq \mathbb{E}\left[\|w_{k-1} - w_*\|^2\right] - \gamma\left(\mathbb{E}\left[F(w_{k-1})\right] - F(w_*)\right)$$

$$- \frac{\gamma}{2L}\mathbb{E}\left[\|\nabla F(\widehat{w}_{k-1})\|^2\right] \tag{S14}$$

$$+ 2\gamma L\mathbb{E}\left[\|\widehat{w}_{k-1} - w_{k-1}\|^2\right] + \frac{\gamma^2\sigma^2(1+\omega_{\mathrm{up}})}{Nb},$$

where we considered that $\gamma L(1 + \omega_{\mathrm{up}}/N) \leq 1/2$, which implies that $\gamma\left(1 - \gamma L\left(1 + \frac{\omega_{\mathrm{up}}}{N}\right)\right) \geq \frac{\gamma}{2}$.

Remark that eq. (S14) is valid for both `Ghost` and `MCM`, and that the proof of `MCM` will follow the same initial line.

With Proposition S2:

$$\mathbb{E}\left[\|w_k - \widehat{w}_k\|^2 \;\Big|\; \widehat{w}_{k-1}\right] \leq \gamma^2\omega_{\mathrm{dwn}}\left(1 + \frac{\omega_{\mathrm{up}}}{N}\right)\|\nabla F(\widehat{w}_{k-1})\|^2 + \frac{\gamma^2\omega_{\mathrm{dwn}}(1+\omega_{\mathrm{up}})\sigma^2}{Nb}. \tag{S15}$$

Defining $V_k := \mathbb{E}\left[w_k - w_*\right] + \frac{\gamma}{2L}\mathbb{E}\left[\|\nabla F(\widehat{w}_{k-1})\|^2\right] + C\mathbb{E}\left[\|\widehat{w}_k - w_k\|^2\right]$ with $C = 2\gamma L$, and combining this two equations as following $(S14) + C(S15)$ leads to:

$$\mathbb{E}\left[\|w_k - w_*\|^2\right] + C\mathbb{E}\left[\|\widehat{w}_{k-1} - w_{k-1}\|^2\right] + \frac{\gamma}{2L}\mathbb{E}\left[\|\nabla F(\widehat{w}_{k-1})\|^2\right]$$

$$\leq \mathbb{E}\left[\|w_{k-1} - w_*\|^2\right] - \gamma\left(\mathbb{E}\left[F(w_{k-1})\right] - F(w_*)\right)$$

$$+ 2\gamma L\mathbb{E}\left[\|\widehat{w}_{k-1} - w_{k-1}\|^2\right] + \frac{\gamma^2\sigma^2(1+\omega_{\mathrm{up}})}{Nb}$$

$$+ 2\gamma L \times \gamma^2\omega_{\mathrm{dwn}}\left(1 + \frac{\omega_{\mathrm{up}}}{N}\right)\|\nabla F(\widehat{w}_{k-1})\|^2 + 2\gamma L \times \frac{\gamma^2\omega_{\mathrm{dwn}}(1+\omega_{\mathrm{up}})\sigma^2}{Nb}.$$

To ensure a contraction of the Lyapunov function we require:

$$\gamma^2\omega_{\mathrm{dwn}}\left(1 + \frac{\omega_{\mathrm{up}}}{N}\right) \leq \frac{\gamma}{2L} \iff \gamma L \leq \frac{1}{2\sqrt{\omega_{\mathrm{dwn}}\left(1 + \frac{\omega_{\mathrm{up}}}{N}\right)}}$$

Under this condition, we obtain:

$$V_k \leq V_{k-1} - \gamma\mathbb{E}\left[F(w_{k-1}) - F(w_*)\right] + \frac{\gamma^2\sigma^2\Phi^{\mathcal{G}}(\gamma)}{Nb},$$

with $\Phi^{\mathcal{G}}(\gamma) := (1 + \omega_{\mathrm{up}})(1 + 2\gamma L\omega_{\mathrm{dwn}})$.

By recurrence and for $k = K$:

$$V_K \leq V_0 - \sum_{k=1}^{K}\gamma\mathbb{E}\left[F(w_{k-1}) - F(w_*)\right] + \sum_{k=1}^{K}\frac{\gamma^2\sigma^2\Phi^{\mathcal{G}}(\gamma)}{Nb},$$

which leads to:

$$\frac{1}{K}\sum_{k=1}^{K}\mathbb{E}\left[F(w_{k-1})-F(w_*)\right]\leq\frac{V_0-V_k}{\gamma K}+\frac{\gamma\sigma^2\Phi^{\mathcal{G}}(\gamma)}{Nb}\,.$$

Finally, for any $K$ in $\mathbb{N}^*$, with $\gamma L\leq\min\left\{\dfrac{1}{2\left(1+\dfrac{\omega_{\mathrm{up}}}{N}\right)},\dfrac{1}{2\sqrt{\omega_{\mathrm{dwn}}\left(1+\dfrac{\omega_{\mathrm{up}}}{N}\right)}}\right\}$ we have:

$$\frac{\gamma}{K}\sum_{t=1}^{K}\mathbb{E}\left[F(w_t)-F(w_*)\right]\leq\frac{\|w_0-w_*\|^2}{K}+\frac{\gamma\sigma^2\Phi^{\mathcal{G}}(\gamma)}{Nb}\,.$$

Note that the bound of $\gamma L$ encompass the case $\omega_{\mathrm{dwn}}=0$ (i.e. no downlink compression), but in the general case of bidirectional compression, we nearly always have $\omega_{\mathrm{dwn}}>1$, and thus the dominant term is in fact $\dfrac{1}{2\sqrt{\omega_{\mathrm{dwn}}\left(1+\dfrac{\omega_{\mathrm{up}}}{N}\right)}}$.

And by Jensen, it implies that:

$$\mathbb{E}\left[F(\bar{w}_K)-F(w_*)\right]\leq\frac{\|w_0-w_*\|^2}{\gamma K}+\frac{\gamma\sigma^2\Phi(\gamma)}{Nb}\quad\text{with }\Phi^{\mathcal{G}}(\gamma):=(1+\omega_{\mathrm{up}})(1+2\omega_{\mathrm{dwn}}\gamma L)\,.$$

$\square$

# E   Proofs for `MCM` (and `Rand-MCM`)

In this section, we provide the proofs for `MCM` in the convex, strongly-convex, and non-convex cases in respectively Theorems S9 to S11. The proofs for `Rand-MCM` (see Theorem 4) are identical and only require to adapt notations as explained in appendix E.5.

We denote for $\gamma$ in $\mathbb{R}$, $\Phi(\gamma):=(1+\omega_{\mathrm{up}})\left(1+\frac{8\gamma L\omega_{\mathrm{dwn}}}{\alpha_{\mathrm{dwn}}}\right)$, for $k$ in $\mathbb{N}$, $\Upsilon_k=\|w_k-H_{k-1}\|^2$ and we define $\gamma_{\max}$ such that:

$$\gamma_{\max}L\leq\min\left\{\frac{1}{8\omega_{\mathrm{dwn}}},\frac{1}{2\left(1+\frac{\omega_{\mathrm{up}}}{N}\right)},\frac{1}{4\sqrt{\frac{\omega_{\mathrm{dwn}}}{\alpha_{\mathrm{dwn}}}\left(\frac{1}{\alpha_{\mathrm{dwn}}}+\frac{\omega_{\mathrm{up}}}{N}\right)}}\right\}\,.$$

Note that this is equivalent to notations given in Section 3 if we take $\alpha_{\mathrm{dwn}}=1/8\omega_{\mathrm{dwn}}$.

## E.1   Control of the Variance of the local model for `MCM` (Theorem 3)

In this section, we provide a control of the variance of the local model for `MCM`, as done previously in Proposition S2 for `Ghost`: this corresponds to Theorem 3. The demonstration is more complex than for `Ghost` and it highlights the trade-offs for the learning rate $\alpha_{\mathrm{dwn}}$. The demonstration builds a bias-variance decomposition of $\|\Omega_k\|^2=\|w_k-H_k\|^2$. The variance is then decomposed in three terms, as a result we will need to compute four terms:

$$\|w_k-H_{k-1}\|^2=\text{Bias}^2+2\gamma^2(\text{Var}_{11}+\text{Var}_{12})+2\alpha_{\mathrm{dwn}}^2\text{Var}_2\,.\tag{S16}$$

---

**Theorem S8.** *Consider the `MCM` update as in eq. (2). Under Assumptions 1, 2 and 4 with $\mu=0$, if $\gamma\leq(8\omega_{\mathrm{dwn}}L)^{-1}$ and $\alpha_{\mathrm{dwn}}\leq(8\omega_{\mathrm{dwn}})^{-1}$, then for all $k$ in $\mathbb{N}$:*

$$\mathbb{E}\left[\Upsilon_k\right]\leq\left(1-\frac{\alpha_{\mathrm{dwn}}}{2}\right)\mathbb{E}\left[\Upsilon_{k-1}\right]+2\gamma^2\left(\frac{1}{\alpha_{\mathrm{dwn}}}+\frac{\omega_{\mathrm{up}}}{N}\right)\mathbb{E}\left[\|\nabla F(\widehat{w}_{k-1})\|^2\right]+\frac{2\gamma^2\sigma^2(1+\omega_{\mathrm{up}})}{Nb}\,.$$

---

*Proof.* Let $k$ in $\mathbb{N}$, we recall that by definition:

$$\begin{cases} \Omega_k = w_k - H_{k-1} \\ \widehat{\Omega}_k = \mathcal{C}_{\mathrm{dwn}}(\Omega_k) \\ \widehat{w}_k = \widehat{\Omega}_k + H_{k-1} \, . \end{cases}$$

We start the proof by introducing $\|\Omega_k\|^2$:

$$\mathbb{E}\left[ \|w_k - \widehat{w}_k\|^2 \ \middle| \ w_k \right] = \mathbb{E}\left[ \left\|\widehat{\Omega}_k - \Omega_k\right\|^2 \ \middle| \ w_k \right] \leq \omega_{\mathrm{dwn}} \|\Omega_k\|^2 \, .$$

Next, we perform a bias-variance decomposition:

$$\begin{aligned} \|\Omega_k\|^2 = \|w_k - H_{k-1}\|^2 = & \, \|w_k - H_{k-1} - \mathbb{E}\left[w_k - H_{k-1} \mid w_{k-1}\right]\|^2 \\ & + \|\mathbb{E}\left[w_k - H_{k-1} \mid w_{k-1}\right]\|^2 \\ & + 2\left\langle \, w_k - H_{k-1} - \mathbb{E}\left[w_k - H_{k-1} \mid w_{k-1}\right], \, \mathbb{E}\left[w_k - H_{k-1} \mid w_{k-1}\right] \, \right\rangle, \end{aligned}$$

taking expectation w.r.t. $w_{k-1}$:

$$\mathbb{E}\left[\Upsilon_k \mid w_{k-1}\right] = \underbrace{\mathbb{E}\left[ \|w_k - H_{k-1} - \mathbb{E}\left[w_k - H_{k-1} \mid w_{k-1}\right]\|^2 \ \middle| \ w_{k-1} \right]}_{\mathrm{Var}}$$
$$+ \underbrace{\|\mathbb{E}\left[w_k - H_{k-1} \mid w_{k-1}\right]\|^2}_{\mathrm{Bias}^2} \, .$$

The first term is the variance Var, and the second term corresponds to the squared bias $\mathrm{Bias}^2$.

Let's handle first the variance, by definition:

$$\begin{aligned} \mathrm{Var} = & \, \mathbb{E}\left[ \|w_k - H_{k-1} - \mathbb{E}\left[w_k - H_{k-1} \mid w_{k-1}\right]\|^2 \ \middle| \ w_{k-1} \right] \\ = & \, \mathbb{E}\left[ \|w_{k-1} - \gamma \widetilde{g}_k - H_{k-2} - \alpha_{\mathrm{dwn}} \mathcal{C}(w_{k-1} - H_{k-2}) \right. \\ & \left. - w_{k-1} - \gamma \mathbb{E}\left[\widetilde{g}_k \mid w_{k-1}\right] - H_{k-2} - \alpha_{\mathrm{dwn}} \mathbb{E}\left[\mathcal{C}(w_{k-1} - H_{k-2} \mid w_{k-1})\right]\|^2 \middle| w_{k-1} \right] \, . \end{aligned}$$

After simplification and using eq. :

$$\begin{aligned} \mathrm{Var} = & \, \mathbb{E}\left[ \| - \gamma \left(\widetilde{g}_k + \mathbb{E}\left[\nabla F(\widehat{w}_{k-1}) \mid w_{k-1}\right]\right) + \alpha_{\mathrm{dwn}}\left(\mathcal{C}(w_{k-1} - H_{k-2})\right) \right. \\ & \left. - (w_{k-1} - H_{k-2})\|^2 \middle| w_{k-1} \right] \\ \leq & \, 2\gamma^2 \mathbb{E}\left[ \|\widetilde{g}_k - \mathbb{E}\left[\nabla F(\widehat{w}_{k-1}) \mid w_{k-1}\right]\|^2 \ \middle| \ w_{k-1} \right] \\ & + 2\alpha_{\mathrm{dwn}}^2 \mathbb{E}\left[ \|\mathcal{C}(w_{k-1} - H_{k-2}) - (w_{k-1} - H_{k-2})\|^2 \ \middle| \ w_{k-1} \right] \\ \leq & \, 2\gamma^2 \underbrace{\mathbb{E}\left[ \|\widetilde{g}_k - \mathbb{E}\left[\nabla F(\widehat{w}_{k-1}) \mid w_{k-1}\right]\|^2 \ \middle| \ w_{k-1} \right]}_{\mathrm{Var}_1} + 2\alpha_{\mathrm{dwn}}^2 \underbrace{\omega_{\mathrm{dwn}} \|w_{k-1} - H_{k-2}\|^2}_{\mathrm{Var}_2} \\ \leq & \, 2\gamma^2 \mathrm{Var}_1 + 2\alpha_{\mathrm{dwn}}^2 \mathrm{Var}_2 \, . \end{aligned}$$

An interpretation of the above decomposition is that:

- $\mathrm{Var}_1$ is the part of the downlink compression caused by the increment $\widetilde{g}_k$, it is similar to Ghost.
- $\mathrm{Var}_2$ is the impact of the propagation of the previous noise.

We compute the first term by introducing $\nabla F(\widehat{w}_{k-1})$, the second being kept as it is:

$$\begin{aligned} \mathrm{Var}_1 = & \, \mathbb{E}\left[ \|\widetilde{g}_k - \mathbb{E}\left[\nabla F(\widehat{w}_{k-1}) \mid w_{k-1}\right]\|^2 \ \middle| \ w_{k-1} \right] \\ = & \, \mathbb{E}\left[ \|\widetilde{g}_k - \nabla F(\widehat{w}_{k-1}) + \nabla F(\widehat{w}_{k-1}) - \mathbb{E}\left[\nabla F(\widehat{w}_{k-1}) \mid w_{k-1}\right]\|^2 \ \middle| \ w_{k-1} \right] \\ = & \, \underbrace{\mathbb{E}\left[ \|\widetilde{g}_k - \nabla F(\widehat{w}_{k-1})\|^2 \ \middle| \ w_{k-1} \right]}_{\mathrm{Var}_{11}} + \underbrace{\mathbb{E}\left[ \|\nabla F(\widehat{w}_{k-1}) - \mathbb{E}\left[\nabla F(\widehat{w}_{k-1}) \mid w_{k-1}\right]\|^2 \ \middle| \ w_{k-1} \right]}_{\mathrm{Var}_{12}} \\ = & \, \mathrm{Var}_{11} + \mathrm{Var}_{12} \, , \end{aligned}$$

the inner product is null given that $\mathbb{E}\left[\nabla F(\widehat{w}_{k-1}) - \mathbb{E}\left[\nabla F(\widehat{w}_{k-1}) \mid w_{k-1}\right] \mid w_{k-1}\right] = 0$.

Moreover:

$$\mathrm{Var}_{11} = \mathbb{E}\left[\left\|\widetilde{g}_k - \nabla F(\widehat{w}_{k-1})\right\|^2 \;\middle|\; w_{k-1}\right] = \mathbb{E}\left[\mathbb{E}\left[\left\|\widetilde{g}_k - \nabla F(\widehat{w}_{k-1})\right\|^2 \;\middle|\; \widehat{w}_{k-1}\right] \;\middle|\; w_{k-1}\right],$$

so, we can use Lemma S3: $\mathrm{Var}_{11} = \mathbb{E}\left[\dfrac{\sigma^2}{Nb}(1 + \omega_{\mathrm{up}}) + \dfrac{\omega_{\mathrm{up}}}{N}\left\|\nabla F(\widehat{w}_{k-1})\right\|^2 \;\middle|\; w_{k-1}\right].$

And now we use smoothness for the second term:

$$\begin{aligned}
\mathrm{Var}_{12} &= \mathbb{E}\left[\left\|\nabla F(\widehat{w}_{k-1}) - \mathbb{E}\left[\nabla F(\widehat{w}_{k-1}) \mid w_{k-1}\right]\right\|^2 \;\middle|\; w_{k-1}\right] \\
&\leq \mathbb{E}\left[\left\|\nabla F(\widehat{w}_{k-1}) - \nabla F(w_{k-1})\right\|^2 \;\middle|\; w_{k-1}\right] \quad \text{by Lemma S1,} \\
&\leq L^2 \mathbb{E}\left[\left\|\widehat{w}_{k-1} - w_{k-1}\right\|^2 \;\middle|\; w_{k-1}\right] \quad \text{using smoothness,} \\
&\leq L^2 \omega_{\mathrm{dwn}} \Upsilon_{k-1} \quad \text{with Assumption 1.}
\end{aligned}$$

At the end:

$$\begin{aligned}
\mathrm{Var} &\leq 2\gamma^2\left(\frac{\sigma^2(1 + \omega_{\mathrm{up}})}{Nb} + \frac{\omega_{\mathrm{up}}}{N}\mathbb{E}\left[\left\|\nabla F(\widehat{w}_{k-1})\right\|^2 \;\middle|\; w_{k-1}\right] + L^2 \omega_{\mathrm{dwn}}\Upsilon_{k-1}\right) \\
&\quad + 2\alpha_{\mathrm{dwn}}^2 \omega_{\mathrm{dwn}}\Upsilon_{k-1}.
\end{aligned} \tag{S17}$$

Now we focus on the squared bias $\mathrm{Bias}^2$, with Lemma S2:

$$\begin{aligned}
\mathrm{Bias}^2 &= \left\|\mathbb{E}\left[w_k - H_{k-1} \mid w_{k-1}\right]\right\|^2 \\
&= \left\|(1 - \alpha_{\mathrm{dwn}})(w_{k-1} - H_{k-2}) - \gamma\mathbb{E}\left[\nabla F(\widehat{w}_{k-1}) \mid w_{k-1}\right]\right\|^2, \quad \text{and with Equation (S3),} \\
&\leq (1 - \alpha_{\mathrm{dwn}})^2\,(1 + \alpha_{\mathrm{dwn}})\,\Upsilon_{k-1} + \gamma^2(1 + \frac{1}{\alpha_{\mathrm{dwn}}})\left\|\mathbb{E}\left[\nabla F(\widehat{w}_{k-1}) \mid w_{k-1}\right]\right\|^2.
\end{aligned}$$

And because $(1 - \alpha_{\mathrm{dwn}})(1 + \alpha_{\mathrm{dwn}}) < 1$, we finally get that:

$$\mathrm{Bias}^2 \leq (1 - \alpha_{\mathrm{dwn}})\Upsilon_{k-1} + \gamma^2\left(1 + \frac{1}{\alpha_{\mathrm{dwn}}}\right)\left\|\mathbb{E}\left[\nabla F(\widehat{w}_{k-1}) \mid w_{k-1}\right]\right\|^2. \tag{S18}$$

Combining all eqs. (S17) and (S18) into eq. (S16):

$$\begin{aligned}
\mathbb{E}\left[\Upsilon_k \mid w_{k-1}\right] &\leq (1 - \alpha_{\mathrm{dwn}})\Upsilon_{k-1} + \gamma^2\left(1 + \frac{1}{\alpha_{\mathrm{dwn}}}\right)\left\|\mathbb{E}\left[\nabla F(\widehat{w}_{k-1}) \mid w_{k-1}\right]\right\|^2 \\
&\quad + 2\gamma^2\left(\frac{\sigma^2(1 + \omega_{\mathrm{up}})}{Nb} + \frac{\omega_{\mathrm{up}}}{N}\mathbb{E}\left[\left\|\nabla F(\widehat{w}_{k-1})\right\|^2 \;\middle|\; w_{k-1}\right]\right) \\
&\quad + 2\gamma^2\left(L^2 \omega_{\mathrm{dwn}}\Upsilon_{k-1}\right) \\
&\quad + 2\alpha_{\mathrm{dwn}}^2 \omega_{\mathrm{dwn}}\Upsilon_{k-1} \\
&\leq \left(1 - \alpha_{\mathrm{dwn}} + 2\gamma^2 L^2 \omega_{\mathrm{dwn}} + 2\alpha_{\mathrm{dwn}}^2 \omega_{\mathrm{dwn}}\right)\left\|w_{k-1} - H_{k-2}\right\|^2 \\
&\quad + \gamma^2\left(1 + \frac{1}{\alpha_{\mathrm{dwn}}}\right)\left\|\mathbb{E}\left[\nabla F(\widehat{w}_{k-1}) \mid w_{k-1}\right]\right\|^2 \\
&\quad + \frac{2\gamma^2 \omega_{\mathrm{up}}}{N}\mathbb{E}\left[\left\|\nabla F(\widehat{w}_{k-1})\right\|^2 \;\middle|\; w_{k-1}\right] + \frac{2\gamma^2 \sigma^2(1 + \omega_{\mathrm{up}})}{Nb}.
\end{aligned}$$

Next, we require:

$$\begin{cases}
2\alpha_{\mathrm{dwn}}^2 \omega_{\mathrm{dwn}} \leq \frac{1}{4}\alpha_{\mathrm{dwn}} \iff \alpha_{\mathrm{dwn}} \leq \frac{1}{8\omega_{\mathrm{dwn}}}, \\
2\gamma^2 L^2 \omega_{\mathrm{dwn}} \leq \frac{1}{4}\alpha_{\mathrm{dwn}} = \frac{1}{32\omega_{\mathrm{dwn}}}, \text{ by taking } \alpha_{\mathrm{dwn}} = \frac{1}{8\omega_{\mathrm{dwn}}} \iff \gamma \leq \frac{1}{8\omega_{\mathrm{dwn}}L}, \\
1 + \frac{1}{\alpha_{\mathrm{dwn}}} \leq \frac{2}{\alpha_{\mathrm{dwn}}} \text{ which is not restrictive if } \omega_{\mathrm{dwn}} \geq 1.
\end{cases}$$

Thus, it leads to:

$$\mathbb{E}\left[\Upsilon_k \mid w_{k-1}\right] \leq \left(1 - \frac{\alpha_{\mathrm{dwn}}}{2}\right) \Upsilon_{k-1} + \frac{2\gamma^2}{\alpha_{\mathrm{dwn}}} \left\|\mathbb{E}\left[\nabla F(\widehat{w}_{k-1}) \mid w_{k-1}\right]\right\|^2$$
$$+ \frac{2\gamma^2 \omega_{\mathrm{up}}}{N} \mathbb{E}\left[\left\|\nabla F(\widehat{w}_{k-1})\right\|^2 \mid w_{k-1}\right] + \frac{2\gamma^2 \sigma^2 (1 + \omega_{\mathrm{up}})}{Nb} .$$

Next, we bound $\left\|\mathbb{E}\left[\nabla F(\widehat{w}_{k-1}) \mid w_{k-1}\right]\right\|^2$ with $\mathbb{E}\left[\left\|\nabla F(\widehat{w}_{k-1})\right\|^2 \mid w_{k-1}\right]$, and we obtain:

$$\mathbb{E}\left[\Upsilon_k \mid w_{k-1}\right] \leq \left(1 - \frac{\alpha_{\mathrm{dwn}}}{2}\right) \Upsilon_{k-1}$$
$$+ 2\gamma^2 \left(\frac{1}{\alpha_{\mathrm{dwn}}} + \frac{\omega_{\mathrm{up}}}{N}\right) \mathbb{E}\left[\left\|\nabla F(\widehat{w}_{k-1})\right\|^2 \mid w_{k-1}\right]$$
$$+ \frac{2\gamma^2 \sigma^2 (1 + \omega_{\mathrm{up}})}{Nb} .$$

Taking the unconditional expectation gives the result.

$\square$

## E.2 Convex case (Theorem 2)

In this section, we give the demonstration of `MCM` in the convex case (Theorem 2).

---

**Theorem S9** (Convergence of `MCM` in the homogeneous and convex case). *Under Assumptions 1 to 4 with $\mu = 0$, for a learning rate $\alpha_{\mathrm{dwn}} \leq \frac{1}{8\omega_{\mathrm{dwn}}}$, for all $k > 0$, for any $\gamma \leq \gamma_{\max}$, defining $V_k := \mathbb{E}[\|w_k - w_*\|^2] + 32\gamma L \omega_{\mathrm{dwn}}^2 \mathbb{E}[\Upsilon_k]$, for $\bar{w}_k = \frac{1}{k}\sum_{i=0}^{k-1} w_i$, we have:*

$$\gamma \mathbb{E}\left[F(w_{k-1}) - F(w_*)\right] \leq V_{k-1} - V_k + \frac{\gamma^2 \sigma^2 \Phi(\gamma)}{Nb} \implies \mathbb{E}[F(\bar{w}_k) - F_*] \leq \frac{V_0}{\gamma k} + \frac{\gamma \sigma^2 \Phi(\gamma)}{Nb} .$$

*Consequently, for $K$ in $\mathbb{N}$ large enough, a step-size $\gamma = \sqrt{\frac{\|w_0 - w_*\|^2 Nb}{(1 + \omega_{\mathrm{up}})\sigma^2 K}}$ and a learning rate $\alpha_{\mathrm{dwn}} = \frac{1}{8\omega_{\mathrm{dwn}}}$, we have:*

$$\mathbb{E}[F(\bar{w}_K) - F_*] \leq 2\sqrt{\frac{\|w_0 - w_*\|^2 (1 + \omega_{\mathrm{up}})\sigma^2}{NbK}} + O(K^{-1}).$$

*Moreover if $\sigma^2 = 0$ (noiseless case), we recover a faster convergence: $\mathbb{E}[F(\bar{w}_K) - F_*] = O(K^{-1})$.*

---

*Proof.* Let $k$ in $\mathbb{N}^*$, the proof follows the one for `Ghost`, and we start from eq. (S14):

$$\mathbb{E}\left[\|w_k - w_*\|^2\right] \leq \mathbb{E}\left[\|w_{k-1} - w_*\|^2\right] - \gamma\left(\mathbb{E}\left[F(w_{k-1})\right] - F(w_*)\right) - \frac{\gamma}{2L}\mathbb{E}\left[\left\|\nabla F(\widehat{w}_{k-1})\right\|^2\right]$$
$$+ 2\gamma L \mathbb{E}\left[\|\widehat{w}_{k-1} - w_{k-1}\|^2\right] + \frac{\gamma^2 \sigma^2 (1 + \omega_{\mathrm{up}})}{Nb} ,$$

with Assumption 1, it easily becomes:

$$\mathbb{E}\left[\|w_k - w_*\|^2\right] \leq \mathbb{E}\left[\|w_{k-1} - w_*\|^2\right] - \gamma\left(\mathbb{E}\left[F(w_{k-1})\right] - F(w_*)\right) - \frac{\gamma}{2L}\mathbb{E}\left[\left\|\nabla F(\widehat{w}_{k-1})\right\|^2\right]$$
$$+ 2\gamma L \omega_{\mathrm{dwn}} \mathbb{E}\left[\Upsilon_{k-1}\right] + \frac{\gamma^2 \sigma^2 (1 + \omega_{\mathrm{up}})}{Nb} .$$

Theorem 3 which is specific to `MCM` gives:

$$\mathbb{E}\left[\Upsilon_k\right] \leq \left(1 - \frac{\alpha_{\mathrm{dwn}}}{2}\right) \mathbb{E}\left[\Upsilon_{k-1}\right] + 2\gamma^2 \left(\frac{1}{\alpha_{\mathrm{dwn}}} + \frac{\omega_{\mathrm{up}}}{N}\right) \mathbb{E}\left[\left\|\nabla F(\widehat{w}_{k-1})\right\|^2\right] + \frac{2\gamma^2 \sigma^2 (1 + \omega_{\mathrm{up}})}{Nb} .$$

Defining: $V_k := \mathbb{E}\left[\|w_k - w_*\|^2\right] + \gamma LC\mathbb{E}\left[\Upsilon_k\right]$ with $C = \frac{4\omega_{\mathrm{dwn}}}{\alpha_{\mathrm{dwn}}}$, and, combining the two last equations:

$$
\begin{aligned}
\mathbb{E}\left[\|w_k - w_*\|^2\right] + \gamma LC\mathbb{E}\left[\Upsilon_k\right] \leq{}& \mathbb{E}\left[\|w_{k-1} - w_*\|^2\right] - \gamma\mathbb{E}\left[F(w_{k-1}) - F(w_*)\right] \\
& + 2\gamma L\omega_{\mathrm{dwn}}\mathbb{E}\left[\Upsilon_{k-1}\right] \\
& - \frac{\gamma}{2L}\mathbb{E}\left[\|\nabla F(\widehat{w}_{k-1})\|^2\right] + \frac{\gamma^2\sigma^2(1+\omega_{\mathrm{up}})}{Nb} \\
& + \left(1 - \frac{\alpha_{\mathrm{dwn}}}{2}\right)\gamma LC\mathbb{E}\left[\Upsilon_{k-1}\right] \\
& + 2\gamma^3 LC\left(\frac{1}{\alpha_{\mathrm{dwn}}} + \frac{\omega_{\mathrm{up}}}{N}\right)\mathbb{E}\left[\|\nabla F(\widehat{w}_{k-1})\|^2\right] \\
& + \frac{2\gamma^3 L\sigma^2(1+\omega_{\mathrm{up}})C}{N},
\end{aligned}
$$

and reordering the terms gives:

$$
\begin{aligned}
V_k \leq{}& \mathbb{E}\left[\|w_{k-1} - w_*\|^2\right] + \left(2\gamma L\omega_{\mathrm{dwn}} + \left(1 - \frac{\alpha_{\mathrm{dwn}}}{2}\right)\gamma LC\right)\mathbb{E}\left[\|w_{k-1} - H_{k-1}\|^2\right] \\
& + \left(2\gamma^3 LC\left(\frac{1}{\alpha_{\mathrm{dwn}}} + \frac{\omega_{\mathrm{up}}}{N}\right) - \frac{\gamma}{2L}\right)\mathbb{E}\left[\|\nabla F(\widehat{w}_{k-1})\|^2\right] \\
& - \gamma\mathbb{E}\left[F(w_{k-1}) - F(w_*)\right] \\
& + (2\gamma LC + 1)\frac{\gamma^2\sigma^2(1+\omega_{\mathrm{up}})}{Nb}.
\end{aligned}
$$

We observe that:

$$
2\gamma L\omega_{\mathrm{dwn}} + \left(1 - \frac{\alpha_{\mathrm{dwn}}}{2}\right)\gamma LC \leq \gamma LC \iff C \geq \frac{4\omega_{\mathrm{dwn}}}{\alpha_{\mathrm{dwn}}} \quad \text{which is true by definition of } C.
$$

Secondly, to get the contraction requires

$$
\begin{aligned}
2\gamma^3 LC\left(\frac{1}{\alpha_{\mathrm{dwn}}} + \frac{\omega_{\mathrm{up}}}{N}\right) - \frac{\gamma}{2L} \leq 0 &\iff \gamma^2 L \leq \frac{1}{4LC\left(\dfrac{1}{\alpha_{\mathrm{dwn}}} + \dfrac{\omega_{\mathrm{up}}}{N}\right)} \\
&\iff \gamma L \leq \frac{1}{4\sqrt{\dfrac{\omega_{\mathrm{dwn}}}{\alpha_{\mathrm{dwn}}}\left(\dfrac{1}{\alpha_{\mathrm{dwn}}} + \dfrac{\omega_{\mathrm{up}}}{N}\right)}},
\end{aligned}
$$

because $C = 4\omega_{\mathrm{dwn}}/\alpha$. Thus, we have that:

$$
V_k \leq V_{k-1} - \gamma\mathbb{E}\left[F(w_{k-1}) - F(w_*)\right] + \frac{\gamma^2\sigma^2\Phi(\gamma)}{Nb} \quad \text{denoting } \Phi(\gamma) := (1+\omega_{\mathrm{up}})\left(1 + \frac{8\gamma L\omega_{\mathrm{dwn}}}{\alpha_{\mathrm{dwn}}}\right),
$$

and then for $k = K \in \mathbb{N}^*$, by recurrence:

$$
V_K \leq V_0 - \gamma\sum_{k=1}^{K}\mathbb{E}\left[F(w_{k-1}) - F(w_*)\right] + \frac{\gamma^2\sigma^2\Phi(\gamma)}{Nb},
$$

which implies:

$$
\frac{1}{K}\sum_{k=1}^{K}\mathbb{E}\left[F(w_{k-1}) - F(w_*)\right] \leq \frac{V_0 - V_K}{\gamma K} + \frac{\gamma^2\sigma^2\Phi(\gamma)}{Nb},
$$

Finally, by Jensen, for any $K$ in $\mathbb{N}^*$ such that $\gamma L \leq$ $\min\left\{\frac{1}{8\omega_{\text{dwn}}}, \frac{1}{2\left(1 + \frac{\omega_{\text{up}}}{N}\right)}, \frac{1}{4\sqrt{\frac{\omega_{\text{dwn}}}{\alpha_{\text{dwn}}}\left(\frac{1}{\alpha_{\text{dwn}}} + \frac{\omega_{\text{up}}}{N}\right)}}\right\}$, we have:

$$\mathbb{E}\left[F(\bar{w}_K) - F(w_*)\right] \leq \frac{V_0}{\gamma K} + \frac{\gamma\sigma^2\Phi(\gamma)}{Nb},$$

which concludes the proof.

$\square$

### E.3 Strongly-convex case (Theorem 1)

In this section, we give the demonstration for MCM in the strongly-convex case (Theorem 1).

> **Theorem S10** (Convergence of MCM in the homogeneous and strongly-convex case). *Under Assumptions 1 to 4 with $\mu > 0$, for $k$ in $\mathbb{N}$, for a learning rate $\alpha_{\text{dwn}} \leq \frac{1}{8\omega_{\text{dwn}}}$, for any sequence $(\gamma_k)_{k \geq 0} \leq \gamma_{\max}$, defining $V_k := \mathbb{E}[\|w_k - w_*\|^2] + 32\gamma L\omega_{\text{dwn}}^2\mathbb{E}[\Upsilon_k]$, we have:*
>
> $$V_k \leq (1 - \gamma_k\mu)V_{k-1} - \gamma_k\mathbb{E}\left[F(\widehat{w}_{k-1}) - F(w_*)\right] + \frac{\gamma_k^2\sigma^2\Phi(\gamma_k)}{Nb},$$
>
> *Consequently,*
>
> 1. *if $\sigma^2 = 0$ (noiseless case), for $\gamma_k \equiv \gamma_{\max}$ we recover a linear convergence rate:* $\mathbb{E}[\|w_K) - w_*\|^2] \leq (1 - \gamma_{\max}\mu)^k V_0$;
> 2. *if $\sigma^2 > 0$, defining $\widetilde{L}$ such that $\gamma_{\max} = (2\widetilde{L})^{-1}$, taking for all $k$ in $\mathbb{N}$, $\gamma_k = 2/(\mu(k+1) + \widetilde{L})$, for the weighted Polyak-Ruppert average $\bar{w}_K = \sum_{k=1}^K \lambda_k w_{k-1}/\sum_{k=1}^K \lambda_k$, with $\lambda_k := \frac{1}{\gamma_{k-1}^{-1}}$, we have:*
>    $$\mathbb{E}\left[F(\bar{w}_K) - F(w_*)\right] \leq \frac{\mu + 2\widetilde{L}}{4\mu K^2}\|w_0 - w_*\|^2 + \frac{4\sigma^2(1 + \omega_{\text{up}})}{\mu K N b}\left(1 + \frac{64L\omega_{\text{dwn}}^2}{\mu K}\ln(\mu K + \widetilde{L})\right).$$

*Proof.* Let $k$ in $\mathbb{N}^*$, the proof starts like the one for Ghost, and we start from eq. (S13) but we consider a variable step size $\gamma_k = 2/(\mu(k+1) + \widetilde{L})$ that depends of the iteration $k$ in $\mathbb{N}$.

$$\begin{aligned}
\mathbb{E}\left[\|w_k - w_*\|^2\right] &\leq \mathbb{E}\left[\|w_{k-1} - w_*\|^2\right] - 2\gamma_k\mathbb{E}\left[\langle\nabla F(\widehat{w}_{k-1}), \widehat{w}_{k-1} - w_*\rangle\right] \\
&\quad + 2\gamma_k L\mathbb{E}\left[\|\widehat{w}_{k-1} - w_{k-1}\|^2\right] + \gamma_k^2\left(1 + \frac{\omega_{\text{up}}}{N}\right)\mathbb{E}\left[\|\nabla F(\widehat{w}_{k-1})\|^2\right] \\
&\quad + \frac{\gamma_k^2\sigma^2(1 + \omega_{\text{up}})}{Nb}.
\end{aligned}$$

Now we apply strong-convexity (eq. (S9) of Proposition S1):

$$\begin{aligned}
\mathbb{E}\left[\|w_k - w_*\|^2\right] &\leq \mathbb{E}\left[\|w_{k-1} - w_*\|^2\right] + 2\gamma_k L\mathbb{E}\left[\|\widehat{w}_{k-1} - w_{k-1}\|^2\right] \\
&\quad - \gamma_k\mathbb{E}\left[F(\widehat{w}_{k-1}) - F(w_*)\right] - \gamma_k\left(\mu\|\widehat{w}_{k-1} - w_*\|^2 + \frac{1}{L}\|\nabla F(\widehat{w}_{k-1})\|^2\right) \\
&\quad + \gamma_k^2\left(1 + \frac{\omega_{\text{up}}}{N}\right)\mathbb{E}\left[\|\nabla F(\widehat{w}_{k-1})\|^2\right] + \frac{\gamma_k^2\sigma^2(1 + \omega_{\text{up}})}{Nb}.
\end{aligned}$$

As $\gamma_k \leq \frac{2}{L} \leq \dfrac{1}{2L\left(1 + \frac{\omega_{\mathrm{up}}}{N}\right)}$, and thus $\left(1 - \gamma_k L\left(1 + \frac{\omega_{\mathrm{up}}}{N}\right)\right) \geq 1/2$; this allows to simplify the coefficient of $\mathbb{E}\left[\|\nabla F(\widehat{w}_{k-1})\|^2\right]$:

$$
\begin{aligned}
\mathbb{E}\left[\|w_k - w_*\|^2\right] \leq{} & (1 - \gamma_k \mu)\|w_{k-1} - w_*\|^2 - \gamma_k \mathbb{E}\left[F(\widehat{w}_{k-1}) - F(w_*)\right] \\
& - \frac{\gamma_k}{2L}\mathbb{E}\left[\|\nabla F(\widehat{w}_{k-1})\|^2\right] + 2\gamma_k L \mathbb{E}\left[\|\widehat{w}_{k-1} - w_{k-1}\|^2\right] \\
& + \frac{\gamma_k^2 \sigma^2 (1 + \omega_{\mathrm{up}})}{Nb}
\end{aligned}
$$

equivalent to:

$$
\begin{aligned}
\mathbb{E}\left[\|w_k - w_*\|^2\right] \leq{} & (1 - \gamma_k \mu)\|w_{k-1} - w_*\|^2 - \gamma_k \mathbb{E}\left[F(\widehat{w}_{k-1}) - F(w_*)\right] \\
& - \frac{\gamma_k}{2L}\mathbb{E}\left[\|\nabla F(\widehat{w}_{k-1})\|^2\right] + 2\gamma_k L \omega_{\mathrm{dwn}} \mathbb{E}\left[\|w_{k-1} - H_{k-1}\|^2\right] \\
& + \frac{\gamma_k^2 \sigma^2 (1 + \omega_{\mathrm{up}})}{Nb}\,.
\end{aligned}
\tag{S19}
$$

Theorem 3 adapted to the case of decaying steps gives:

$$
\begin{aligned}
\mathbb{E}\left[\Upsilon_k\right] \leq{} & \left(1 - \frac{\alpha_{\mathrm{dwn}}}{2}\right)\mathbb{E}\left[\Upsilon_{k-1}\right] + 2\gamma_k^2\left(\frac{1}{\alpha_{\mathrm{dwn}}} + \frac{\omega_{\mathrm{up}}}{N}\right)\mathbb{E}\left[\|\nabla F(\widehat{w}_{k-1})\|^2\right] \\
& + \frac{2\gamma_k^2 \sigma^2 (1 + \omega_{\mathrm{up}})}{Nb}\,.
\end{aligned}
\tag{S20}
$$

Defining $V_k := \mathbb{E}\left[\|w_k - w_*\|^2\right] + \gamma_k LC\mathbb{E}\left[\Upsilon_k\right]$ with $C = 4\omega_{\mathrm{dwn}}/\alpha$, combining the two later equations (S9) + $\gamma_k LC$ (S20):

$$
\begin{aligned}
\mathbb{E}&\left[\|w_k - w_*\|^2\right] + \gamma_k LC\mathbb{E}\left[\Upsilon_k\right] \\
\leq{} & (1 - \gamma_k \mu)\|w_{k-1} - w_*\|^2 - \gamma_k \mathbb{E}\left[F(\widehat{w}_{k-1}) - F(w_*)\right] \\
& - \frac{\gamma_k}{2L}\mathbb{E}\left[\|\nabla F(\widehat{w}_{k-1})\|^2\right] + 2\gamma_k L \omega_{\mathrm{dwn}} \mathbb{E}\left[\|w_{k-1} - H_{k-1}\|^2\right] + \frac{\gamma_k^2 \sigma^2 (1 + \omega_{\mathrm{up}})}{Nb} \\
& + \left(1 - \frac{\alpha_{\mathrm{dwn}}}{2}\right)\gamma_k LC\mathbb{E}\left[\Upsilon_{k-1}\right] + 2\gamma_k^3 LC\left(\frac{1}{\alpha_{\mathrm{dwn}}} + \frac{\omega_{\mathrm{up}}}{N}\right)\mathbb{E}\left[\|\nabla F(\widehat{w}_{k-1})\|^2\right] \\
& + \frac{2\gamma_k^3 L\sigma^2 (1 + \omega_{\mathrm{up}})C}{Nb}\,,
\end{aligned}
$$

and reordering the terms gives:

$$
\begin{aligned}
V_k \leq{} & (1 - \gamma_k \mu)\|w_{k-1} - w_*\|^2 - \gamma_k \mathbb{E}\left[F(\widehat{w}_{k-1}) - F(w_*)\right] \\
& + \left(1 - \frac{\alpha_{\mathrm{dwn}}}{2} + \frac{2\omega_{\mathrm{dwn}}}{C}\right)\gamma_k LC\mathbb{E}\left[\|w_{k-1} - H_{k-1}\|^2\right] \\
& + \left(2\gamma_k^3 LC\left(\frac{1}{\alpha_{\mathrm{dwn}}} + \frac{\omega_{\mathrm{up}}}{N}\right) - \frac{\gamma_k}{2L}\right)\mathbb{E}\left[\|\nabla F(\widehat{w}_{k-1})\|^2\right] \\
& + (2\gamma_k LC + 1)\frac{\gamma_k^2 \sigma^2 (1 + \omega_{\mathrm{up}})}{Nb}\,,
\end{aligned}
$$

To reach a $(1 - \gamma\mu)$-convergence we first need $\left(1 - \frac{\alpha_{\mathrm{dwn}}}{2} + \frac{2\omega_{\mathrm{dwn}}}{C}\right)\gamma_k LC \leq (1 - \gamma_k \mu)\gamma_{k-1} LC$ i.e $1 - \frac{\alpha_{\mathrm{dwn}}}{2} + \frac{2\omega_{\mathrm{dwn}}}{C} \leq \frac{(1 - \gamma_k \mu)\gamma_{k-1}}{\gamma_k}$.

We need that for all $k \in \mathbb{N}$, $\frac{1 - \gamma_k \mu}{\gamma_k} \leq \frac{1}{\gamma_{k-1}}$ i.e., $1 - \gamma_k \mu \leq \frac{\gamma_k}{\gamma_{k-1}}$, but:

$$\frac{\gamma_k}{\gamma_{k-1}} = \frac{\mu k - \mu + \widetilde{L}}{\mu k + \widetilde{L}} = 1 - \frac{\mu}{\mu k + \widetilde{L}} \quad \text{and} \quad 1 - \gamma_k \mu = 1 - \frac{2\mu}{\mu k + \widetilde{L}},$$

and so, the inequality is always true.

Thus we must have $2\omega_{\mathrm{dwn}}/C \leq \alpha_{\mathrm{dwn}}/2$ which is true by definition of $C$.

Secondly, it requires:

$$2\gamma_k^2 C\left(\frac{1}{\alpha_{\mathrm{dwn}}} + \frac{\omega_{\mathrm{up}}}{N}\right) - \frac{\gamma_k}{2L} \leq 0 \Longleftrightarrow \gamma_k L \leq \frac{1}{4C\left(\dfrac{1}{\alpha_{\mathrm{dwn}}} + \dfrac{\omega_{\mathrm{up}}}{N}\right)}$$

$$\Longleftrightarrow \gamma_k L \leq \frac{1}{4\sqrt{\dfrac{\omega_{\mathrm{dwn}}}{\alpha_{\mathrm{dwn}}}\left(\dfrac{1}{\alpha_{\mathrm{dwn}}} + \dfrac{\omega_{\mathrm{up}}}{N}\right)}},$$

by definition of $C$. And it follows that the first part of the theorem is proved:

$$V_k \leq (1 - \gamma_k \mu)V_{k-1} - \gamma_k \mathbb{E}\left[F(\widehat{w}_{k-1}) - F(w_*)\right] + \frac{\gamma_k^2 \sigma^2 \Phi(\gamma_k)}{Nb},$$

where $\Phi(\gamma_k) := (1 + \omega_{\mathrm{up}})\left(1 + \frac{8\gamma_k L \omega_{\mathrm{dwn}}}{\alpha_{\mathrm{dwn}}}\right)$.

We now prove the second part, which requires to carefully handle the term of noise. By definition $\gamma_k = \frac{2}{\mu(k+1) + L}$, we denote $\lambda_k = \frac{1}{\gamma_{k-1}}$ and we sum the above equation weighted with the sequence of $(\lambda_k)_{k=1}^K$:

$$\frac{1}{\sum_{k=1}^K \lambda_k} \sum_{k=1}^K \lambda_k \mathbb{E}\left[F(\widehat{w}_{k-1}) - F(w_*)\right] \leq \frac{1}{\sum_{k=1}^K \lambda_k} \sum_{k=1}^K \frac{(1 - \gamma_k \mu)\lambda_k}{\gamma_k} V_{k-1} - \frac{\lambda_k}{\gamma_k} V_k$$

$$+ \frac{1}{\sum_{k=1}^K \lambda_k} \sum_{k=1}^K \lambda_k \frac{\gamma_k \sigma^2 \Phi(\gamma_k)}{Nb}.$$

The weights are chosen to ensure that the sum of $(V_k)_{k=1}^K$ is telescopic. Because $(1 - \gamma_k \mu)/\gamma_k = \gamma_{k-2}^{-1}$, we have:

$$\frac{1}{\sum_{k=1}^K \lambda_k} \sum_{k=1}^K \lambda_k \mathbb{E}\left[F(\widehat{w}_{k-1}) - F(w_*)\right] \leq \frac{1}{\sum_{k=1}^K \lambda_k} \sum_{k=1}^K \frac{1}{\gamma_{k-2}\gamma_{k-1}} V_{k-1} - \frac{1}{\gamma_k \gamma_{k-1}} V_k$$

$$+ \frac{1}{\sum_{k=1}^K \lambda_k} \sum_{k=1}^K \lambda_k \frac{\gamma_k \sigma^2 \Phi(\gamma_k)}{Nb},$$

and because for $K \in \mathbb{N}^*$ big enough $\frac{1}{\sum_{k=1}^K \lambda_k} = \frac{1}{\mu(K+1)K/4 + (\widetilde{L}K)/2} \leq \frac{4}{\mu K^2}$, it results that:

$$\frac{1}{\sum_{k=1}^K \lambda_k} \sum_{k=1}^K \lambda_k \mathbb{E}\left[F(\widehat{w}_{k-1}) - F(w_*)\right] \leq \frac{V_0}{\gamma_0 \gamma_{-1} \mu K^2} + \frac{4}{\mu K^2} \sum_{k=1}^K \lambda_k \frac{\gamma_k \sigma^2 \Phi(\gamma_k)}{Nb}. \tag{S21}$$

At the end, using the Jensen inequality - $\mathbb{E}\left[\mathbb{E}\left[F(\widehat{w}_{k-1}) \mid w_{k-1}\right]\right] \leq \mathbb{E}\left[F(w_{k-1})\right]$, see Equation (S7) - we have for all $K$ in $\mathbb{N}$:

$$\frac{1}{\sum_{k=1}^{K} \lambda_k} \sum_{k=1}^{K} \lambda_k \mathbb{E}\left[F(w_{k-1}) - F(w_*)\right]$$

$$\leq \frac{V_0}{\gamma_0 \gamma_{-1} \mu K^2} + \frac{4}{\mu K^2} \sum_{k=1}^{K} \frac{1}{\gamma_{k-1}}\left(1 + \frac{8\gamma_k L \omega_{\mathrm{dwn}}}{\alpha_{\mathrm{dwn}}}\right) \frac{\gamma_k \sigma^2 (1 + \omega_{\mathrm{up}})}{Nb}$$

$$\leq \frac{V_0}{\gamma_0 \gamma_{-1} \mu K^2} + \frac{4}{\mu K^2} \sum_{k=1}^{K}\left(1 + \frac{8\gamma_k L \omega_{\mathrm{dwn}}}{\alpha_{\mathrm{dwn}}}\right) \frac{\sigma^2 (1 + \omega_{\mathrm{up}})}{Nb} ,$$

because for all $k$ in $N^*$, $\gamma_k \leq \gamma_{k-1}$. We need to compute the following classical sum:

$$\sum_{k=1}^{K} \frac{1}{\mu k + \widetilde{L}} \leq \int_{x=0}^{K} \frac{1}{\mu x + \widetilde{L}} \mathrm{dx} \leq \frac{1}{\mu} \ln\left(\mu K + \widetilde{L}\right) .$$

At the end, using again the Jensen inequality, defining $\widetilde{L} = \max\left\{4L\sqrt{\frac{\omega_{\mathrm{dwn}}}{\alpha_{\mathrm{dwn}}}\left(\frac{1}{\alpha_{\mathrm{dwn}}} + \frac{\omega_{\mathrm{up}}}{N}\right)}, 4L\left(1 + \frac{\omega_{\mathrm{up}}}{N}\right)\right\}$, taking for all $k$ in $\mathbb{N}$, $\gamma_k = \frac{2}{\mu(k+1) + \widetilde{L}}$, for all $k$ in $N^*$, $\lambda_k = \frac{1}{\gamma_{k-1}}$ and denoting $\bar{w}_K = \frac{\sum_{k=1}^{K} \lambda_k w_{k-1}}{\sum_{k=1}^{K} \lambda_k}$, then for any $K$ in $\mathbb{N}^*$, we have:

$$\mathbb{E}\left[F(\bar{w}_K) - F(w_*)\right] \leq \frac{\mu + 2\widetilde{L}}{4\mu K^2}\|w_0 - w_*\|^2 + \left(1 + \frac{64L\omega_{\mathrm{dwn}}^2}{\mu K}\ln\left(\mu K + \widetilde{L}\right)\right) \cdot \frac{4\sigma^2 (1 + \omega_{\mathrm{up}})}{\mu K Nb} ,$$

and the demonstration is completed.

$\square$

### E.4 Non-convex case (extra theorem)

In this section, we detail the convergence guarantee given for MCM in the non-convex case. In this scenario, the theorem will hold on the average of gradients after $K$ in $\mathbb{N}^*$ iterations. The structure of the proof is different from the one used for Ghost and MCM in convex and strongly-convex case. Instead, the demonstration starts from the equation resulting from smoothness and use the polarization identity to handle the inner product of gradients taken at two different points.

---

**Theorem S11** (Convergence of MCM in the non-convex case). *Under Assumptions 1, 2 and 4 (non-convex case), for a learning rate $\alpha_{\mathrm{dwn}} = \frac{1}{8\omega_{\mathrm{dwn}}}$, for any step size $\gamma$ s.t.*

$$\gamma L \leq \min\left\{\frac{1}{8\omega_{\mathrm{dwn}}}, \frac{1}{2\left(1 + \frac{\omega_{\mathrm{up}}}{N}\right)}, \frac{1}{8\sqrt{\omega_{\mathrm{dwn}}^2\left(8\omega_{\mathrm{dwn}} + \frac{\omega_{\mathrm{up}}}{N}\right)}}\right\} ,$$

*after running $K$ in $\mathbb{N}^*$ iterations, we have:*

$$\frac{1}{K}\sum_{k=1}^{K}\mathbb{E}\left[\|\nabla F(w_{k-1})\|^2\right] \leq \frac{2\left(F(w_0) - F(w_*)\right)}{\gamma K} + \frac{\gamma L \sigma^2 \Phi^{\mathrm{non-cvx}}(\gamma)}{Nb} ,$$

*with $\Phi^{\mathrm{non-cvx}}(\gamma) := (1 + \omega_{\mathrm{up}})\left(1 + 32\gamma L \omega_{\mathrm{dwn}}^2\right)$. Thus, for $K$ in $\mathbb{N}^*$ large enough, taking $\gamma = \sqrt{\frac{2Nb\left(F(w_0) - F(w_*)\right)}{\sigma^2 L(1 + \omega_{\mathrm{up}})K}}$:*

$$\frac{1}{K}\sum_{k=1}^{K}\mathbb{E}\left[\|\nabla F(w_{k-1})\|^2\right] \leq 2\sqrt{\frac{2L\sigma^2 (1 + \omega_{\mathrm{up}})\left(F(w_0) - F(w_*)\right)}{NbK}} + O(K^{-1}) .$$

---

*Proof.* Let $k$ in $\mathbb{N}^*$, then smoothness (see Assumption 2) implies:

$$F(w_k) \leq F(w_{k-1}) + \langle \nabla F(w_{k-1}),\ w_k - w_{k-1} \rangle + \frac{L}{2} \|w_k - w_{k-1}\|^2$$

$$\iff \quad F(w_k) \leq F(w_{k-1}) - \gamma \langle \nabla F(w_{k-1}),\ \widetilde{g}_k \rangle + \frac{\gamma^2 L}{2} \|\widetilde{g}_k\|^2 \ .$$

The inner product is not easy to handle because it implies two gradients computed at two different points: $w_{k-1}$ and $\widehat{w}_{k-1}$. To turn around this difficulty, we use the polarization identity, and so we have:

$$-\mathbb{E}\left[\langle \nabla F(w_{k-1}),\ \widetilde{g}_k \rangle \mid w_{k-1}\right] = -\langle \nabla F(w_{k-1}),\ \mathbb{E}\left[\nabla F(\widehat{w}_{k-1}) \mid w_{k-1}\right] \rangle$$
$$= \frac{1}{2}\left(-\|\nabla F(w_{k-1})\|^2 - \mathbb{E}\left[\|\nabla F(\widehat{w}_{k-1})\|^2 \ \middle|\ w_{k-1}\right]\right.$$
$$\left. + \mathbb{E}\left[\|\nabla F(w_{k-1}) - \nabla F(\widehat{w}_{k-1})\|^2 \ \middle|\ w_{k-1}\right]\right)$$

where we used the Polarization identity (eq. (S6)), and next with smoothness:

$$-\mathbb{E}\left[\langle \nabla F(w_{k-1}),\ \widetilde{g}_k \rangle \mid w_{k-1}\right] \leq \frac{1}{2}\left(-\|\nabla F(w_{k-1})\|^2 - \mathbb{E}\left[\|\nabla F(\widehat{w}_{k-1})\|^2 \ \middle|\ w_{k-1}\right]\right.$$
$$\left. + L^2 \mathbb{E}\left[\|w_{k-1} - \widehat{w}_{k-1}\|^2 \ \middle|\ w_{k-1}\right]\right) ,$$

Combining with Lemma S3, we obtain:

$$F(w_k) \leq F(w_{k-1}) - \frac{\gamma}{2}\|\nabla F(w_{k-1})\|^2 - \frac{\gamma}{2}\mathbb{E}\left[\|\nabla F(\widehat{w}_{k-1})\|^2 \ \middle|\ w_{k-1}\right]$$
$$+ \frac{\gamma L^2}{2}\mathbb{E}\left[\|w_{k-1} - \widehat{w}_{k-1}\|^2 \ \middle|\ w_{k-1}\right]$$
$$+ \frac{\gamma^2 L}{2}\left(\left(1 + \frac{\omega_{\mathrm{up}}}{N}\right)\|\nabla F(\widehat{w}_{k-1})\|^2 + \frac{\sigma^2(1 + \omega_{\mathrm{up}})}{Nb}\right) .$$

Taking the full expectation and re-ordering the terms gives:

$$\mathbb{E}\left[F(w_k)\right] \leq \mathbb{E}\left[F(w_{k-1})\right] - \frac{\gamma}{2}\mathbb{E}\left[\|\nabla F(w_{k-1})\|^2\right] - \frac{\gamma}{2}\left(1 - \gamma L\left(1 + \frac{\omega_{\mathrm{up}}}{N}\right)\right)\mathbb{E}\left[\|\nabla F(\widehat{w}_{k-1})\|^2\right]$$
$$+ \frac{\gamma L^2}{2}\mathbb{E}\left[\|w_{k-1} - \widehat{w}_{k-1}\|^2\right] + \frac{\gamma^2 L}{2} \times \frac{\sigma^2(1 + \omega_{\mathrm{up}})}{Nb} .$$

Exactly like the convex case, we consider that $\gamma L(1 + \omega_{\mathrm{up}}/N) \leq 1/2$ and because $\mathbb{E}\left[\|w_{k-1} - \widehat{w}_{k-1}\|^2\right] = \mathbb{E}\left[\mathbb{E}\left[\|w_{k-1} - \widehat{w}_{k-1}\|^2 \ \middle|\ \widehat{w}_{k-2}\right]\right]$ we can use Assumption 1:

$$\mathbb{E}\left[F(w_k)\right] \leq \mathbb{E}\left[F(w_{k-1})\right] - \frac{\gamma}{2}\mathbb{E}\left[\|\nabla F(w_{k-1})\|^2\right] - \frac{\gamma}{4}\mathbb{E}\left[\|\nabla F(\widehat{w}_{k-1})\|^2\right]$$
$$+ \frac{\omega_{\mathrm{dwn}}\gamma L^2}{2}\mathbb{E}\left[\Upsilon_k\right] + \frac{\gamma^2 L}{2} \times \frac{\sigma^2(1 + \omega_{\mathrm{up}})}{Nb} . \tag{S22}$$

Next, Theorem 3 gives:

$$\mathbb{E}\left[\Upsilon_k\right] \leq \left(1 - \frac{\alpha_{\mathrm{dwn}}}{2}\right)\mathbb{E}\left[\Upsilon_{k-1}\right] + 2\gamma^2\left(\frac{1}{\alpha_{\mathrm{dwn}}} + \frac{\omega_{\mathrm{up}}}{N}\right)\mathbb{E}\left[\|\nabla F(\widehat{w}_{k-1})\|^2\right] + \frac{2\gamma^2\sigma^2(1 + \omega_{\mathrm{up}})}{Nb} .$$

We iterate over $k$ and compute the resulting geometric sum, it gives:

$$\mathbb{E}\left[\Upsilon_k\right] \leq \left(1 - \frac{\alpha_{\mathrm{dwn}}}{2}\right)^k \|\Upsilon_0\|^2 + 2\gamma^2\left(\frac{1}{\alpha_{\mathrm{dwn}}} + \frac{\omega_{\mathrm{up}}}{N}\right)\sum_{t=1}^{k}\left(1 - \frac{\alpha}{2}\right)^{k-t}\mathbb{E}\left[\|\nabla F(\widehat{w}_{t-1})\|^2\right]$$
$$+ \frac{4\gamma^2\sigma^2(1 + \omega_{\mathrm{up}})}{\alpha_{\mathrm{dwn}}Nb} ,$$

where we considered for the last term of the above equation that $\sum_{t=1}^{k} \left(1 - \frac{\alpha_{\text{dwn}}}{2}\right)^{k} \leq \frac{2}{\alpha_{\text{dwn}}}$. This is equivalent to:

$$\mathbb{E}\left[\Upsilon_k\right] \leq 2\gamma^2 \left(\frac{1}{\alpha_{\text{dwn}}} + \frac{\omega_{\text{up}}}{N}\right) \sum_{t=1}^{k} \left(1 - \frac{\alpha_{\text{dwn}}}{2}\right)^{k-t} \mathbb{E}\left[\|\nabla F(\widehat{w}_{t-1})\|^2\right] + \frac{4\gamma^2\sigma^2(1+\omega_{\text{up}})}{\alpha_{\text{dwn}}Nb} \ .$$

We apply this last result to eq. (S22):

$$
\begin{aligned}
\frac{\gamma}{2}\mathbb{E}\left[\|\nabla F(w_{k-1})\|^2\right] &\leq \mathbb{E}\left[F(w_{k-1}) - F(w_k)\right] - \frac{\gamma}{4}\mathbb{E}\left[\|\nabla F(\widehat{w}_{k-1})\|^2\right] \\
&\quad + \frac{\gamma L^2}{2}\left(\frac{4\omega_{\text{dwn}}\gamma^2\sigma^2(1+\omega_{\text{up}})}{Nb\alpha_{\text{dwn}}}\right. \\
&\qquad\qquad \left. + 2\omega_{\text{dwn}}\gamma^2\left(\frac{1}{\alpha_{\text{dwn}}} + \frac{\omega_{\text{up}}}{N}\right)\sum_{t=1}^{k}\left(1 - \frac{\alpha_{\text{dwn}}}{2}\right)^{k-t}\mathbb{E}\left[\|\nabla F(\widehat{w}_{t-1})\|^2\right]\right) \\
&\quad + \frac{\gamma^2 L}{2} \times \frac{\sigma^2(1+\omega_{\text{up}})}{Nb} \\
&\leq \mathbb{E}\left[F(w_{k-1}) - F(w_k)\right] - \frac{\gamma}{4}\mathbb{E}\left[\|\nabla F(\widehat{w}_{k-1})\|^2\right] \\
&\quad + \gamma^3 L^2\omega_{\text{dwn}}\left(\frac{1}{\alpha_{\text{dwn}}} + \frac{\omega_{\text{up}}}{N}\right)\sum_{t=1}^{k}\left(1 - \frac{\alpha_{\text{dwn}}}{2}\right)^{k-t}\mathbb{E}\left[\|\nabla F(\widehat{w}_{t-1})\|^2\right] \\
&\quad + \frac{\gamma^2\sigma^2 L(1+\omega_{\text{up}})}{2Nb}\left(1 + \frac{4\gamma L\omega_{\text{dwn}}}{\alpha_{\text{dwn}}}\right) \ .
\end{aligned}
$$

Summing this equation, for $k$ in range 1 to $K$:

$$
\begin{aligned}
\frac{\gamma}{2}\sum_{k=1}^{K}\mathbb{E}\left[\|\nabla F(w_{k-1})\|^2\right] &\leq \mathbb{E}\left[F(w_0) - F(w_k)\right] - \frac{\gamma}{4}\sum_{k=1}^{K}\mathbb{E}\left[\|\nabla F(\widehat{w}_{k-1})\|^2\right] \\
&\quad + \gamma^3 L^2\omega_{\text{dwn}}\left(\frac{1}{\alpha_{\text{dwn}}} + \frac{\omega_{\text{up}}}{N}\right)\sum_{k=1}^{K}\sum_{t=1}^{k}\left(1 - \frac{\alpha_{\text{dwn}}}{2}\right)^{k-t}\mathbb{E}\left[\|\nabla F(\widehat{w}_{t-1})\|^2\right] \\
&\quad + \frac{\gamma^2\sigma^2 L(1+\omega_{\text{up}})}{2Nb}\left(1 + \frac{4\gamma L\omega_{\text{dwn}}}{\alpha_{\text{dwn}}}\right)K \ .
\end{aligned}
$$

We need to invert the double-sum and we obtain:

$$
\begin{aligned}
\frac{\gamma}{2}\sum_{k=1}^{K}\mathbb{E}\left[\|\nabla F(w_{k-1})\|^2\right] &\leq \gamma F(w_0) - F(w_k) - \frac{\gamma}{4}\sum_{i=1}^{K}\mathbb{E}\left[\|\nabla F(\widehat{w}_{k-1})\|^2\right] \\
&\quad + \frac{2}{\alpha_{\text{dwn}}} \times \gamma^3 L^2\omega_{\text{dwn}}\left(\frac{1}{\alpha_{\text{dwn}}} + \frac{\omega_{\text{up}}}{N}\right)\sum_{k=1}^{K}\mathbb{E}\left[\|\nabla F(\widehat{w}_{k-1})\|^2\right] \\
&\quad + \frac{\gamma^2\sigma^2 L(1+\omega_{\text{up}})}{2Nb}\left(1 + \frac{4\gamma L\omega_{\text{dwn}}}{\alpha_{\text{dwn}}}\right)K \\
&\leq \mathbb{E}\left[F(w_0) - F(w_k)\right] \\
&\quad + \left(2\gamma^3 L^2\frac{\omega_{\text{dwn}}}{\alpha_{\text{dwn}}}\left(\frac{1}{\alpha_{\text{dwn}}} + \frac{\omega_{\text{up}}}{N}\right) - \frac{\gamma}{4}\right)\sum_{k=1}^{K}\mathbb{E}\left[\|\nabla F(\widehat{w}_{k-1})\|^2\right] \\
&\quad + \frac{\gamma^2\sigma^2 L(1+\omega_{\text{up}})}{2Nb}\left(1 + \frac{4\gamma L\omega_{\text{dwn}}}{\alpha_{\text{dwn}}}\right)K \ .
\end{aligned}
$$

Now we consider that $2\gamma^3 L^2 \frac{\omega_{\mathrm{dwn}}}{\alpha_{\mathrm{dwn}}} \left( \frac{1}{\alpha_{\mathrm{dwn}}} + \frac{\omega_{\mathrm{up}}}{N} \right) \leq \gamma/4$, and because for all $k$ in $\mathbb{N}$, $F(w_0) - F(w_k) \leq F(w_0) - F(w_*)$:

$$\frac{1}{K} \sum_{k=1}^{K} \mathbb{E} \left[ \|\nabla F(w_{k-1})\|^2 \right] \leq \frac{2\left(F(w_0) - F(w_*)\right)}{\gamma K} + \frac{\gamma \sigma^2 L (1 + \omega_{\mathrm{up}})}{Nb} \left( 1 + \frac{4\gamma L \omega_{\mathrm{dwn}}}{\alpha_{\mathrm{dwn}}} \right) .$$

Finally, for any $K$ in $\mathbb{N}^*$, such that $\gamma L \leq \min \left\{ \dfrac{1}{8\omega_{\mathrm{dwn}}}, \dfrac{1}{2\left(1 + \frac{\omega_{\mathrm{up}}}{N}\right)}, \dfrac{1}{2\sqrt{2 \frac{\omega_{\mathrm{dwn}}}{\alpha_{\mathrm{dwn}}} \left( \frac{1}{\alpha_{\mathrm{dwn}}} + \frac{\omega_{\mathrm{up}}}{N} \right)}} \right\}$

and $\alpha_{\mathrm{dwn}} \leq \frac{1}{8\omega_{\mathrm{dwn}}}$, we have:

$$\frac{1}{K} \sum_{k=1}^{K} \mathbb{E} \left[ \|\nabla F(w_{k-1})\|^2 \right] \leq \frac{2\left(F(w_0) - F(w_*)\right)}{\gamma K} + \frac{\gamma L \sigma^2 \Phi^{\mathrm{non-cvx}}(\gamma)}{Nb} ,$$

denoting $\Phi^{\mathrm{non-cvx}}(\gamma) := (1 + \omega_{\mathrm{up}}) \left( 1 + \frac{4\gamma L \omega_{\mathrm{dwn}}}{\alpha_{\mathrm{dwn}}} \right)$.

Thus, for $K$ in $\mathbb{N}^*$ large enough, taking $\gamma = \sqrt{\dfrac{2Nb\left(F(w_0) - F(w_*)\right)}{\sigma^2 L (1 + \omega_{\mathrm{up}}) K}}$ and $\alpha_{\mathrm{dwn}} = 1/(8\omega_{\mathrm{dwn}})$:

$$\frac{1}{K} \sum_{k=1}^{K} \mathbb{E} \left[ \|\nabla F(w_{k-1})\|^2 \right] \leq 2\sqrt{\frac{2L\sigma^2 (1 + \omega_{\mathrm{up}}) \left(F(w_0) - F(w_*)\right)}{NbK}} + O(K^{-1}) .$$

$\square$

### E.5  Proof for `Rand-MCM` (Theorem 4)

The proof for `Rand-MCM` is almost identical to the `MCM`-scenario. It only requires to modify some notations because each device $i$ in $[\![1, N]\!]$ holds a unique model $\widehat{w}_{k-1}^i$.

For $k$ in $\mathbb{N}$:

1. $\widetilde{g}_k$ is now defined as $\widetilde{g}_k = \frac{1}{N} \sum_{i=1}^{N} \widehat{g}_k^i(\widehat{w}_{k-1}^i)$,

2. for all $i$ in $[\![1, N]\!]$, $\widehat{g}_k^i(\widehat{w}_{k-1})$ and $\nabla F(\widehat{w}_{k-1})$ must be replaced by $\widehat{g}_k^i(\widehat{w}_{k-1}^i)$ and $\nabla F(\widehat{w}_{k-1}^i)$,

3. instead of having a unique memory $H_k$, there is $N$ memories $(H_k^i)_{i=1}^N$ that keep track of the updates done on each worker,

4. furthermore the notation $w_{k-1} - H_{k-2}$ is no more correct as we have $N$ different memories. Thus, it must be replaced by $\frac{1}{N} \sum_{i=1}^{N} w_{k-1} - H_{k-2}^i$.

## F  Proofs in the quadratic case for `MCM` and `Rand-MCM`

In this section, for ease of notation we denote for $k$ in $\mathbb{N}^*$, $\widetilde{g}_k = \frac{1}{N} \sum_{i=1}^{N} \widehat{g}_k^i(\widehat{w}_{k-1}^i)$.

`MCM` has a unique memory $H_k$, and `Rand-MCM` has $N$ different memories $(H_k^i)_{i=1}^N$. But for the sake of factorization, we will consider that both algorithm have $N$ memories, thus we will always consider the quantity $\frac{1}{N} \sum_{i=1}^{N} \|w_{k-1} - H_{k-2}^i\|^2$, while we should consider the quantity $\frac{1}{N} \sum_{i=1}^{N} \|w_{k-1} - H_{k-2}\|^2$ for `MCM`. However this notation is correct considering that for `MCM`, for all $i$ in $[\![1, N]\!]$, $H_k^i = H_k$. And it follows that we have $\frac{1}{N} \sum_{i=1}^{N} \|w_{k-1} - H_{k-2}^i\|^2 = \frac{1}{N} \sum_{i=1}^{N} \|w_{k-1} - H_{k-2}^i\|^2$.

Unlike the previous sections where the proofs for MCM and Rand-MCM do not require any distinction, here in the quadratic case, we will on the contrary stress on the difference between the two. The difference appears in Lemma S4 and comes from the way we handle the expectation of $\left\| \frac{1}{N} \sum_{i=1}^{N} \nabla F(\widehat{w}_{k-1}^i) - \nabla F(w_{k-1}) \right\|^2$ for $k$ in $\mathbb{N}^*$. For this purpose we define a constant $\mathbf{C}$ such that $\mathbf{C} = 1$ in the MCM-case and $\mathbf{C} = N$ in the Rand-MCM-case.

The proofs for quadratic functions relies on the fact that for any $k$ in $\mathbb{N}^*$, $\mathbb{E}\left[\nabla F(\widehat{w}_{k-1} \mid w_{k-1}\right] = \nabla F(w_{k-1})$.

**Definition 2** (Quadratic function). *A function $f : \mathbb{R}^d \mapsto \mathbb{R}$ is said to be quadratic if there exists a symmetric matrix $A$ in $\mathcal{M}_{d,d}(\mathbb{R})$ such that for all $x$ in $\mathbb{R}^d$: $f(x) - f(x_*) = \frac{1}{2}(x - x_*)^T A (x - x_*)$. And then its gradient is defined for all $x$ in $\mathbb{R}^d$ as: $\nabla f(x) = A(x - x_*)$.*

### F.1 Two other lemmas

In this section, we detail two lemmas required to prove the convergence of MCM and Rand-MCM in the case of quadratic functions.

The first lemma allows to factorize all the results obtained for both MCM and Rand-MCM algorithms. For $k$ in $\mathbb{N}^*$ and $i$ in $[\![1, N]\!]$, the difference between the MCM-case and the Rand-MCM-case results from the tigher control of $\left\| \sum_{i=1}^{N} \nabla F(\widehat{w}_{k-1}^i) - \nabla F(w_{k-1}) \right\|^2$.

**Lemma S4.** *We define $\mathbf{C}$ such that $\mathbf{C} = 1$ in the MCM-case and $\mathbf{C} = N$ in the Rand-MCM-case. Then for any $k$ in $\mathbb{N}^*$, we have:*

$$\mathbb{E}\left[\left\|\frac{1}{N}\sum_{i=1}^{N}\nabla F(\widehat{w}_{k-1}^i) - \nabla F(w_{k-1})\right\|^2 \;\middle|\; w_{k-1}\right] \leq \frac{L^2 \omega_{\mathrm{dwn}}}{\mathbf{C}} \frac{1}{N}\sum_{i=1}^{N}\left\|w_{k-1} - H_{k-2}^i\right\|^2 .$$

*Proof.* Let $k$ in $\mathbb{N}^*$, we apply smoothness (see Assumption 2), and then we upper bound the variance of the quantization operator with Assumption 1. But we must distinguish MCM and Rand-MCM because in the first case we have $\widehat{w}_{k-1}^i$ equal to $\widehat{w}_{k-1}$ for all $i$ in $[\![1, N]\!]$.

In the MCM-case:

$$\mathbb{E}\left[\left\|\frac{1}{N}\sum_{i=1}^{N}\nabla F(\widehat{w}_{k-1}^i) - \nabla F(w_{k-1})\right\|^2 \;\middle|\; w_{k-1}\right] = \mathbb{E}\left[\nabla F(\widehat{w}_{k-1}) - F(w_{k-1}) \mid w_{k-1}\right]$$

$$\leq L^2 \mathbb{E}\left[\left\|\widehat{w}_{k-1} - w_{k-1}\right\|^2 \;\middle|\; w_{k-1}\right]$$

$$\leq L^2 \omega_{\mathrm{dwn}} \left\|\Omega_{k-1}\right\|^2$$

$$\leq L^2 \omega_{\mathrm{dwn}} \frac{1}{N}\sum_{i=1}^{N}\left\|w_{k-1} - H_{k-2}^i\right\|^2 ,$$

because we consider that $\left\|\Omega_{k-1}\right\|^2 = \left\|w_{k-1} - H_{k-2}\right\|^2 = \frac{1}{N}\sum_{i=1}^{N}\left\|w_{k-1} - H_{k-2}^i\right\|^2$.

In the Rand-MCM-case, by independence of the compressions on the downlink direction:

$$\mathbb{E}\left[\left\|\frac{1}{N}\sum_{i=1}^{N}\nabla F(\widehat{w}_{k-1}^i) - \nabla F(w_{k-1})\right\|^2 \;\middle|\; w_{k-1}\right] = \frac{1}{N^2}\sum_{i=1}^{N}\mathbb{E}\left[\left\|\nabla F(\widehat{w}_{k-1}^i) - \nabla F(w_{k-1})\right\|^2 \;\middle|\; w_{k-1}\right]$$

$$\leq \frac{L^2}{N^2}\sum_{i=1}^{N}\left\|\widehat{w}_{k-1}^i - w_{k-1}\right\|^2$$

$$\leq \frac{L^2 \omega_{\mathrm{dwn}}}{N} \times \frac{1}{N}\sum_{i=1}^{N}\left\|w_{k-1} - H_{k-2}^i\right\|^2$$

$$\leq \frac{L^2 \omega_{\mathrm{dwn}}}{N}\frac{1}{N}\sum_{i=1}^{N}\left\|w_{k-1} - H_{k-2}^i\right\|^2 .$$

We factorize the two results and define $\mathbf{C}$ such that $\mathbf{C} = 1$ in the MCM-case and $\mathbf{C} = N$ in the Rand-MCM-case, and the result follows.

$$\mathbb{E}\left[\left\|\frac{1}{N}\sum_{i=1}^{N}\nabla F(\widehat{w}_{k-1}^i) - \nabla F(w_{k-1})\right\|^2 \;\middle|\; w_{k-1}\right] \le \frac{L^2\omega_{\mathrm{dwn}}}{\mathbf{C}}\frac{1}{N}\sum_{i=1}^{N}\left\|w_{k-1} - H_{k-2}^i\right\|^2 \;.$$

$\square$

The next lemma replaces Lemma S3 in the context of randomization and quadratic functions. Note that the conditioning in Lemma S3 is w.r.t. to $\widehat{w}_{k-1}$ while here we take the expectation w.r.t. $w_{k-1}$. This is because we remove $\widehat{w}_{k-1}$ from the gradient and give a result which depends of $\|\nabla F(w_{k-1})\|^2$ instead of $\|\nabla F(\widehat{w}_{k-1})\|^2$. This is made possible by the fact that for all $k$ in $\mathbb{N}$, for quadratic functions, we have $\mathbb{E}\left[\nabla F(\widehat{w}_{k-1})\right] = \nabla F(w_{k-1})$.

**Lemma S5** (Squared-norm of stochastic gradients). *For any $k$ in $\mathbb{N}^*$, the squared-norm of gradients can be bounded a.s.:*

$$\mathbb{E}\left[\left\|\frac{1}{N}\sum_{i=1}^{N}\widehat{g}_k^i(\widehat{w}_{k-1}^i) - \nabla F(\widehat{w}_{k-1}^i)\right\|^2 \;\middle|\; w_{k-1}\right] \le \frac{\omega_{\mathrm{up}}}{N}\|\nabla F(w_{k-1})\|^2 + \frac{\sigma^2(1+\omega_{\mathrm{up}})}{Nb}$$

$$+ \frac{\omega_{\mathrm{up}}\omega_{\mathrm{dwn}}L^2}{N}\frac{1}{N}\sum_{i=1}^{N}\left\|w_{k-1} - H_{k-2}^i\right\|^2 , \tag{S23}$$

$$\mathbb{E}\left[\left\|\widetilde{g}_k\right\|^2 \;\middle|\; w_{k-1}\right] \le \left(1 + \frac{\omega_{\mathrm{up}}}{N}\right)\|\nabla F(w_{k-1})\|^2 + \frac{\sigma^2(1+\omega_{\mathrm{up}})}{Nb}$$

$$+ L^2\omega_{\mathrm{dwn}}\left(\frac{1}{\mathbf{C}} + \frac{\omega_{\mathrm{up}}}{N}\right)\frac{1}{N}\sum_{i=1}^{N}\left\|w_{k-1} - H_{k-2}^i\right\|^2 . \tag{S24}$$

The demonstration will be in two stages. We first show eq. (S23), and in a second time, we show eq. (S24).

*Proof.* Let $k$ in $\mathbb{N}^*$.

**First part (eq. (S23)).** We can decompose the squared-norm in two terms:

$$\mathbb{E}\left[\left\|\frac{1}{N}\sum_{i=1}^{N}\left(\widehat{g}_k^i(\widehat{w}_{k-1}^i) - \nabla F(\widehat{w}_{k-1}^i)\right)\right\|^2 \;\middle|\; w_{k-1}\right]$$

$$= \mathbb{E}\left[\left\|\frac{1}{N}\sum_{i=1}^{N}\left(\widehat{g}_k^i(\widehat{w}_{k-1}^i) - g_k^i(\widehat{w}_{k-1}^i)\right)\right\|^2 \;\middle|\; w_{k-1}\right]$$

$$+ \mathbb{E}\left[\left\|\frac{1}{N}\sum_{i=1}^{N}\left(g_k^i(\widehat{w}_{k-1}^i) - \nabla F(\widehat{w}_{k-1}^i)\right)\right\|^2 \;\middle|\; w_{k-1}\right] ,$$

the first term is bounded by Assumption 1 and the last term by Assumption 4:

$$\mathbb{E}\left[\left\|\frac{1}{N}\sum_{i=1}^{N}\left(\widehat{g}_k^i(\widehat{w}_{k-1}^i)-\nabla F(\widehat{w}_{k-1}^i)\right)\right\|^2 \;\middle|\; w_{k-1}\right]$$

$$\leq \frac{\omega_{\mathrm{up}}}{N^2}\sum_{i=1}^{N}\mathbb{E}\left[\left\|g_k^i(\widehat{w}_{k-1}^i)\right\|^2 \;\middle|\; w_{k-1}\right]+\frac{\sigma^2}{Nb}$$

$$\leq \frac{\omega_{\mathrm{up}}}{N^2}\sum_{i=1}^{N}\mathbb{E}\left[\left\|g_k^i(\widehat{w}_{k-1}^i)-\nabla F(\widehat{w}_{k-1}^i)\right\|^2 \;\middle|\; w_{k-1}\right]$$

$$+\frac{\omega_{\mathrm{up}}}{N^2}\sum_{i=1}^{N}\mathbb{E}\left[\left\|\nabla F(\widehat{w}_{k-1}^i)\right\|^2 \;\middle|\; w_{k-1}\right]+\frac{\sigma^2}{Nb}\,.$$

And again applying Assumption 4 on $\mathbb{E}\left[\left\|g_k^i(\widehat{w}_{k-1}^i)-\nabla F(\widehat{w}_{k-1}^i)\right\|^2 \;\middle|\; w_{k-1}\right]$ for $i$ in $\{1,\cdots N\}$:

$$\mathbb{E}\left[\left\|\frac{1}{N}\sum_{i=1}^{N}\left(\widehat{g}_k^i(\widehat{w}_{k-1}^i)-\nabla F(\widehat{w}_{k-1}^i)\right)\right\|^2 \;\middle|\; w_{k-1}\right]=\frac{\omega_{\mathrm{up}}}{N^2}\sum_{i=1}^{N}\mathbb{E}\left[\left\|\nabla F(\widehat{w}_{k-1}^i)\right\|^2 \;\middle|\; w_{k-1}\right]$$

$$+\frac{\sigma^2(1+\omega_{\mathrm{up}})}{Nb}\,.$$

Now, we have:

$$\frac{\omega_{\mathrm{up}}}{N^2}\sum_{i=1}^{N}\mathbb{E}\left[\left\|\nabla F(\widehat{w}_{k-1}^i)\right\|^2 \;\middle|\; w_{k-1}\right]=\frac{\omega_{\mathrm{up}}}{N^2}\sum_{i=1}^{N}\mathbb{E}\left[\left\|\nabla F(\widehat{w}_{k-1}^i)-\nabla F(w_{k-1})\right\|^2 \;\middle|\; w_{k-1}\right]$$

$$+\frac{\omega_{\mathrm{up}}}{N^2}\sum_{i=1}^{N}\mathbb{E}\left[\left\|\nabla F(w_{k-1})\right\|^2 \;\middle|\; w_{k-1}\right]\,,$$

using smoothness (Assumption 2) gives:

$$\frac{\omega_{\mathrm{up}}}{N^2}\sum_{i=1}^{N}\mathbb{E}\left[\left\|\nabla F(\widehat{w}_{k-1}^i)\right\|^2 \;\middle|\; w_{k-1}\right]=\frac{\omega_{\mathrm{up}}\omega_{\mathrm{dwn}}L^2}{N}\frac{1}{N}\sum_{i=1}^{N}\left\|w_{k-1}-H_{k-2}^i\right\|^2+\frac{\omega_{\mathrm{up}}}{N}\left\|\nabla F(w_{k-1})\right\|^2\,,$$

and putting everythings together allows to conclude for eq. (S23).

**Second part (eq. (S24)).**   We start by introducing $\|\nabla F(w_{k-1})\|^2$:

$$\mathbb{E}\left[\left\|\frac{1}{N}\sum_{i=1}^{N}\widehat{g}_k^i(\widehat{w}_{k-1}^i)\right\|^2 \;\middle|\; w_{k-1}\right]=\mathbb{E}\left[\left\|\frac{1}{N}\sum_{i=1}^{N}\widehat{g}_k^i(\widehat{w}_{k-1}^i)-\nabla F(w_{k-1})\right\|^2 \;\middle|\; w_{k-1}\right]$$

$$+\|\nabla F(w_{k-1})\|^2$$

$$=\mathbb{E}\left[\left\|\frac{1}{N}\sum_{i=1}^{N}\widehat{g}_k^i(\widehat{w}_{k-1}^i)-\nabla F(\widehat{w}_{k-1}^i)\right\|^2 \;\middle|\; w_{k-1}\right]$$

$$+\mathbb{E}\left[\left\|\frac{1}{N}\sum_{i=1}^{N}\nabla F(\widehat{w}_{k-1}^i)-\nabla F(w_{k-1})\right\|^2 \;\middle|\; w_{k-1}\right]$$

$$+\|\nabla F(w_{k-1})\|^2\,.$$

The second term of the previous line is controlled by Lemma S4 which distinguish the MCM and Rand-MCM-cases by defining a constant $\mathbf{C}$ such that $\mathbf{C}=1$ for MCM and $\mathbf{C}=N$ for Rand-MCM:

$$\mathbb{E}\left[\left\|\frac{1}{N}\sum_{i=1}^{N}\nabla F(\widehat{w}_{k-1}^i)-\nabla F(w_{k-1})\right\|^2 \;\middle|\; w_{k-1}\right]\leq\frac{L^2\omega_{\mathrm{dwn}}}{\mathbf{C}}\frac{1}{N}\sum_{i=1}^{N}\left\|w_{k-1}-H_{k-2}^i\right\|^2\,.$$

Thus, we have:

$$\mathbb{E}\left[\left\|\frac{1}{N}\sum_{i=1}^{N}\widehat{g}_k^i(\widehat{w}_{k-1}^i)\right\|^2 \;\middle|\; w_{k-1}\right] = \mathbb{E}\left[\left\|\frac{1}{N}\sum_{i=1}^{N}\widehat{g}_k^i(\widehat{w}_{k-1}^i) - \nabla F(\widehat{w}_{k-1}^i)\right\|^2 \;\middle|\; w_{k-1}\right]$$
$$+ \frac{\omega_{\mathrm{dwn}}L^2}{\mathbf{C}}\frac{1}{N}\sum_{i=1}^{N}\left\|w_{k-1} - H_{k-2}^i\right\|^2 + \|\nabla F(w_{k-1})\|^2 \;,$$

and eq. (S23) allows to conclude. $\qquad\square$

## F.2 Control of the Variance of the local model for quadratic function (both `MCM` and `Rand-MCM`)

The next theorem replaces the Theorem 3 in the case of quadratic functions. The results are almost identical except that in these settings we control the variance using non-degraded points $(w_t)_{t\in\mathbb{N}}$. This is necessary because, for quadratic functions, the analysis is slightly different. Previously, we upper-bounded the inner product in the decomposition (eq. (S12)) by a "strong contraction" that was allowing to subtract $\|\nabla F(\widehat{w}_{k-1})\|^2$ and an extra residual term. Here we instead directly get a smaller contraction proportional to $\|\nabla F(w_{k-1})\|^2$ (but without any residual!). Indeed for all $k$ in $\mathbb{N}$, we have $\mathbb{E}\left[\nabla F(\widehat{w}_{k-1})\right] = \nabla F(w_{k-1})$. This difference will appear in Appendix F.3.

As a consequence, we need to also control the variance of the local iterates that will appear when expanding the expected squared gradient $\mathbb{E}\|\tilde{g}_k\|^2$ by an affine function of the squared norms of the gradients **at the non perturbed points**. This is what Theorem S12 provides.

---

**Theorem S12.** *Consider the `MCM` update as in eq. (2) or the `Rand-MCM` update as described in Subsection 2.2. Under Assumptions 1 to 4 with $\mu = 0$, if $\gamma \leq \dfrac{1}{8L\omega_{\mathrm{dwn}}\sqrt{(1/\mathbf{C} + \omega_{\mathrm{up}}/N)}}$ and $\alpha_{\mathrm{dwn}} \leq 1/(8\omega_{\mathrm{dwn}})$, then for all $k$ in $\mathbb{N}$:*

$$\frac{1}{N}\sum_{i=1}^{N}\mathbb{E}\left[\left\|w_k - H_{k-1}^i\right\|^2 \;\middle|\; w_{k-1}\right]$$
$$\leq 2\gamma^2\left(\frac{1}{\alpha_{\mathrm{dwn}}} + \frac{\omega_{\mathrm{up}}}{N}\right)\sum_{t=1}^{k}(1 - \frac{\alpha_{\mathrm{dwn}}}{2})^{k-t}\mathbb{E}\left[\|\nabla F(w_{t-1})\|^2 \;\middle|\; w_{t-1}\right]$$
$$+ \frac{4\gamma^2\sigma^2(1 + \omega_{\mathrm{up}})}{\alpha_{\mathrm{dwn}}Nb}\;.$$

---

*Proof.* Let $k$ in $\mathbb{N}^*$ and $i$ in $\{1, \ldots N\}$, from Theorem S8 we have:

$$\mathbb{E}\left[\left\|w_k - H_{k-1}^i\right\|^2 \;\middle|\; w_{k-1}\right] = \mathrm{Var} + \mathrm{Bias}^2 = 2\gamma^2\mathrm{Var}_1 + 2\alpha_{\mathrm{dwn}}^2\mathrm{Var}_2 + \mathrm{Bias}^2\;,$$

with

$$\begin{cases} \mathrm{Var}_1 & = \mathbb{E}\left[\left\|\frac{1}{N}\sum_{i=1}^{N}\widehat{g}_k^i(\widehat{w}_{k-1}^i) + \mathbb{E}\left[\nabla F(\widehat{w}_{k-1}^i) \;\middle|\; w_{k-1}\right]\right\|^2 \;\middle|\; w_{k-1}\right] \\ \mathrm{Var}_2 & = \omega_{\mathrm{dwn}}\frac{1}{N}\sum_{i=1}^{N}\left\|w_{k-1} - H_{k-2}^i\right\|^2 \\ \mathrm{Bias}^2 & = \|\mathbb{E}\left[w_k - H_{k-1} \;\middle|\; w_{k-1}\right]\|^2\;. \end{cases}$$

Recall that in the case of quadratic functions, we have for all $i$ in $[\![1, N]\!]$: $\mathbb{E}\left[\nabla F(\widehat{w}_{k-1}^i) \mid w_{k-1}\right] = \nabla F(w_{k-1})$. And so for the first term of variance we can decompose as following:

$$
\begin{aligned}
\mathrm{Var}_1 &= \mathbb{E}\left[\left\|\frac{1}{N}\sum_{i=1}^{N}\widehat{g}_k^i(\widehat{w}_{k-1}^i) - \mathbb{E}\left[\nabla F(\widehat{w}_{k-1}^i) \mid w_{k-1}\right]\right\|^2 \;\middle|\; w_{k-1}\right] \\
&= \mathbb{E}\left[\left\|\frac{1}{N}\sum_{i=1}^{N}\widehat{g}_k^i(\widehat{w}_{k-1}^i) - \nabla F(w_{k-1})\right\|^2 \;\middle|\; w_{k-1}\right] \\
&= \mathbb{E}\left[\left\|\frac{1}{N}\sum_{i=1}^{N}\widehat{g}_k^i(\widehat{w}_{k-1}^i) - \nabla F(\widehat{w}_{k-1}^i)\right\|^2 \;\middle|\; w_{k-1}\right] \\
&\quad + \mathbb{E}\left[\left\|\frac{1}{N}\sum_{i=1}^{N}\nabla F(\widehat{w}_{k-1}^i) - \nabla F(w_{k-1})\right\|^2 \;\middle|\; w_{k-1}\right].
\end{aligned}
$$

The first part is handled by eq. (S23) of Lemma S5:

$$
\begin{aligned}
\mathbb{E}\left[\left\|\frac{1}{N}\sum_{i=1}^{N}\widehat{g}_k^i(\widehat{w}_{k-1}^i) - \nabla F(\widehat{w}_{k-1}^i)\right\|^2 \;\middle|\; w_{k-1}\right] &= \frac{\omega_{\mathrm{up}}\omega_{\mathrm{dwn}}L^2}{N}\frac{1}{N}\sum_{i=1}^{N}\left\|w_{k-1} - H_{k-2}^i\right\|^2 \\
&\quad + \frac{\omega_{\mathrm{up}}}{N}\left\|\nabla F(w_{k-1})\right\|^2 \\
&\quad + \frac{\sigma^2(1+\omega_{\mathrm{up}})}{Nb},
\end{aligned}
$$

and the second part is tackled by Lemma S4 where is defined a constant $\mathbf{C}$ such that $\mathbf{C} = 1$ in the MCM-case, and $\mathbf{C} = N$ in the Rand-MCM-case: $\mathbb{E}\left[\left\|\frac{1}{N}\sum_{i=1}^{N}\nabla F(\widehat{w}_{k-1}^i) - \nabla F(w_{k-1})\right\|^2 \;\middle|\; w_{k-1}\right] \leq \frac{L^2\omega_{\mathrm{dwn}}}{\mathbf{C}}\frac{1}{N}\sum_{i=1}^{N}\left\|w_{k-1} - H_{k-2}^i\right\|^2$.

Finally, given that $\mathrm{Var} = 2\gamma^2\mathrm{Var}_1 + 2\alpha_{\mathrm{dwn}}^2\mathrm{Var}_2$ we have:

$$
\begin{aligned}
\mathrm{Var} &\leq 2\gamma^2 L^2\omega_{\mathrm{dwn}}\left(\frac{1}{\mathbf{C}} + \frac{\omega_{\mathrm{up}}}{N}\right)\frac{1}{N}\sum_{i=1}^{N}\left\|w_{k-1} - H_{k-2}^i\right\|^2 + 2\alpha_{\mathrm{dwn}}^2\omega_{\mathrm{dwn}}\left\|w_{k-1} - H_{k-2}^i\right\|^2 \\
&\quad + \frac{2\gamma^2\omega_{\mathrm{up}}}{N}\left\|\nabla F(w_{k-1})\right\|^2 + \frac{2\gamma^2\sigma^2(1+\omega_{\mathrm{up}})}{Nb}.
\end{aligned}
$$

Now we focus on the squared bias $\mathrm{Bias}^2$ exactly like in Theorem S8 and we obtain:

$$
\mathrm{Bias}^2 \leq (1-\alpha_{\mathrm{dwn}})\left\|w_{k-1} - H_{k-2}^i\right\|^2 + \gamma^2(1+\frac{1}{\alpha_{\mathrm{dwn}}})\left\|\nabla F(w_{k-1})\right\|^2.
$$

At the end:

$$
\begin{aligned}
\mathbb{E}\left[\left\|w_k - H_{k-1}^i\right\|^2 \;\middle|\; w_{k-1}\right] &\leq 2\gamma^2 L^2\omega_{\mathrm{dwn}}\left(\frac{1}{\mathbf{C}} + \frac{\omega_{\mathrm{up}}}{N}\right)\frac{1}{N}\sum_{i=1}^{N}\left\|w_{k-1} - H_{k-2}^i\right\|^2 \\
&\quad + \gamma^2(1 + \frac{1}{\alpha_{\mathrm{dwn}}} + \frac{2\omega_{\mathrm{up}}}{N})\left\|\nabla F(w_{k-1})\right\|^2 \\
&\quad + \left((1-\alpha_{\mathrm{dwn}}) + 2\alpha_{\mathrm{dwn}}^2\omega_{\mathrm{dwn}}\right)\left\|w_{k-1} - H_{k-2}^i\right\|^2 \\
&\quad + \frac{2\gamma^2\sigma^2(1+\omega_{\mathrm{up}})}{Nb}.
\end{aligned}
$$

Summing this last equation over the $N$ devices gives:

$$\frac{1}{N} \sum_{i=1}^{N} \mathbb{E}\left[\left\|w_k - H_{k-1}^i\right\|^2 \mid w_{k-1}\right]$$

$$\leq \left(1 - \alpha_{\mathrm{dwn}} + 2\alpha_{\mathrm{dwn}}^2 \omega_{\mathrm{dwn}} + \gamma^2 L^2 \omega_{\mathrm{dwn}} \left(\frac{1}{\mathbf{C}} + \frac{\omega_{\mathrm{up}}}{N}\right)\right) \frac{1}{N} \sum_{i=1}^{N} \left\|w_{k-1} - H_{k-2}^i\right\|^2$$

$$+ \gamma^2 (1 + \frac{1}{\alpha_{\mathrm{dwn}}} + \frac{2\omega_{\mathrm{up}}}{N}) \left\|\nabla F(w_{k-1})\right\|^2$$

$$+ \frac{2\gamma^2 \sigma^2 (1 + \omega_{\mathrm{up}})}{Nb} .$$

Exactly like in Theorem S8, we need and by taking $\alpha_{\mathrm{dwn}} = 1/(8\omega_{\mathrm{dwn}})$:

$$\begin{cases} 2\alpha_{\mathrm{dwn}}^2 \omega_{\mathrm{dwn}} \leq \frac{1}{4}\alpha_{\mathrm{dwn}} \Longleftrightarrow \alpha_{\mathrm{dwn}} \leq \frac{1}{8\omega_{\mathrm{dwn}}} , \\ 2\gamma^2 L^2 \omega_{\mathrm{dwn}} \left(\frac{1}{\mathbf{C}} + \frac{\omega_{\mathrm{up}}}{N}\right) \leq \frac{1}{4}\alpha_{\mathrm{dwn}} = \frac{1}{32\omega_{\mathrm{dwn}}} \Longleftrightarrow \gamma \leq \frac{1}{8L\omega_{\mathrm{dwn}}\sqrt{(1/\mathbf{C} + \omega_{\mathrm{up}}/N)}} , \\ 1 + \frac{1}{\alpha_{\mathrm{dwn}}} \leq \frac{2}{\alpha_{\mathrm{dwn}}} \text{ which is not restrictive.} \end{cases}$$

Thus, we can write:

$$\frac{1}{N} \sum_{i=1}^{N} \mathbb{E}\left[\left\|w_k - H_{k-1}^i\right\|^2 \mid w_{k-1}\right] \leq \left(1 - \frac{\alpha_{\mathrm{dwn}}}{2}\right) \frac{1}{N} \sum_{i=1}^{N} \left\|w_{k-1} - H_{k-2}^i\right\|^2$$

$$+ 2\gamma^2 (\frac{1}{\alpha_{\mathrm{dwn}}} + \frac{\omega_{\mathrm{up}}}{N}) \left\|\nabla F(w_{k-1})\right\|^2$$

$$+ \frac{2\gamma^2 \sigma^2 (1 + \omega_{\mathrm{up}})}{Nb} .$$

Finally, we take the full expectation without any conditioning, we iterate over $k$ and compute the geometric sums:

$$\frac{1}{N} \sum_{i=1}^{N} \mathbb{E}\left[\left\|w_k - H_{k-1}^i\right\|^2\right] \leq (1 - \frac{\alpha_{\mathrm{dwn}}}{2})^k \left\|w_0 - H_{-1}\right\|^2 + \frac{4\gamma^2 \sigma^2 (1 + \omega_{\mathrm{up}})}{\alpha_{\mathrm{dwn}} Nb}$$

$$+ 2\gamma^2 (\frac{1}{\alpha_{\mathrm{dwn}}} + \frac{\omega_{\mathrm{up}}}{N}) \sum_{t=1}^{k} (1 - \frac{\alpha_{\mathrm{dwn}}}{2})^{k-t} \mathbb{E}\left[\left\|\nabla F(w_{t-1})\right\|^2\right] .$$

and the result follows.

$\square$

### F.3 Proof for quadratic function (Theorem 5)

**Theorem S13.** *Under Assumptions 1 to 4 with $\mu = 0$, if the function is quadratic, for $\gamma = 1/(L\sqrt{K})$ and a given learning rate $\alpha_{\mathrm{dwn}} = 1/(8\omega_{\mathrm{dwn}})$, after running $K$ iterations:*

$$\mathbb{E}\left[F(\bar{w}_K) - F_*\right] \leq \frac{\left\|w_0 - w_*\right\|^2 L}{\sqrt{K}} + \frac{\sigma^2 \Phi(\gamma)}{NbL\sqrt{K}} .$$

*with $\Phi = (1 + \omega_{\mathrm{up}}) \left(1 + 32 \frac{\omega_{\mathrm{dwn}}^2}{\sqrt{K}} \times \frac{1}{\sqrt{K}} \left(\frac{1}{\mathbf{C}} + \frac{\omega_{\mathrm{up}}}{N}\right)\right)$ and $\mathbf{C} = N$ for `Rand-MCM`, and 1 for `MCM`.*

The structure of the proof is different from the one used in Appendices D and E.

*Proof.* Let $k$ in $\mathbb{N}$, by definition:

$$\left\|w_k - w_*\right\|^2 \leq \left\|w_{k-1} - w_*\right\|^2 - 2\gamma \left\langle \widetilde{g}_k, w_{k-1} - w_* \right\rangle + \gamma^2 \left\|\widetilde{g}_k\right\|^2 .$$

Because $F$ is quadratic, we have $\mathbb{E}\left[\nabla F(\widehat{w}_{k-1}) \mid w_{k-1}\right] = \nabla F(w_{k-1})$, thus taking expectation gives:

$$\mathbb{E}\left[\|w_k - w_*\|^2 \,\Big|\, w_{k-1}\right] \leq \|w_{k-1} - w_*\|^2 - 2\gamma\left\langle \nabla F(w_{k-1}),\, w_{k-1} - w_* \right\rangle + \gamma^2 \mathbb{E}\left[\|\widetilde{g}_k\|^2 \,\Big|\, w_{k-1}\right].$$

We can directly apply convexity with eq. (S8) from Proposition S1:

$$\mathbb{E}\left[\|w_k - w_*\|^2 \,\Big|\, w_{k-1}\right] \leq \|w_{k-1} - w_*\|^2 - \gamma\left(F(w_{k-1}) - F(w_*) + \frac{1}{L}\|\nabla F(w_{k-1})\|^2\right)$$
$$+ \gamma^2 \mathbb{E}\left[\|\widetilde{g}_k\|^2 \,\Big|\, w_{k-1}\right].$$

Now, with eq. (S24) of Lemma S5:

$$\mathbb{E}\left[\|w_k - w_*\|^2 \,\Big|\, w_{k-1}\right] \leq \|w_{k-1} - w_*\|^2 - \gamma(F(w_{k-1}) - F(w_*)) - \frac{\gamma}{L}\|\nabla F(w_{k-1})\|^2$$
$$+ \gamma^2\left(\left(1 + \frac{\omega_{\text{up}}}{N}\right)\|\nabla F(w_{k-1})\|^2\right.$$
$$+ L^2\omega_{\text{dwn}}\left(\frac{1}{\mathbf{C}} + \frac{\omega_{\text{up}}}{N}\right)\frac{1}{N}\sum_{i=1}^{N}\left\|w_{k-1} - H_{k-2}^i\right\|^2$$
$$\left.+ \frac{\sigma^2(1 + \omega_{\text{up}})}{Nb}\right),$$

which gives:

$$\mathbb{E}\left[\|w_k - w_*\|^2 \,\Big|\, w_{k-1}\right] \leq \|w_{k-1} - w_*\|^2 - \gamma(F(w_{k-1}) - F(w_*)) - \frac{\gamma}{L}\|\nabla F(w_{k-1})\|^2$$
$$+ \gamma^2\left(1 + \frac{\omega_{\text{up}}}{N}\right)\|\nabla F(w_{k-1})\|^2$$
$$+ \gamma^2 L^2\omega_{\text{dwn}}\left(\frac{1}{\mathbf{C}} + \frac{\omega_{\text{up}}}{N}\right)\frac{1}{N}\sum_{i=1}^{N}\left\|w_{k-1} - H_{k-2}^i\right\|^2$$
$$+ \frac{\sigma^2\gamma^2(1 + \omega_{\text{up}})}{Nb}.$$

Taking full expectation, and because for all $i$ in $\{1, \cdots, N\}$, $\mathbb{E}\left[\left\|w_{k-1} - H_{k-2}^i\right\|^2\right] = \mathbb{E}\left[\mathbb{E}\left[\left\|w_{k-1} - H_{k-2}^i\right\|^2 \,\Big|\, \widehat{w}_{k-2}\right]\right]$, we can use the inequality controlling $\frac{1}{N}\sum_{i=1}^{N}\left\|w_{k-1} - H_{k-2}^i\right\|^2$ (see Theorem S12):

$$\mathbb{E}\left[\|w_k - w_*\|^2\right] \leq \mathbb{E}\left[\|w_{k-1} - w_*\|^2\right] - \gamma\mathbb{E}\left[F(w_{k-1}) - F(w_*)\right]$$
$$- \frac{\gamma}{L}\left(1 - \gamma L\left(1 + \frac{\omega_{\text{up}}}{N}\right)\right)\mathbb{E}\left[\|\nabla F(w_{k-1})\|^2\right]$$
$$+ \gamma^2 L^2\omega_{\text{dwn}}\left(\frac{1}{\mathbf{C}} + \frac{\omega_{\text{up}}}{N}\right) \times 2\gamma^2\left(\frac{1}{\alpha_{\text{dwn}}} + \frac{\omega_{\text{up}}}{N}\right)\sum_{t=1}^{k}(1 - \frac{\alpha_{\text{dwn}}}{2})^{k-t}\mathbb{E}\left[\|\nabla F(w_{t-1})\|^2\right]$$
$$+ \frac{\sigma^2\gamma^2(1 + \omega_{\text{up}})}{Nb} + \gamma^2 L^2\omega_{\text{dwn}}\left(\frac{1}{\mathbf{C}} + \frac{\omega_{\text{up}}}{N}\right) \times \frac{4\sigma^2\gamma^2(1 + \omega_{\text{up}})}{\alpha_{\text{dwn}}Nb}.$$

Next, we consider - as in previous proofs - that $\gamma L(1 + \omega_{\mathrm{up}}/N) \leq 1/2$, and thus $\frac{\gamma}{L}\left(1 - \gamma L\left(1 + \frac{\omega_{\mathrm{up}}}{N}\right)\right) \geq \frac{\gamma}{2}$. Next we carry out the "top-down recurrence":

$$
\mathbb{E}\left[\|w_k - w_*\|^2\right] \leq \|w_0 - w_*\|^2 - \gamma \sum_{j=1}^{k} \mathbb{E}\left[F(w_{k-j}) - F(w_*)\right]
$$

$$
- \frac{\gamma}{2L} \sum_{j=1}^{k} \mathbb{E}\left[\|\nabla F(w_{k-j-1})\|^2\right]
$$

$$
+ \sum_{j=1}^{k} 2\gamma^4 L^2 \omega_{\mathrm{dwn}} \left(\frac{1}{\mathbf{C}} + \frac{\omega_{\mathrm{up}}}{N}\right)\left(\frac{1}{\alpha_{\mathrm{dwn}}} + \frac{\omega_{\mathrm{up}}}{N}\right) \sum_{t=1}^{k-j}\left(1 - \frac{\alpha_{\mathrm{dwn}}}{2}\right)^{k-j-t} \mathbb{E}\left[\|\nabla F(w_{t-1})\|^2\right]
$$

$$
+ \sum_{j=1}^{k} \frac{\gamma^2 \sigma^2 (1 + \omega_{\mathrm{up}})}{Nb}\left(1 + \frac{4\gamma^2 L^2 \omega_{\mathrm{dwn}}}{\alpha_{\mathrm{dwn}}}\left(\frac{1}{\mathbf{C}} + \frac{\omega_{\mathrm{up}}}{N}\right)\right).
$$

We invert the double-sum, it leads to:

$$
\mathbb{E}\left[\|w_k - w_*\|^2\right] \leq \|w_0 - w_*\|^2 - \gamma \sum_{j=1}^{k} \mathbb{E}\left[F(w_{j-1}) - F(w_*)\right]
$$

$$
- \frac{\gamma}{2L}\mathbb{E}\left[\|\nabla F(w_{k-1})\|^2\right]
$$

$$
+ \frac{2}{\alpha_{\mathrm{dwn}}} \times 2\gamma^4 L^2 \omega_{\mathrm{dwn}}\left(\frac{1}{\mathbf{C}} + \frac{\omega_{\mathrm{up}}}{N}\right)\left(\frac{1}{\alpha_{\mathrm{dwn}}} + \frac{\omega_{\mathrm{up}}}{N}\right)\mathbb{E}\left[\|\nabla F(w_{-1})\|^2\right]
$$

$$
+ \sum_{j=1}^{k-1}\left(\frac{2}{\alpha_{\mathrm{dwn}}} \times 2\gamma^4 L^2 \omega_{\mathrm{dwn}}\left(\frac{1}{\mathbf{C}} + \frac{\omega_{\mathrm{up}}}{N}\right)\left(\frac{1}{\alpha_{\mathrm{dwn}}} + \frac{\omega_{\mathrm{up}}}{N}\right) - \frac{\gamma}{2L}\right)\mathbb{E}\left[\|\nabla F(w_{j-1})\|^2\right]
$$

$$
+ \frac{\gamma^2 \sigma^2 (1 + \omega_{\mathrm{up}})}{Nb}\left(1 + \frac{4\gamma^2 L^2 \omega_{\mathrm{dwn}}}{\alpha_{\mathrm{dwn}}}\left(\frac{1}{\mathbf{C}} + \frac{\omega_{\mathrm{up}}}{N}\right)\right) \times k.
$$

Now, we consider that $\frac{4\omega_{\mathrm{dwn}}\gamma^4 L^2}{\alpha_{\mathrm{dwn}}}\left(\frac{1}{\mathbf{C}} + \frac{\omega_{\mathrm{up}}}{N}\right)\left(\frac{1}{\alpha_{\mathrm{dwn}}} + \frac{\omega_{\mathrm{up}}}{N}\right) < \frac{\gamma}{2L}$, thus we have:

$$
\frac{\gamma}{k}\sum_{t=1}^{k}\mathbb{E}\left[F(w_{t-1}) - F(w_*)\right] \leq \frac{\|w_0 - w_*\|^2}{k} + \frac{\gamma^2 \sigma^2 (1 + \omega_{\mathrm{up}})}{Nb}\left(1 + \frac{4\gamma^2 L^2 \omega_{\mathrm{dwn}}}{\alpha_{\mathrm{dwn}}}\left(\frac{1}{\mathbf{C}} + \frac{\omega_{\mathrm{up}}}{N}\right)\right).
$$

Finally, by Jensen, for any $K$ in $\mathbb{N}^*$, taking $\gamma$ such that:

$$
\gamma L \leq \min\left\{ \frac{1}{8\omega_{\mathrm{dwn}}\sqrt{\frac{1}{\mathbf{C}} + \frac{\omega_{\mathrm{up}}}{N}}},\ \frac{1}{2\left(1 + \frac{\omega_{\mathrm{up}}}{N}\right)},\ \frac{1}{\sqrt[3]{\frac{8\omega_{\mathrm{dwn}}}{\alpha_{\mathrm{dwn}}}\left(\frac{1}{\mathbf{C}} + \frac{\omega_{\mathrm{up}}}{N}\right)\left(\frac{1}{\alpha_{\mathrm{dwn}}} + \frac{\omega_{\mathrm{up}}}{N}\right)}} \right\}
$$

and with $\alpha_{\mathrm{dwn}} \leq \frac{1}{8\omega_{\mathrm{dwn}}}$, we recover Theorem 5:

$$
\mathbb{E}\left[F(\bar{w}_K) - F(w_*)\right] \leq \frac{\|w_0 - w_*\|^2}{\gamma K} + \frac{\gamma \sigma^2 \Phi^{\mathrm{Rd}}(\gamma)}{Nb},
$$

denoting $\Phi^{\mathrm{Rd}}(\gamma) = (1 + \omega_{\mathrm{up}})\left(1 + \frac{4\gamma^2 L^2 \omega_{\mathrm{dwn}}}{K}\left(\frac{1}{\mathbf{C}} + \frac{\omega_{\mathrm{up}}}{N}\right)\right).$

$\square$

# G  Adataptation to the heterogeneous scenario

In this section, we give the complete proof of Theorems 1 and 2 in the case of heterogeneous workers.

> We choose to not merge the proofs in the homogeneous and heterogeneous cases. This is to avoid the technicalities associated with the heterogeneity and the uplink compression (that have been extensively studied in previous works [34, 18, 28, 36]) in the proof of our main results which aim at alleviating the impact of downlink compression. We thus propose two proofs that can be read almost independently in order to make proof-checking easier. We stress that the result in the homogeneous setting is not exactly a consequence of the heterogeneous case (the constants are degraded in the heterogeneous framework) but merging the proofs is ultimately possible.

Appendix G.1 first presents some lemmas from [36] required to handle the additional uplink memory. Lemma S6 (resp. Lemma S7) corresponds to Lemma S5 (resp. Lemma S7) evaluated at point $\widehat{w}_{k-1}$; and Lemma S8 corresponds to Lemma S13. Secondly, Appendix G.2 gives the demonstration of MCM. We denote $\Phi^{\text{Heterog}}(\gamma) := (1 + 8\omega_{\text{up}})\left(1 + \frac{8\gamma L \omega_{\text{dwn}}}{\alpha_{\text{dwn}}}\right)$ and $\gamma_{\max}^{\text{Heterog}}$ such that:

$$\gamma_{\max}^{\text{Heterog}} L \leq \min\left\{\gamma_{\max}, \frac{1}{16\frac{\omega_{\text{up}}}{N}}, \frac{1}{8\sqrt{2\frac{\omega_{\text{dwn}}}{\alpha_{\text{dwn}}} \cdot \frac{\omega_{\text{up}}}{N}}}\right\}.$$

We make the following assumption on the heterogeneity.

**Assumption 6** (Bounded gradient at $w_*$). *There is a constant $B$ in $\mathbb{R}_+$, s.t.: $\frac{1}{N}\sum_{i=0}^{N} \|\nabla F_i(w_*)\|^2 = B^2$. And we denote for all $i$ in $[\![1, N]\!]$, $h_*^i = \nabla F_i(w_*)$.*

## G.1  Control of the uplink memory

In this section we give the theorems that are required by the uplink memory.

**Lemma S6** (Bounding the compressed term). *The squared norm of the compressed term sent by each node to the central server can be bounded as following:*

$$\forall k \in \mathbb{N}, \forall i \in [\![1, N]\!], \quad \left\|\Delta_k^i\right\|^2 \leq 2\left(\left\|g_k^i(\widehat{w}_{k-1}) - h_*^i\right\|^2 + \left\|h_k^i - h_*^i\right\|^2\right).$$

**Lemma S7** (Noise over local gradients). *Let $k \in N^*$ and $i \in [\![1, N]\!]$. The noise in the stochastic gradients as defined in Assumptions 4 and 6 can be controlled as following:*

$$\frac{1}{N^2}\sum_{i=1}^{N}\mathbb{E}\left[\left\|g_k^i(\widehat{w}_{k-1}) - h_*^i\right\|^2 \mid w_{k-1}\right] \leq \frac{2L}{N}\mathbb{E}\left[\langle \nabla F(\widehat{w}_{k-1}), \widehat{w}_{k-1} - w_* \rangle \mid w_{k-1}\right] + \frac{2\sigma^2}{Nb}.$$

**Lemma S8** (Recursive inequalities over memory term). *Let $k \in \mathbb{N}$ and let $i \in [\![1, N]\!]$. The memory term used in the uplink broadcasting can be bounded using a recursion:*

$$\mathbb{E}\left[\Xi_k \mid w_{k-1}\right] \leq (1 - \alpha_{\text{up}})\mathbb{E}\left[\Xi_{k-1} \mid w_{k-1}\right]$$
$$+ \frac{2\alpha_{\text{up}} L}{N}\mathbb{E}\left[\langle \nabla F(\widehat{w}_{k-1}), \widehat{w}_{k-1} - w_* \rangle \mid w_{k-1}\right]$$
$$+ \frac{2}{N}\frac{\sigma^2}{b}\alpha_{\text{up}}.$$

**Lemma S9** (Squared-norm of stochastic gradients). *For any $k$ in $\mathbb{N}^*$, the squared-norm of gradients can be bounded a.s.:*

$$\mathbb{E}\left[\|\widetilde{g}_k\|^2 \mid \widehat{w}_{k-1}\right] \leq \left(1 + \frac{4\omega_{\text{up}}}{N}\right)L\mathbb{E}\left[\langle \nabla F(\widehat{w}_{k-1}), \widehat{w}_{k-1} - w_* \rangle \mid \widehat{w}_{k-1}\right]$$
$$+ 2\omega_{\text{up}}\mathbb{E}\left[\Xi_{k-1} \mid \widehat{w}_{k-1}\right] + \frac{\sigma^2}{Nb}(1 + 4\omega_{\text{up}}),$$

$$\mathbb{E}\left[\|\widetilde{g}_k - \nabla F(\widehat{w}_{k-1})\|^2 \mid \widehat{w}_{k-1}\right] \leq \frac{4\omega_{\text{up}} L}{N}\mathbb{E}\left[\langle \nabla F(\widehat{w}_{k-1}), \widehat{w}_{k-1} - w_* \rangle \mid \widehat{w}_{k-1}\right]$$
$$+ 2\omega_{\text{up}}\mathbb{E}\left[\Xi_{k-1} \mid \widehat{w}_{k-1}\right] + \frac{\sigma^2}{Nb}(1 + 4\omega_{\text{up}}),$$

*Lemma S9 extends Lemma S3.*

*Proof.* Let $k$ in $\mathbb{N}^*$, then:

$$\mathbb{E}\left[\|\widetilde{g}_k\|^2 \,\Big|\, \widehat{w}_{k-1}\right] = \|\nabla F(\widehat{w}_{k-1})\|^2 + \mathbb{E}\left[\|\widetilde{g}_k - \nabla F(\widehat{w}_{k-1})\|^2 \,\Big|\, \widehat{w}_{k-1}\right]$$

$$\leq L\mathbb{E}\left[\langle\, \nabla F(\widehat{w}_{k-1}),\, \widehat{w}_{k-1} - w_* \,\rangle \mid \widehat{w}_{k-1}\right] + \mathbb{E}\left[\|\widetilde{g}_k - \nabla F(\widehat{w}_{k-1})\|^2 \,\Big|\, \widehat{w}_{k-1}\right]$$

Secondly:

$$\mathbb{E}\left[\|\widetilde{g}_k - \nabla F(\widehat{w}_{k-1})\|^2 \,\Big|\, \widehat{w}_{k-1}\right]$$

$$= \mathbb{E}\left[\left\|\frac{1}{N}\sum_{i=1}^N\left(\widehat{\Delta}_{k-1}^i + h_{k-1}^i - \nabla F_i(\widehat{w}_{k-1})\right)\right\|^2 \,\Bigg|\, \widehat{w}_{k-1}\right]$$

$$= \mathbb{E}\left[\left\|\frac{1}{N}\sum_{i=1}^N\left(\widehat{\Delta}_{k-1}^i + h_{k-1}^i - g_k^i(\widehat{w}_{k-1}) + g_k^i(\widehat{w}_{k-1}) - \nabla F_i(\widehat{w}_{k-1})\right)\right\|^2 \,\Bigg|\, \widehat{w}_{k-1}\right]$$

$$= \mathbb{E}\left[\left\|\frac{1}{N}\sum_{i=1}^N\widehat{\Delta}_{k-1}^i - \Delta_{k-1}^i\right\|^2 \,\Bigg|\, \widehat{w}_{k-1}\right] + \mathbb{E}\left[\left\|\frac{1}{N}\sum_{i=1}^N g_k^i(\widehat{w}_{k-1}) - \nabla F_i(\widehat{w}_{k-1})\right\|^2 \,\Bigg|\, \widehat{w}_{k-1}\right],$$

the inner product being null.

Next, expanding the squared norm again, and because the two sums of inner products are null as the stochastic oracle and uplink compressions are independent:

$$\mathbb{E}\left[\|\widetilde{g}_k - \nabla F(\widehat{w}_{k-1})\|^2 \,\Big|\, \widehat{w}_{k-1}\right] = \frac{1}{N^2}\sum_{i=1}^N\mathbb{E}\left[\left\|\widehat{\Delta}_{k-1}^i - \Delta_{k-1}^i\right\|^2 \,\Big|\, \widehat{w}_{k-1}\right]$$

$$+ \frac{1}{N^2}\sum_{i=1}^N\mathbb{E}\left[\left\|g_k^i(\widehat{w}_{k-1}) - \nabla F_i(\widehat{w}_{k-1})\right\|^2 \,\Big|\, \widehat{w}_{k-1}\right].$$

Then, for any $i$ in $[\![1, N]\!]$ as $\mathbb{E}\left[\left\|\widehat{\Delta}_{k-1}^i - \Delta_{k-1}^i\right\|^2 \,\Big|\, \widehat{w}_{k-1}\right] = \mathbb{E}\left[\mathbb{E}\left[\left\|\widehat{\Delta}_{k-1}^i - \Delta_{k-1}^i\right\|^2 \,\Big|\, g_k^i\right] \,\Big|\, \widehat{w}_{k-1}\right]$, and using Assumption 1 we have:

$$\mathbb{E}\left[\|\widetilde{g}_k - \nabla F(\widehat{w}_{k-1})\|^2 \,\Big|\, \widehat{w}_{k-1}\right] \leq \frac{\omega_{\text{up}}}{N^2}\sum_{i=1}^N\mathbb{E}\left[\left\|\Delta_{k-1}^i\right\|^2 \,\Big|\, \widehat{w}_{k-1}\right]$$

$$+ \frac{1}{N^2}\sum_{i=1}^N\mathbb{E}\left[\left\|g_k^i(\widehat{w}_{k-1}) - \nabla F_i(\widehat{w}_{k-1})\right\|^2 \,\Big|\, \widehat{w}_{k-1}\right].$$

Furthermore with Lemma S6 and Assumption 4:

$$\mathbb{E}\left[\|\widetilde{g}_k - \nabla F(\widehat{w}_{k-1})\|^2 \,\Big|\, \widehat{w}_{k-1}\right] \leq \frac{2\omega_{\text{up}}}{N^2}\sum_{i=1}^N\mathbb{E}\left[\left\|g_k(\widehat{w}_{k-1}) - h_*^i\right\|^2 \,\Big|\, \widehat{w}_{k-1}\right]$$

$$+ 2\omega_{\text{up}}\mathbb{E}\left[\Xi_{k-1} \mid \widehat{w}_{k-1}\right] + \frac{\sigma^2}{Nb}.$$

And finally with Lemma S7:

$$\mathbb{E}\left[\|\widetilde{g}_k - \nabla F(\widehat{w}_{k-1})\|^2 \,\Big|\, \widehat{w}_{k-1}\right] \leq \frac{4\omega_{\text{up}}L}{N}\mathbb{E}\left[\langle\, \nabla F(\widehat{w}_{k-1}),\, \widehat{w}_{k-1} - w_* \,\rangle \mid \widehat{w}_{k-1}\right] + \frac{4\omega_{\text{up}}\sigma^2}{Nb}$$

$$+ 2\omega_{\text{up}}\mathbb{E}\left[\Xi_{k-1} \mid \widehat{w}_{k-1}\right] + \frac{\sigma^2}{Nb},$$

from which we derive the two inequalities of the lemma.

$$\square$$

## G.2 Proofs for `MCM`

In this section, we provide the demonstration of Theorems 1 and 2 in the convex and strongly-convex cases with heterogeneous workers.

### G.2.1 Control of the Variance of the local model for `MCM`

In this section, the aim is to control the variance of the local model for `MCM` but in the setting of heterogeneous worker, as done previously in Theorem S8.

---

**Theorem S14.** *Consider the `MCM` update as in eq.* (2). *Under Assumptions 1, 2 and 4, if* $\gamma \leq 1/(8\omega_{\mathrm{dwn}}L)$ *and* $\alpha_{\mathrm{dwn}} \leq 1/(8\omega_{\mathrm{dwn}})$, *then for all $k$ in* $\mathbb{N}$:

$$
\mathbb{E}\left[\Upsilon_k \mid w_{k-1}\right] \leq \left(1 - \frac{\alpha_{\mathrm{dwn}}}{2}\right)\Upsilon_{k-1}
$$
$$
+ 2\gamma^2 L\left(\frac{1}{\alpha_{\mathrm{dwn}}} + \frac{4\omega_{\mathrm{up}}}{N}\right)\mathbb{E}\left[\langle\,\nabla F(\widehat{w}_{k-1}),\,\widehat{w}_{k-1} - w_* \,\rangle \mid \widehat{w}_{k-1}\right]
$$
$$
+ 4\gamma^2\omega_{\mathrm{up}}\mathbb{E}\left[\Xi_{k-1} \mid \widehat{w}_{k-1}\right] + \frac{2\gamma^2\sigma^2(1 + 4\omega_{\mathrm{up}})}{Nb}.
$$

---

*Proof.* Let $k$ in $\mathbb{N}$, we recall that by definition:

$$
\begin{cases}
\Omega_k = w_k - H_{k-1} \\
\widehat{\Omega}_k = \mathcal{C}_{\mathrm{dwn}}(\Omega_k) \\
\widehat{w}_k = \widehat{\Omega}_k + H_{k-1}\,.
\end{cases}
$$

We start the proof by performing a bias-variance decomposition, and exactly like in the proof of Theorem S8, we obtain:

$$
\|\Omega_k\|^2 = \mathrm{Bias}^2 + 2\gamma^2\mathrm{Var}_{12} + 2\gamma^2\mathrm{Var}_{12} + 2\alpha_{\mathrm{dwn}}^2\mathrm{Var}_2
$$

We first have:

$$
\mathrm{Var}_{11} = \mathbb{E}\left[\left\|\widetilde{g}_k - \nabla F(\widehat{w}_{k-1})\right\|^2 \,\Big|\, w_{k-1}\right] = \mathbb{E}\left[\mathbb{E}\left[\left\|\widetilde{g}_k - \nabla F(\widehat{w}_{k-1})\right\|^2 \,\Big|\, \widehat{w}_{k-1}\right] \,\Big|\, w_{k-1}\right],
$$

so, we can use Lemma S9:

$$
\mathrm{Var}_{11} = \frac{4\omega_{\mathrm{up}}L}{N}\mathbb{E}\left[\langle\,\nabla F(\widehat{w}_{k-1}),\,\widehat{w}_{k-1} - w_* \,\rangle \mid \widehat{w}_{k-1}\right] + 2\omega_{\mathrm{up}}\mathbb{E}\left[\Xi_{k-1} \mid \widehat{w}_{k-1}\right] + \frac{\sigma^2}{Nb}(1 + 4\omega_{\mathrm{up}}).
$$

The other terms are exactly as before in Theorem S8:

$$
\begin{cases}
\mathrm{Var}_{12} \leq L^2\omega_{\mathrm{dwn}}\Upsilon_{k-1} \\
\mathrm{Var}_2 \leq \omega_{\mathrm{dwn}}\Upsilon_{k-1} \\
\mathrm{Bias}^2 \leq (1 - \alpha_{\mathrm{dwn}})\Upsilon_{k-1} + \gamma^2 L(1 + \frac{1}{\alpha_{\mathrm{dwn}}})\mathbb{E}\left[\langle\,\nabla F(\widehat{w}_{k-1}),\,\widehat{w}_{k-1} - w_* \,\rangle \mid \widehat{w}_{k-1}\right].
\end{cases}
$$

At the end:

$$
\mathbb{E}\left[\Upsilon_k \mid w_{k-1}\right] \leq (1 - \alpha_{\mathrm{dwn}})\Upsilon_{k-1} + \gamma^2 L(1 + \frac{1}{\alpha_{\mathrm{dwn}}})\mathbb{E}\left[\langle\,\nabla F(\widehat{w}_{k-1}),\,\widehat{w}_{k-1} - w_* \,\rangle \mid \widehat{w}_{k-1}\right]
$$
$$
+ \frac{8\omega_{\mathrm{up}}\gamma^2 L}{N}\mathbb{E}\left[\langle\,\nabla F(\widehat{w}_{k-1}),\,\widehat{w}_{k-1} - w_* \,\rangle \mid \widehat{w}_{k-1}\right]
$$
$$
+ 4\gamma^2\omega_{\mathrm{up}}\mathbb{E}\left[\Xi_{k-1} \mid \widehat{w}_{k-1}\right] + \frac{2\gamma^2\sigma^2}{Nb}(1 + 4\omega_{\mathrm{up}})
$$
$$
+ 2\gamma^2 L^2\omega_{\mathrm{dwn}}\Upsilon_{k-1} + 2\alpha_{\mathrm{dwn}}^2\omega_{\mathrm{dwn}}\Upsilon_{k-1},
$$

which is equivalent to:

$$\mathbb{E}\left[\Upsilon_k \mid w_{k-1}\right] \leq \left(1 - \alpha_{\mathrm{dwn}} + 2\gamma^2 L^2 \omega_{\mathrm{dwn}} + 2\alpha_{\mathrm{dwn}}^2 \omega_{\mathrm{dwn}}\right) \|w_{k-1} - \Xi_{k-2}\|^2$$

$$+ \gamma^2 L \left(1 + \frac{1}{\alpha_{\mathrm{dwn}}} + \frac{8\omega_{\mathrm{up}}}{N}\right) \mathbb{E}\left[\langle \nabla F(\widehat{w}_{k-1}), \widehat{w}_{k-1} - w_* \rangle \mid \widehat{w}_{k-1}\right]$$

$$+ 4\gamma^2 \omega_{\mathrm{up}} \mathbb{E}\left[\Xi_{k-1} \mid \widehat{w}_{k-1}\right] + \frac{2\gamma^2 \sigma^2 (1 + 4\omega_{\mathrm{up}})}{Nb} .$$

Next, we require as in Theorem S8:

$$\begin{cases} 2\alpha_{\mathrm{dwn}}^2 \omega_{\mathrm{dwn}} \leq \frac{1}{4}\alpha_{\mathrm{dwn}} \iff \alpha_{\mathrm{dwn}} \leq \frac{1}{8\omega_{\mathrm{dwn}}} , \\ 2\gamma^2 L^2 \omega_{\mathrm{dwn}} \leq \frac{1}{4}\alpha_{\mathrm{dwn}} = \frac{1}{32\omega_{\mathrm{dwn}}} , \text{ by taking } \alpha_{\mathrm{dwn}} = \frac{1}{8\omega_{\mathrm{dwn}}} \iff \gamma \leq \frac{1}{8\omega_{\mathrm{dwn}}L} , \\ 1 + \frac{1}{\alpha_{\mathrm{dwn}}} \leq \frac{2}{\alpha_{\mathrm{dwn}}} \text{ which is not restrictive if } \omega_{\mathrm{dwn}} \geq 1, \end{cases}$$

and it leads to the final result taking unconditional expectation. $\qquad\square$

### G.2.2 Convex case

> **Theorem S15** (Convergence of MCM in the heterogeneous and convex case). *Under Assumptions 1 to 4 with $\mu = 0$ (convex case), for learning rates $\alpha_{\mathrm{dwn}} \leq \frac{1}{8\omega_{\mathrm{dwn}}}$ and $\alpha_{\mathrm{up}}(1 + \omega_{\mathrm{up}}) \leq 1$, taking a step size s.t. $\gamma \leq \gamma_{\max}^{\mathrm{Heterog}}$, for any $k$ in $\mathbb{N}$, defining:*
>
> $$V_k := \mathbb{E}\left[\|w_k - w_*\|^2\right] + \gamma^2 C_1 \mathbb{E}[\Xi_k] + \gamma L C_2 \mathbb{E}[\Upsilon_k] ,$$
>
> *with $C_1 = 2\omega_{\mathrm{up}}(1 + 8\gamma L\omega_{\mathrm{dwn}}/\alpha_{\mathrm{dwn}})/\alpha_{\mathrm{up}}$, $C_2 = 4\gamma L\omega_{\mathrm{dwn}}/\alpha_{\mathrm{dwn}}$, we have:*
>
> $$V_k \leq V_{k-1} - \gamma\mathbb{E}\left[F(\widehat{w}_{k-1}) - F(w_*)\right] + \frac{\gamma^2\sigma^2\Phi^{\mathrm{Heterog}}(\gamma)}{Nb} .$$

*Proof.* We denote for $k$ in $\mathbb{N}^*$ $\widetilde{g}_k = \frac{1}{N}\sum_{i=1}^N \widehat{\Delta}_{k-1}^i + h_{k-1}^i$ with $\Delta_{k-1}^i = g_k^i(\widehat{w}_{k-1}) - h_{k-1}^i$, and $\Xi_k = \frac{1}{N^2}\sum_{i=1}^N \mathbb{E}\left[\|h_k^i - \nabla F_i(w_*)\|^2 \mid \widehat{w}_{k-1}\right]$.

Let $k$ in $\mathbb{N}^*$, by definition:

$$\|w_k - w_*\|^2 \leq \|w_{k-1} - w_*\|^2 - 2\gamma\langle \widetilde{g}_k, w_{k-1} - w_* \rangle + \gamma^2 \|\widetilde{g}_k\|^2 .$$

Next, we expend the inner product as following:

$$\|w_k - w_*\|^2 \leq \|w_{k-1} - w_*\|^2 - 2\gamma\langle \widetilde{g}_k, \widehat{w}_{k-1} - w_* \rangle - 2\gamma\langle \widetilde{g}_k, w_{k-1} - \widehat{w}_{k-1} \rangle + \gamma^2 \|\widetilde{g}_k\|^2 .$$

Taking expectation conditionally to $w_{k-1}$, and using $\mathbb{E}\left[\widetilde{g}_k \mid w_{k-1}\right] = \mathbb{E}\left[\mathbb{E}\left[\widetilde{g}_k \mid \widehat{w}_{k-1}\right] \mid w_{k-1}\right] = \mathbb{E}\left[\nabla F(\widehat{w}_{k-1}) \mid w_{k-1}\right]$, we obtain:

$$\mathbb{E}\left[\|w_k - w_*\|^2 \mid w_{k-1}\right] \leq \|w_{k-1} - w_*\|^2 - \mathbb{E}\left[2\gamma\langle \nabla F(\widehat{w}_{k-1}), \widehat{w}_{k-1} - w_* \rangle \mid w_{k-1}\right]$$

$$- 2\gamma\mathbb{E}\left[\langle \nabla F(\widehat{w}_{k-1}), w_{k-1} - \widehat{w}_{k-1} \rangle \mid w_{k-1}\right]$$

$$+ \gamma^2\mathbb{E}\left[\|\widetilde{g}_k\|^2 \mid w_{k-1}\right] .$$

Then invoking Lemma S3 to upper bound the squared norm of the stochastic gradients, and noticing that $\mathbb{E}\left[\langle \nabla F(w_{k-1}), \widehat{w}_{k-1} - w_{k-1} \rangle \mid w_{k-1}\right] = 0$ leads to:

$$\mathbb{E}\left[\|w_k - w_*\|^2 \mid w_{k-1}\right] \leq \|w_{k-1} - w_*\|^2 - 2\gamma\mathbb{E}\left[\langle \nabla F(\widehat{w}_{k-1}), \widehat{w}_{k-1} - w_* \rangle \mid w_{k-1}\right]$$

$$- 2\gamma\mathbb{E}\left[\langle \nabla F(\widehat{w}_{k-1}) - \nabla F(w_{k-1}), w_{k-1} - \widehat{w}_{k-1} \rangle \mid w_{k-1}\right] \quad \text{(S25)}$$

$$+ \gamma^2\left(\left(1 + \frac{4\omega_{\mathrm{up}}}{N}\right) L\mathbb{E}\left[\langle \nabla F(\widehat{w}_{k-1}), \widehat{w}_{k-1} - w_* \rangle \mid \widehat{w}_{k-1}\right]\right.$$

$$\left. + 2\omega_{\mathrm{up}}\mathbb{E}\left[\Xi_{k-1} \mid \widehat{w}_{k-1}\right] + \frac{\sigma^2}{Nb}(1 + 4\omega_{\mathrm{up}})\right) .$$

Now using Cauchy-Schwarz inequality (eq. (S5)) and smoothness:

$$
\begin{aligned}
- \mathbb{E}\left[2\gamma \left\langle \nabla F(\widehat{w}_{k-1}) - \nabla F(w_{k-1}),\ w_{k-1} - \widehat{w}_{k-1} \right\rangle \mid w_{k-1}\right] \\
= 2\gamma \mathbb{E}\left[\left\langle \nabla F(\widehat{w}_{k-1}) - \nabla F(w_{k-1}),\ \widehat{w}_{k-1} - w_{k-1} \right\rangle \mid w_{k-1}\right] \\
\leq 2\gamma L \mathbb{E}\left[\left\| \widehat{w}_{k-1} - w_{k-1} \right\|^2 \;\Big|\; w_{k-1}\right],
\end{aligned}
$$

and thus:

$$
\begin{aligned}
\mathbb{E}\left[\left\| w_k - w_* \right\|^2 \;\Big|\; w_{k-1}\right] \leq&\ \left\| w_{k-1} - w_* \right\|^2 - 2\gamma \mathbb{E}\left[\left\langle \nabla F(\widehat{w}_{k-1}),\ \widehat{w}_{k-1} - w_* \right\rangle \mid w_{k-1}\right] \\
&+ 2\gamma L \mathbb{E}\left[\left\| \widehat{w}_{k-1} - w_{k-1} \right\|^2 \;\Big|\; w_{k-1}\right] \\
&+ \left(1 + \frac{4\omega_{\mathrm{up}}}{N}\right)\gamma^2 L \mathbb{E}\left[\left\langle \nabla F(\widehat{w}_{k-1}),\ \widehat{w}_{k-1} - w_* \right\rangle \mid \widehat{w}_{k-1}\right] \\
&+ 2\omega_{\mathrm{up}}\gamma^2 \mathbb{E}\left[\Xi_{k-1} \mid \widehat{w}_{k-1}\right] + \frac{\sigma^2 \gamma^2}{Nb}(1 + 4\omega_{\mathrm{up}}).
\end{aligned}
$$

As $\gamma \leq \dfrac{1}{2L\left(1 + \frac{\omega_{\mathrm{up}}}{N}\right)}$, and thus $\left(1 - \frac{\gamma L}{2}\left(1 + \frac{4\omega_{\mathrm{up}}}{N}\right)\right) \geq 1/2$; this allows to simplify the coefficient of the scalar product:

$$
\begin{aligned}
\mathbb{E}\left[\left\| w_k - w_* \right\|^2 \;\Big|\; w_{k-1}\right] \leq&\ \left\| w_{k-1} - w_* \right\|^2 - \gamma \mathbb{E}\left[\left\langle \nabla F(\widehat{w}_{k-1}),\ \widehat{w}_{k-1} - w_* \right\rangle \mid w_{k-1}\right] \\
&+ 2\gamma L \mathbb{E}\left[\left\| \widehat{w}_{k-1} - w_{k-1} \right\|^2 \;\Big|\; w_{k-1}\right] \\
&+ 2\omega_{\mathrm{up}}\gamma^2 \mathbb{E}\left[\Xi_{k-1} \mid \widehat{w}_{k-1}\right] + \frac{\sigma^2 \gamma^2}{Nb}(1 + 4\omega_{\mathrm{up}}).
\end{aligned}
\tag{S26}
$$

With Lemma S8, we have :

$$
\begin{aligned}
\mathbb{E}\left[\Xi_k \mid w_{k-1}\right] \leq&\ (1 - \alpha_{\mathrm{up}})\mathbb{E}\left[\Xi_{k-1} \mid w_{k-1}\right] \\
&+ \frac{2\alpha_{\mathrm{up}}L}{N}\mathbb{E}\left[\left\langle \nabla F(\widehat{w}_{k-1}),\ \widehat{w}_{k-1} - w_* \right\rangle \mid w_{k-1}\right] \\
&+ \frac{2}{N}\frac{\sigma^2}{b}\alpha_{\mathrm{up}}.
\end{aligned}
\tag{S27}
$$

and Theorem S14 gives:

$$
\begin{aligned}
\mathbb{E}\left[\Upsilon_k \mid w_{k-1}\right] \leq&\ \left(1 - \frac{\alpha_{\mathrm{dwn}}}{2}\right)\Upsilon_{k-1} \\
&+ 2\gamma^2 L \left(\frac{1}{\alpha_{\mathrm{dwn}}} + \frac{4\omega_{\mathrm{up}}}{N}\right)\mathbb{E}\left[\left\langle \nabla F(\widehat{w}_{k-1}),\ \widehat{w}_{k-1} - w_* \right\rangle \mid \widehat{w}_{k-1}\right] \\
&+ 4\gamma^2 \omega_{\mathrm{up}}\mathbb{E}\left[\Xi_{k-1} \mid \widehat{w}_{k-1}\right] + \frac{2\gamma^2 \sigma^2(1 + 4\omega_{\mathrm{up}})}{Nb}.
\end{aligned}
\tag{S28}
$$

We take the full expectation (without conditioning) and we set:

$$
V_k := \mathbb{E}\left[\left\| w_k - w_* \right\|^2\right] + \gamma^2 C_1 \mathbb{E}\left[\Xi_k\right] + \gamma L C_2 \mathbb{E}\left[\Upsilon_k\right],
$$

with $C_1 = 2\omega_{\mathrm{up}}(1 + 8\gamma L \omega_{\mathrm{dwn}}/\alpha_{\mathrm{dwn}})/\alpha_{\mathrm{up}}$ and $C_2 = 4\omega_{\mathrm{dwn}}/\alpha_{\mathrm{dwn}}$.

We combine previous equations as follows $(S26) + \gamma^2 C_1 (S27) + C_2 (S28)$:

$$\mathbb{E}\left[\|w_k - w_*\|^2\right] + \gamma^2 C_1 \mathbb{E}\left[\Xi_k\right] + \gamma L C_2 \mathbb{E}\left[\Upsilon_k\right] \leq \|w_{k-1} - w_*\|^2$$
$$- \gamma \left(1 - \gamma L \left(\left(\frac{1}{\alpha_{\text{dwn}}} + \frac{4\omega_{\text{up}}}{N}\right)\gamma L C_2 + \frac{\alpha_{\text{up}} C_1}{N}\right)\right) \mathbb{E}\left[\langle\, \nabla F(\widehat{w}_{k-1}),\ \widehat{w}_{k-1} - w_* \,\rangle\right]$$
$$+ \left(2\omega_{\text{up}}(1 + 2\gamma L C_2) + (1 - \alpha_{\text{up}})C_1\right)\gamma^2 \mathbb{E}\left[\Xi_{k-1}\right]$$
$$+ \left(2\gamma L \omega_{\text{dwn}} + \left(1 - \frac{\alpha_{\text{dwn}}}{2}\right)\gamma L C_2\right) \mathbb{E}\left[\Upsilon_{k-1}\right]$$
$$+ \frac{\gamma^2 \sigma^2}{Nb}\left((1 + 4\omega_{\text{up}})(1 + 2\gamma L C_2) + 2\alpha_{\text{up}} C_1\right),$$

$$(S29)$$

We first observe that:

$$2\gamma L \omega_{\text{dwn}} + \left(1 - \frac{\alpha_{\text{dwn}}}{2}\right)\gamma L C_2 \leq \gamma L C_2 \iff C_2 \geq \frac{4\omega_{\text{dwn}}}{\alpha_{\text{dwn}}}, \quad \text{which is true by definition of } C_2.$$

Secondly, ensuring that the factor multiplying $\mathbb{E}\left[\Xi_{k-1}\right]$ on the right hand side is smaller than $\gamma^2 C_1$ requires:

$$2\omega_{\text{up}}(1 + 2\gamma L C_2) + (1 - \alpha_{\text{up}})C_1 \leq C_1$$
$$\implies \quad C_1 \geq \frac{2\omega_{\text{up}}(1 + 8\gamma L \omega_{\text{dwn}}/\alpha_{\text{dwn}})}{\alpha_{\text{up}}} \quad \text{because } C_2 = 4\omega_{\text{dwn}}/\alpha_{\text{dwn}}.$$

Finally, we have that $1 - \gamma L \left(\left(\frac{1}{\alpha_{\text{dwn}}} + \frac{4\omega_{\text{up}}}{N}\right)\gamma L C_2 + \frac{\alpha_{\text{up}} C_1}{N}\right) \geq \frac{1}{2}$, if we take $\gamma$ such that:

$$\begin{cases} \left(\frac{1}{\alpha_{\text{dwn}}} + \frac{4\omega_{\text{up}}}{N}\right)(\gamma L)^2 C_2 \leq 1/4 \implies \gamma \leq \dfrac{1}{4L\sqrt{\dfrac{\omega_{\text{dwn}}}{\alpha_{\text{dwn}}}\left(\dfrac{1}{\alpha_{\text{dwn}}} + \dfrac{4\omega_{\text{up}}}{N}\right)}} \\[4mm] \dfrac{\gamma L \alpha_{\text{up}} C_1}{N} \leq 1/4 \iff \dfrac{2\gamma L \omega_{\text{up}}}{N}\left(1 + 8\gamma L \omega_{\text{dwn}}/\alpha_{\text{dwn}}\right) \leq 1/4 \end{cases}$$

We rewrite the second condition as follows:

$$\begin{cases} 16(\gamma L)^2 \dfrac{\omega_{\text{up}}\omega_{\text{dwn}}}{\alpha_{\text{dwn}} N} \leq 1/8 \iff \gamma \leq \dfrac{1}{8L\sqrt{2\dfrac{\omega_{\text{dwn}}}{\alpha_{\text{dwn}}}\cdot\dfrac{\omega_{\text{up}}}{N}}} \\[4mm] \dfrac{2\gamma L \omega_{\text{up}}}{N\alpha_{\text{up}}} \leq 1/8 \iff \gamma \leq \dfrac{N}{16L\omega_{\text{up}}}. \end{cases}$$

Applying convexity, we derive:

$$V_k \leq V_{k-1} - \gamma \mathbb{E}\left[F(\widehat{w}_{k-1}) - F(w_*)\right] + \frac{\gamma^2 \sigma^2 \Phi^{\text{Heterog}}(\gamma)}{Nb},$$

with $\Phi^{\text{Heterog}}(\gamma) = (1 + 8\omega_{\text{up}})\left(1 + \frac{8\gamma L \omega_{\text{dwn}}}{\alpha_{\text{dwn}}}\right)$. Invoking Jensen inequality (S7) leads to $\mathbb{E}\left[F(\widehat{w}_{k-1})\right] \geq \mathbb{E}\left[F(w_{k-1})\right]$, and we finally obtain:

$$V_k \leq V_{k-1} - \gamma \mathbb{E}\left[F(w_{k-1}) - F(w_*)\right] + \frac{\gamma^2 \sigma^2 \Phi^{\text{Heterog}}(\gamma)}{Nb}.$$

$$\square$$

### G.2.3 Strongly-convex case

> **Theorem S16** (Convergence of MCM in the heterogeneous and strongly-convex case). *Under Assumptions 1 to 4 with $\mu = 0$ (convex case), for learning rates $\alpha_{\mathrm{dwn}} \leq \frac{1}{8\omega_{\mathrm{dwn}}}$ and $\alpha_{\mathrm{up}}(1 + \omega_{\mathrm{up}}) \leq 1$, for any sequence $(\gamma_k)_{k \in \mathbb{N}} \leq \gamma_{\max}^{\mathrm{Heterog}}$, for any $k$ in $\mathbb{N}$, defining:*
>
> $$V_k := \mathbb{E}\left[\|w_k - w_*\|^2\right] + \gamma_k^2 C_1 \mathbb{E}\left[\Xi_k\right] + \gamma_k L C_2 \mathbb{E}\left[\Upsilon_k\right],$$
>
> *with $C_1 = 2\omega_{\mathrm{up}}(1 + 8\gamma L\omega_{\mathrm{dwn}}/\alpha_{\mathrm{dwn}})/\alpha_{\mathrm{up}}$, $C_2 = 4\gamma L\omega_{\mathrm{dwn}}/\alpha_{\mathrm{dwn}}$, we have:*
>
> $$V_k \leq (1 - \gamma_k\mu)V_{k-1} - \gamma\mathbb{E}\left[F(\widehat{w}_{k-1}) - F(w_*)\right] + \frac{\gamma^2\sigma^2\Phi^{\mathrm{Heterog}}(\gamma)}{Nb}.$$

*Proof.* Let $k$ in $\mathbb{N}^*$, the proof starts like the one for MCM in the convex case with heterogeneous worker, and we start from eq. (S26) but we consider a variable step size $\gamma_k = 2/(\mu(k+1) + \widetilde{L})$ that depends of the iteration $k$ in $\mathbb{N}$.

We consider this following Lyapunov function:

$$V_k = \mathbb{E}\left[\|w_k - w_*\|^2\right] + \gamma_k^2 C_1 \mathbb{E}\left[\Xi_k\right] + \gamma_k L C_2 \mathbb{E}\left[\Upsilon_k\right],$$

with $C_1 = 2\omega_{\mathrm{up}}(1 + 8\gamma_k L\omega_{\mathrm{dwn}}/\alpha_{\mathrm{dwn}})/\alpha_{\mathrm{up}}$ and $C_2 = 4\omega_{\mathrm{dwn}}/\alpha_{\mathrm{dwn}}$.

$$
\begin{aligned}
\mathbb{E}\left[\|w_k - w_*\|^2\right] &+ \gamma_k^2 C_1 \mathbb{E}\left[\Xi_k\right] + \gamma_k L C_2 \mathbb{E}\left[\Upsilon_k\right] \leq \|w_{k-1} - w_*\|^2 \\
&- \gamma_k\left(1 - \gamma_k L\left(\left(\frac{1}{\alpha_{\mathrm{dwn}}} + \frac{4\omega_{\mathrm{up}}}{N}\right)\gamma_k L C_2 + \frac{\alpha_{\mathrm{up}}C_1}{N}\right)\right)\mathbb{E}\left[\langle\,\nabla F(\widehat{w}_{k-1}),\, \widehat{w}_{k-1} - w_*\,\rangle\right] \\
&+ \left(2\omega_{\mathrm{up}}(1 + 2\gamma_k L C_2) + (1 - \alpha_{\mathrm{up}})C_1\right)\gamma_k^2\mathbb{E}\left[\Xi_{k-1}\right] \\
&+ \left(2\gamma_k L\omega_{\mathrm{dwn}} + \left(1 - \frac{\alpha_{\mathrm{dwn}}}{2}\right)\gamma_k L C_2\right)\mathbb{E}\left[\Upsilon_{k-1}\right] \\
&+ \frac{\gamma_k^2\sigma^2}{Nb}\left((1 + 4\omega_{\mathrm{up}})(1 + 2\gamma_k L C_2) + 2\alpha_{\mathrm{up}}C_1\right),
\end{aligned}
$$

To ensure a $(1 - \gamma\mu)$-convergence we first choose $\left(1 - \frac{\alpha_{\mathrm{dwn}}}{2} + \frac{2\omega_{\mathrm{dwn}}}{C_2}\right)\gamma_k L C_2 \leq (1 - \gamma_k\mu)\gamma_{k-1}L C_2$ i.e $1 - \frac{\alpha_{\mathrm{dwn}}}{2} + \frac{2\omega_{\mathrm{dwn}}}{C_2} \leq \frac{(1 - \gamma_k\mu)\gamma_{k-1}}{\gamma_k}$.

We need that for all $k \in \mathbb{N}$, $\frac{1 - \gamma_k\mu}{\gamma_k} \leq \frac{1}{\gamma_{k-1}}$ i.e., $1 - \gamma_k\mu \leq \frac{\gamma_k}{\gamma_{k-1}}$, but:

$$\frac{\gamma_k}{\gamma_{k-1}} = \frac{\mu k - \mu + \widetilde{L}}{\mu k + \widetilde{L}} = 1 - \frac{\mu}{\mu k + \widetilde{L}} \quad\text{and}\quad 1 - \gamma_k\mu = 1 - \frac{2\mu}{\mu k + \widetilde{L}},$$

and so, the inequality is always true.

Thus we must have $2\omega_{\mathrm{dwn}}/C_2 \leq \alpha_{\mathrm{dwn}}/2$ which is true by definition of $C_2$.

Secondly, we need:

$$
\begin{aligned}
&\left(2\omega_{\mathrm{up}}(1 + 2\gamma_k L C_2) + (1 - \alpha_{\mathrm{up}})C_1\right)\gamma_k^2 \leq (1 - \gamma_k\mu)\gamma_{k-1}^2 C_1 \\
\iff\quad &2\omega_{\mathrm{up}}(1 + 2\gamma_k L C_2) + (1 - \alpha_{\mathrm{up}})C_1 \leq \frac{\gamma_{k-1}}{\gamma_k}C_1 \quad\text{because } \frac{1 - \gamma_k\mu}{\gamma_k} \leq \frac{1}{\gamma_{k-1}},
\end{aligned}
$$

because $\gamma_k/\gamma_k \leq \gamma_{k-1}/\gamma_k$, it is true if we verify the following stronger condition:

$$2\omega_{\mathrm{up}}(1 + 2\gamma_k L C_2) + (1 - \alpha_{\mathrm{up}})C_1 \leq \frac{\gamma_k}{\gamma_k}C_1$$

$$C_1 \geq \frac{2\omega_{\mathrm{up}}\left(1 + 8\gamma_k L\omega_{\mathrm{dwn}}\right)/\alpha_{\mathrm{dwn}}}{\alpha_{\mathrm{up}}} \quad\text{because } C_2 = 4\omega_{\mathrm{dwn}}/\alpha_{\mathrm{dwn}}.$$

Finally, in order to apply convexity we must verify: $1 - \gamma L \left( \left( \frac{1}{\alpha_{\mathrm{dwn}}} + \frac{4\omega_{\mathrm{up}}}{N} \right) C_2 + \frac{\alpha_{\mathrm{up}} C_1}{N} \right) \geq \frac{1}{2}$.

We take $\gamma_k$ such that:

$$
\begin{cases}
\left( \frac{1}{\alpha_{\mathrm{dwn}}} + \frac{4\omega_{\mathrm{up}}}{N} \right) \gamma_k L C_2 \leq 1/4 \implies \gamma_k \leq \dfrac{1}{4L\sqrt{\dfrac{\omega_{\mathrm{dwn}}}{\alpha_{\mathrm{dwn}}} \left( \dfrac{1}{\alpha_{\mathrm{dwn}}} + \dfrac{4\omega_{\mathrm{up}}}{N} \right)}} \\[4ex]
\dfrac{\gamma L \alpha_{\mathrm{up}} C_1}{N} \leq 1/4 \iff \dfrac{2\gamma_k L \omega_{\mathrm{up}}}{N} \left( 1 + 8\gamma_k L \omega_{\mathrm{dwn}}/\alpha_{\mathrm{dwn}} \right) \leq 1/4
\end{cases}
$$

We rewrite the second condition as following:

$$
\begin{cases}
16(\gamma_k L)^2 \dfrac{\omega_{\mathrm{dwn}}}{\alpha_{\mathrm{dwn}} N} \leq 1/8 \iff \gamma_k \leq \dfrac{1}{8L\sqrt{2\dfrac{\omega_{\mathrm{dwn}}}{\alpha_{\mathrm{dwn}}} \cdot \dfrac{\omega_{\mathrm{up}}}{N}}} \\[4ex]
\dfrac{2\gamma_k L \omega_{\mathrm{up}}}{N} \leq 1/8 \iff \gamma_k \leq \dfrac{1}{16L\dfrac{\omega_{\mathrm{up}}}{N}} \cdot
\end{cases}
$$

Now, we can apply strong-convexity:

$$
V_k \leq (1 - \gamma_k \mu) V_{k-1} - \gamma_k \mathbb{E}\left[ F(\widehat{w}_{k-1}) - F(w_*) \right] + \frac{\gamma_k^2 \sigma^2 \Phi^{\mathrm{Heterog}}(\gamma_k)}{Nb},
$$

with $\Phi^{\mathrm{Heterog}}(\gamma) = (1 + 8\omega_{\mathrm{up}}) \left( 1 + \frac{8\gamma_k L \omega_{\mathrm{dwn}}}{\alpha_{\mathrm{dwn}}} \right)$.

Invoking Jensen inequality (S7) leads to $\mathbb{E}\left[ F(\widehat{w}_{k-1}) \right] \geq \mathbb{E}\left[ F(w_{k-1}) \right]$, we finally obtain:

$$
V_k \leq (1 - \gamma_k \mu) V_{k-1} - \gamma_k \mathbb{E}\left[ F(w_{k-1}) - F(w_*) \right] + \frac{\gamma_k^2 \sigma^2 \Phi^{\mathrm{Heterog}}(\gamma_k)}{Nb}.
$$

$\square$