# OpenReview forum: "Preserved central model for faster bidirectional compression in distributed settings"
_NeurIPS.cc/2021/Conference — NeurIPS 2021 Poster_

### Official Review · Reviewer_TFWi · 2021-07-14

**Rating:** 6
**Confidence:** 4

**Summary:**

This paper proposed a new method: MCM for bidirectional compression in federated learning. Convergence analysis showed that linear convergence can be achieved with the proposed method. Empirical results also show the method could achieve non-trivial performance with high compression rate.

**Limitations And Societal Impact:**

Limitations and suggestions are listed in the main review.

**Main Review:**

I'm a bit concerned with the novelty of this work. Compression with memory method has been previously proposed, but only in the uplink case. This paper seems simply applying it during downlink and analyze the convergence behavior. Hence, I wonder that the contribution is a bit incremental here.

In terms of the experiment section, the compression operator being used is specifically s-quantization. However, there are other naive unbiased compression methods like random masking. To show the generality of this method, multiple compression operators for C_up and C_down should also be presented in the main experiment section. Other than that, biased compression methods like top-k masking without using MCM are also interesting to compare with.

In practice, people would also care about generalizability as well as convergence. Hence, I suggest adding test metrics(loss/accuracy) for cifar and mnist to compare the performance of MCM with baselines.

**Time Spent Reviewing:**

2.5hr

---

> ### Author Response · Authors · 2021-08-10
> **Answer 5**
>
>
> We thanks the reviewer for his time and comments. We provide detailed answers to the two main concerns, *novelty* and *experiments*, hereafter:
>
> ### Novelty.
> We would also like to summarize the novel insights brought by our paper:
>
> * This is the first algorithm for double compression to focus on a preserved central model. Our approach is also the first to demonstrate, both theoretically and in practice, the same asymptotic convergence rate for simple and double compression, which is a major improvement.
> * We use a memory process (and _Not Error Feedback,_ which has not been shown to be useful in theory for unbiased compressors or double compression [DORE], and does not bring any substantial difference numerically) on the downlink communication. While the idea of memory is not new, we highlight that:
>     - (1) **Different goal.** In previous work, memory was introduced to tackle heterogeneity, while we use it here to avoid divergence of the local model’s variance.
>     - (2) **Different impact on the convergence** In the uplink direction, memory helps to handle heterogeneity, but *does not prevent the increase of the limit variance* by a factor proportional to  the compression constant $\omega_{up}$ (thus the  convergence rate is still degraded, with or without memory). On the other hand, we show that on the downlink direction, memory _improves convergence speed_: the variance resulting from downlink compression (thus constant $\omega_{dwn}$ only appears on lower order terms, with an extra 1/T factor).
> * Our approach is one of the first to allow for worker dependent model, to theoretically quantify the impact of using worker dependent compressions on the downlink direction, and to naturally adapt to worker dependent compression levels. Also, our approach is also one of the first to result in a practical solution for PP with double compression.
>
> **Overall, we believe we tackle a problem of major importance in practice and that we offer an elegant and intuitive solution, that also opens many new directions.**
>
> ### Experiments.
>
> **Other compressors.** We can easily add other *unbiased*-compression operators, and in particular the unbiased Rand-k, or top_k with induced debiasing, etc. We generated corresponding plots, that result in similar behavior of all algorithms. We will add a selection of those results to the final version.
> Remark that for biased compressors, the theoretical part would no be valid, and results are expected to be worse. Indeed, it is fundamental to our approach that the perturbed iterate $\hat \omega_k$ is centered around $\omega_k$, i.e. $\mathbb E [\hat \omega_k]=\omega_k$ .
>
> **Test accuracy and more experiments.** As mentioned in the general comment following the paper, we performed supplementary experiments on EMNIST (heterogeneous setting) and Fashion MNIST. We report accuracy and loss figure below.
>
> Note that as in the paper, those experiments are performed **without any tuning of the algorithms**, (e.g., with the **same** learning rate for all algorithms and without ever reducing it).
>     **The goal is *not* to reach the baseline accuracy for MCM, but to show that our method is able to achieve a performance close to the one of single compression framework (DIANA), while performing a huge downlink compression  (16x and more, depending on the model size) (potentially worker dependent).**
>
> |Nonconvex framework | MNIST (CNN, d=2e4, 2 bits-quantization with norm 2) | Fashion MNIST (FashionSimpleNet, d=4e5, 2 bits-quantization with norm 2) | Heterogeneous EMNIST (CNN, d=2e4, 2 bits-quantization with norm 2) | CIFAR-10 (LeNet, d=62e3, 2-bits-quantization with norm inf) |
> |---| ---:| ---:| ---:| ---:|
> | Baseline accuracy for the selected network [Ref] |  | 92.3% [Link2]|| 67.52% [Link1] |
> | Accuracy after 300 epochs   | SGD: 99.0%| SGD: 92.4% | SGD: 99.0%|SGD: 69.1%    |
> |                             | Diana: 98.9% | Diana: 92.4%| Diana: 98.9%|Diana: 64.0%  |
> |                             | MCM: 98.8%| MCM: 90.6%| MCM: 98.9%|MCM: 63.5%    |
> |                             | Artemis: 97.9%| Artemis: 86.7%| Artemis: 98.3%|Artemis: 54.8%|
> |                             | Dore: 97.9% | Dore: 87.9%| Dore: 98.5% |Dore: 56.3%   |
> | Train loss after 300 epochs | SGD:0.025| SGD: 0.093| SGD: 0.026|SGD: 0.909    |
> |                             | Diana: 0.034| Diana: 0.141| Diana: 0.031|Diana: 1.047  |
> |                             | MCM: 0.033| MCM: 0.209| MCM: 0.030|MCM: 1.096   |
> |                             | Artemis:0.075| Artemis: 0.332| Artemis:  0.052|Artemis: 1.342|
> |                             | Dore: 0.072| Dore: 0.300| Dore: 0.048|Dore: 1.292   |
> |_________________ |_________________ |_________________ |_________________ |_________________ |_________________ |
>
> We hope these elements alleviate the reviewer's concerns and that he will consider updating his evaluation. We would be happy to provide more elements if there remain any unresolved questions.

---

> > ### Comment · Reviewer_TFWi · 2021-08-30
> > **Re: response**
> >
> > Thanks for the response. The contributions are now clear to me. The new results comparing with Artemis and Dore also look solid to me. Thus I'm willing to change my score to 6.

---

### Official Review · Reviewer_T3QV · 2021-07-16

**Rating:** 6
**Confidence:** 2

**Summary:**

In this paper, the authors propose MCM, a new algorithm achieves bidirectional compression with better convergence rate. Theoretical analysis is provided. The empirical results show good performance.


**Limitations And Societal Impact:**

The experiments are still not large enough. In distributed training, we usually focus on large-scale problems like imagenet. Cifar100 or some NLP tasks are also recommended. However, since I believe that this paper focuses mostly on the theoretical analysis, I don’t think this is a big issue.

**Main Review:**

The paper is well-written. The theorems give detailed analysis in strongly convex, convex, and non-convex cases. The experimental results show that MCM can achieve better convergence when using the same communication overhead as the baselines.
Also, I like the idea of calculating the carbon footprint in the appendix.

**Time Spent Reviewing:**

4

---

> ### Author Response · Authors · 2021-08-10
> **Answer 4**
>
>
> Thank you very much for your time and your positive remarks. We agree on the point that our experiments could be improved, however, as pointed out, our paper aims to provide theory and we share your opinion that experiments in large-scale problems are not a mandatory. We added FashionMnist, Heterogeneous Emnist, and also we will report accuracies.
>
> We provided the corresponding results in the general comment to all reviewers above.

---

> > ### Comment · Reviewer_T3QV · 2021-08-31
> > **response to the authors' feedback**
> >
> > I've read the additional experiment results and I'm satisfied with the authors' feedback. Thus, I'm going to keep the positive review and recommend to accept.

---

### Official Review · Reviewer_NENJ · 2021-07-23

**Rating:** 6
**Confidence:** 4

**Summary:**

This paper proposed to compress model different for improving downlink compression. The theoretical analysis on convex problems shows that the dominant term does not depend on the compression error of the downlink, thus improving upon on the existing results on two-way compression.

**Limitations And Societal Impact:**

Yes

**Main Review:**

Overall, the idea is interesting and the paper is well written. However, I have a few concerns:

1. One motivation of the paper is that the downlink communication is cheaper than the uplink one. This depends on the actual network topology. If we consider a mesh topology, recursive doubling allows us to perform broadcasting faster.  But if we consider a general server-worker bipartite graph (Jiang, Yimin, et al., 2020), the uplink and downlink communication can be overlapped and the downlink one is not more efficient than the uplink one.

2. The theories of  Zheng et al. [41] and Tang et al. [36] are for nonconvex problems. What is the convergence result of the proposed method for nonconvex objective? Can we still remove \omega_{dwn} from the dominant term? And, Corollary 2 in Tang et al. [36] also shows that \omega_{up/dwn} is in a higher order term 1/T^{2/3}.

3. Bound (5) has a quadratic dependency on \omega_{dwn}, which is worse than the bound for two-way compression.

4. Given a fixed communication budget, how do we distribute it over uplink and downlink to optimize the convergence rate?

5. How does the distribution of the difference of gradient differ from the one of model? And, how does this difference affect the choice of the uplink and downlink compressors?

6. What does d=20 mean in Figure 2?

7. The experimental setup is weak, i.e., synthetic dataset and small image classification problems.

Jiang, Yimin, et al. "A Unified Architecture for Accelerating Distributed {DNN} Training in Heterogeneous GPU/CPU Clusters." OSDI. 2020.

**Time Spent Reviewing:**

1

---

> ### Author Response · Authors · 2021-08-10
> **Answer 3**
>
>
> Thank you for your time reviewing our paper.
>
> #### 1. Impact of downlink.
> Thank you for providing the reference to (Jiang, Yimin, et al., 2020). Actually, this is precisely our motivation to highlight that the impact of compressing the downlink message can be of major importance.
> It is because some readers are mostly used to uni-directional compression that we recalled that in some particular cases, broadcasting is cheaper. On the contrary, *we are focusing on the numerous cases in which downlink cannot be ignored*.
>
> Unfortunately, the very large majority of the literature is focused on uni-directional compression. This is why we believe **our approach can have major consequences, by providing a novel way to perform double compression without strongly degrading convergence.**
>
> #### 2.a. Behavior in the non-convex case.
> We provide a convergence theorem in Theorem S11 in the Appendix (page 37). This result was placed in the appendix for space constraints. **The behavior is similar**: for a step size decaying as $1/\sqrt{K}$, the $\omega_{dwn}$ constant only appears in lower order terms, in a term scaling as $O(K^{-1})$.
>
>
> #### 2.b Comparison to Zheng et al. [41] and Tang et al. [36]
> -  Zheng et al. [41] consider a $(1-\delta)$ contractive biased operator [Def 1 in 41], and the same compression level for uplink and downlink compression, which makes the comparison impossible. They also rely on much stronger assumptions on the gradient (uniformly bounded, in Assumption 3), and only tackle the homogeneous case.
> - Tang et al. [36] make a completely different assumption on the compression operator, namely that it has *bounded magnitude of error* (assumption 1.3). This explains why the convergence bound in their Theorem 1 and Corollary 2 does not have any increase on the dominating variance term $\varpropto \sigma^2$ by factors involving $\omega_{up}, \omega_{dwn}$. They also only tackle the homogeneous case.
>
> #### 3. Quadratic dependence on $\omega$.
> Bound (5) has indeed a quadratic dependence on $\omega_{dwn}$, but the corresponding term is **divided by an extra factor $K$**, the number of iterations.
> In practice, $\omega_{dwn}$ is typically of the order of 10 to 40 at most, while $K$ is typically at least of the order of $10^5$. Hence, this term is vanishing through iterations. This is why we asymptotically recover a rate of convergence equivalent to algorithms using uni-compression.
>
> On the other hand, in classical double compression, the product of compression constants $\omega_{up}\times \omega_{dwn}$ appears in the *dominating term.*
>
> #### 4. How to split the communication budget?
> The distribution of a given communication budget over uplink and downlink to optimize the convergence is an excellent and open question which is intrinsically related to the situation. Indeed it depends on the selected operators of compression, the upload/downlink speed, the number of participating workers at each iteration, etc.
>
> Yet, our approach provides insights on that questions. As *asymptotically*, the impact of double compression is marginal, for a fixed budget, this suggests to strongly compress on the downlink direction (large $\omega_{dwn}$), but to perform a weaker compression in the uplink direction (small $\omega_{up}$).
>
> #### 5. Distribution of the difference of gradient vs distribution of the model difference.
>
> This is a very interesting question and an open direction. Most analysis (ours included), only control the second order moments (of the gradients difference for uplink, and of model-memory difference for downlink). We agree that a more in-depth analysis would be insightful, as the distribution could influence the behavior of the compressors (for example if the difference remains sparse, it could be beneficial to use a sparsification technique).
> However, a thorough analysis of those complicated aspects is out of the scope of the paper.
>
> #### 6. For figure 2a, 2b.
> Notation $d$ corresponds to the number of features, hence $20$ in this case. This is meant to be a small scale experiment, to illustrate the behavior of all algorithms in terms of Bias and Variance, a point that was appreciated by Reviewer VLMx.
>
> #### 7. Complementary experiments.
> Concerning the experiments in the convex case, in addition to superconduct and quantum, we have added several other classical dataset: a9a, abalone, covtype, madelon, phishing, w8a. We have also added FashionMnist and FeMnist, and will add a larger model on CIFAR.
>
> |Nonconvex framework | MNIST (CNN, d=2e4, 2 bits-quantization with norm 2) | Fashion MNIST (FashionSimpleNet, d=4e5, 2 bits-quantization with norm 2) | Heterogeneous EMNIST (CNN, d=2e4, 2 bits-quantization with norm 2) | CIFAR-10 (LeNet, d=62e3, 2-bits-quantization with norm inf) |
> |---| ---:| ---:| ---:| ---:|
> | Baseline accuracy for the selected network [Ref] |  | 92.3% [Link2]|| 67.52% [Link1] |
> | Accuracy after 300 epochs   | SGD: 99.0%| SGD: 92.4% | SGD: 99.0%|SGD: 69.1%    |
> |                             | Diana: 98.9% | Diana: 92.4%| Diana: 98.9%|Diana: 64.0%  |
> |                             | MCM: 98.8%| MCM: 90.6%| MCM: 98.9%|MCM: 63.5%    |
> |                             | Artemis: 97.9%| Artemis: 86.7%| Artemis: 98.3%|Artemis: 54.8%|
> |                             | Dore: 97.9% | Dore: 87.9%| Dore: 98.5% |Dore: 56.3%   |
> | Train loss after 300 epochs | SGD:0.025| SGD: 0.093| SGD: 0.026|SGD: 0.909    |
> |                             | Diana: 0.034| Diana: 0.141| Diana: 0.031|Diana: 1.047  |
> |                             | MCM: 0.033| MCM: 0.209| MCM: 0.030|MCM: 1.096   |
> |                             | Artemis:0.075| Artemis: 0.332| Artemis:  0.052|Artemis: 1.342|
> |                             | Dore: 0.072| Dore: 0.300| Dore: 0.048|Dore: 1.292   |
> |_________________ |_________________ |_________________ |_________________ |_________________ |_________________ |
>
> Note that as in the paper, those experiments are performed **without any tuning of the algorithms**, (e.g., with the **same** learning rate for all algorithms and without ever reducing it).
>     **The goal is *not* to reach the baseline accuracy for MCM, but to show that our method is able to achieve a performance close to the one of single compression framework (DIANA), while performing a huge downlink compression  (16x and more, depending on the model size) (potentially worker dependent).**
>
>
> We thank the reviewer for his support and questions. We hope our answers clarified the corresponding points. We would be happy to provide more elements if there remain any unresolved questions.

---

### Official Review · Reviewer_VLMx · 2021-08-03

**Rating:** 6
**Confidence:** 4

**Summary:**

The paper proposes a bidirectional compression algorithm for parameter-server-based distributed training.
Two variants, MCM and Rand-MCM, are proposed to achieve the same improvement as the uplink-compression-only compression algorithm, where the downlink-compression-only impacts local models while the global model is preserved.
The Rand-MCM also introduces the concept of (simulated) random smoothing via model compression.

**Limitations And Societal Impact:**

Please check the main review.

**Main Review:**

### Advantages
* The paper has been written well and is theoretically sound.
* The use case for introducing downlink and uplink compression is critical and could have significant consequences.
* The concept of using different compressed worker models to perform random smoothing is intriguing.


### Disadvantages
1. The paper's convergence analysis results were not fully validated (through some numerical results).
    1.  The memory mechanism is crucial for controlling the variance of the local model, but I cannot find a detailed (empirical) discussion related to $\alpha_{\text{dwn}}$. Figure 1 is insufficient to justify the trade-off of the different hyper-parameter choices; some numerical results are required here to justify the tightness of the convergence rate.
    2. Similar issues appeared in the $\omega_{\text{up}}$. Hyper-parameters associated with different levels of $\omega_{\text{up}}$ should be included, and studied, for example, in terms of complexity.
    3. It is a good idea to use a constant learning rate to observe the bias and variance independently. However, the current results in Figures 2 & 3 are hard to tell the difference, e.g., (1) it is difficult to determine whether all curves enter the saturation level or not due to the log-scale visualization, (2) why the Rand-MCM results are omitted in Figure 2c & d, and (3) for MCM results in Figures 2d & 3d, the reached training loss values are far behind other baselines and I am not sure if prolonging the training period can close this gap or not. It is necessary to provide a more detailed illustration.
    4. The numerical evaluations only take into account a fixed number of workers and a fixed level of compression/quantization. Because of the limited experimental setups, understanding the empirical behavior of the proposed algorithms, as well as the alignment between numerical results and convergence analysis, is difficult.
2. The concept of using the perturbed model in distributed training to perform random smoothing is not novel; the authors should at the very least include a brief discussion with [1, 2, 3, 4]. It would be great if the authors could use the insights of improved gradient Lipschitz in [5] to further justify the gain of their proposed Rand-MCM algorithm.

### Reference
1. SmoothOut: Smoothing Out Sharp Minima to Improve Generalization in Deep Learning, https://arxiv.org/abs/1805.07898
2. Gradient Noise Convolution (GNC): Smoothing Loss Function for Distributed Large-Batch SGD, https://arxiv.org/abs/1906.10822
3. Extrapolation for Large-batch Training in Deep Learning, https://arxiv.org/abs/2006.05720, ICML 2020
4. Sharpness-Aware Minimization for Efficiently Improving Generalization, https://arxiv.org/abs/2010.01412, ICLR 2021
5. Randomized smoothing for stochastic optimization, https://arxiv.org/abs/1103.4296, SIAM Journal on Optimization.


=== review update ===
I am happy with the responses and I will raise my score to 6.



**Time Spent Reviewing:**

4

---

> ### Author Response · Authors · 2021-08-10
> **Answer 2**
>
>
>
> We thank the reviewer for his/her time and constructive review, and for the multiple questions raised. Thank you also for acknowledging the quality and soundness of the paper as well as its potential consequences.
> Hereafter, we provide a detailed answer to your main concerns.
>
> #### 1.1; 1.2 -- Hyperparameter tuning/ sensibility
> This is an interesting remark: Figure 1 shows that taking $\alpha_{dwn}= 0$ (too small) or $=1$ (too large) makes MCM diverge. The conditions on $\alpha_{dwn}$ result from the proofs given in Appendix, see Th. S8 and Th. S14. Theory suggests to use the largest possible $\alpha_{dwn}$ smaller than a value $\frac{1}{ 4 (\omega_{dwn} +1)}$. As the constant 4 is partially an artifact of the proof, in experiments we used $\alpha_{dwn} = \frac{1}{ (\omega_{dwn} +1)}$.
>
> Same remarks apply to the uplink direction. We will add some experiments regarding the evolution of the error w.r.t $\alpha_{up/dwn}$.
>
>
>
> #### 1.3 -- Number of epochs for saturation
> Indeed, more iterations would help to ensure that saturation has been reached ! We believe that our experiments are sufficient to check that SGD > DIANA $\simeq$ MCM >> DORE, ARTEMIS, which is the main message of our paper and our most important theoretical result.
>
> We can provide complementary experiments on (a9a, covtype, madelon, phishing, w8a), on which this ordering is even clearer and saturation is reached in only a few epochs. For example:
>
> | a9a, d=123 , b=50| SGD | DIANA | MCM | DORE
> |---| ---:| ---:| ---:| ---:|
> |epochs for saturation  |  300| 150 | 150 |30  |
> |loss after 450 epochs  |  -3.5 | -2.7 | -2.7 | -1.8 |
>
> | Phishing, d=68 , b=50| SGD | DIANA | MCM | DORE|
> |---| ---:| ---:| ---:| ---:|
> |epochs for saturation  | 1000  | 750 | 750 | 300 |
> |loss after 450 epochs  | -3.7 | -3.5 | -3.4 | -2.7 |
>
> | w8a, d=300 , b=8| SGD | DIANA | MCM | DORE|
> |---| ---:| ---:| ---:| ---:|
> |epochs for saturation  | 600 | 300 | 300 | 30 |
> |loss after 450 epochs  | -3.5 | -3.0 | -2.5 | -1.75|
>
> **Why the Rand-MCM results are omitted in Figure 2c & d.**
> Just to alleviate the reading of the curves. We will add them, as on other figures, it is slightly better and more stable than MCM.
>
> **For MCM results in Figures 2d & 3d.** We are unsure whether the reviewer meant fig 2.c.et 2.d or 2.d. and 3.d.
> Regarding figures in non convex case, the performance of MCM was improved by correcting the initialization (see general comment). Otherwise, on all figures, the convergence of SGD (no compression) seems to be correct, and the fact that the limit value is not optimal can depend on the step size tuning.
>
> For deep learning experiments, we will also report accuracies.
>
> |Nonconvex framework | MNIST (CNN, d=2e4, 2 bits-quantization with norm 2) | Fashion MNIST (FashionSimpleNet, d=4e5, 2 bits-quantization with norm 2) | Heterogeneous EMNIST (CNN, d=2e4, 2 bits-quantization with norm 2) | CIFAR-10 (LeNet, d=62e3, 2-bits-quantization with norm inf) |
> |---| ---:| ---:| ---:| ---:|
> | Baseline accuracy for the selected network [Ref] |  | 92.3% [Link2]|| 67.52% [Link1] |
> | Accuracy after 300 epochs   | SGD: 99.0%| SGD: 92.4% | SGD: 99.0%|SGD: 69.1%    |
> |                             | Diana: 98.9% | Diana: 92.4%| Diana: 98.9%|Diana: 64.0%  |
> |                             | MCM: 98.8%| MCM: 90.6%| MCM: 98.9%|MCM: 63.5%    |
> |                             | Artemis: 97.9%| Artemis: 86.7%| Artemis: 98.3%|Artemis: 54.8%|
> |                             | Dore: 97.9% | Dore: 87.9%| Dore: 98.5% |Dore: 56.3%   |
> | Train loss after 300 epochs | SGD:0.025| SGD: 0.093| SGD: 0.026|SGD: 0.909    |
> |                             | Diana: 0.034| Diana: 0.141| Diana: 0.031|Diana: 1.047  |
> |                             | MCM: 0.033| MCM: 0.209| MCM: 0.030|MCM: 1.096   |
> |                             | Artemis:0.075| Artemis: 0.332| Artemis:  0.052|Artemis: 1.342|
> |                             | Dore: 0.072| Dore: 0.300| Dore: 0.048|Dore: 1.292   |
> |_________________ |_________________ |_________________ |_________________ |_________________ |_________________ |
>
> Note that as in the paper, those experiments are performed **without any tuning of the algorithms**, (e.g., with the **same** learning rate for all algorithms and without ever reducing it).
>     **The goal is *not* to reach the baseline accuracy for MCM, but to show that our method is able to achieve a performance close to the one of single compression framework (DIANA), while performing a huge downlink compression  (16x and more, depending on the model size) (potentially worker dependent).**
>
> #### 1.4 -- Fixed level of compression/quantization.
> We have added experiments, using **variable quantization levels** (worker dependent), leveraging the flexibility of R-MCM.
>
> #### 2 -- Randomized smoothing
> We agree that using the perturbed model in distributed training to perform random smoothing is not novel. We included a short discussion in Remark 3 and Appendix A.2 on this topic, e.g. [Duchi12] is already referred to in this section, but we will enrich this section with other references. However, because of space constraints and because the smoothing effect is not leveraged in the paper (the quantization noise is typically "very not Gaussian", thus generally does not improve or increase the regularity of the function), we prefer to keep it in the Appendix.
>
> We thank the reviewer for his questions and our answers clarify the reviewer's concerns and that he/she will consider supporting the paper. We would be happy to provide more elements if there remain any unresolved questions.
>
> ---
> - [Duchi12] Randomized smoothing for stochastic optimization [https://arxiv.org/abs/1103.4296](https://arxiv.org/abs/1103.4296), SIAM Journal on Optimization.

---

### Official Review · Reviewer_9w9P · 2021-08-03

**Rating:** 5
**Confidence:** 4

**Summary:**

The paper proposes MCM ~ algorithm to perform bidirectional compression in distributed setting. The authors claim similar convergence guarantees as vanilla setting.
They introduce the notion of sending different models to different clients while keeping the global model preserved.


**Ethics Review Area:**

["I don’t know"]

**Limitations And Societal Impact:**

See Above

**Main Review:**

Overall, the presentation has cyclomatic complexity. Its hard to read and the main idea is not clear at a fast glance. I would strongly encourage improving the presentation and writing.

For example, using algorithmic blocks would greatly help.

Novelty:
I have concerns regarding the novelty of the algorithm. The idea of error feedback is leveraged in both client and server to deal with bidirectional compression degradation. While client EF is common, server EF is less common but not new [1]

Also, using EF at both end seems like a natural approach and not very novel or insightful .

The main contribution of the paper is thus the analysis. However, I find the analysis straightforward extensions of EF-SGD [2] proof technique.

Experiments :
The experiments are tiny. MNIST , using simple toy like datasets
I would strongly encourage performing experiments on sufficiently large models and more sophisticated datasets.

The deep learning exp in appendix show S11(a) slower convergence.

Time Complexity:
Note that computing the server memory etc has computational overhead and it would be great to run experiments wrt wall clock time.
Also, a precise asymptotic complexity analysis might be useful

[1] https://arxiv.org/pdf/2106.08882.pdf
[2] https://arxiv.org/abs/1901.09847

**Time Spent Reviewing:**

2

---

> ### Author Response · Authors · 2021-08-10
> **Answer 1**
>
> We would like to thank you for your time and your remarks.
>
> 1. **Clarity of the paper**.  We are sorry that you found the paper hard to read. Other reviewers seemed to consider the presentation to be clear. Rev VLMX: “*The paper has been written well and is theoretically sound*”,  Rev NENJ “*Overall, the idea is interesting and the paper is well written*”.  Rev T3QV “*The paper is well-written*”. We chose to give a step by step description of the algorithm, building up from naive solutions, and introducing each mechanism at once, and to provide the pseudocode as appendix.  If the paper is accepted, we will use the extra page to place these pseudo-codes in the main text.
> 2. **Link between memory and error-feedback. We first would like to highlight that there might be a misunderstanding. We do NOT use error feedback in this algorithm but a memory process**. In the context of double compression, EF has been considered in DORE, which we carefully compare to, algorithmically, theoretically and experimentally.
> A few remarks regarding the differences:
>     - **Algorithmic differences**. The memory technique and EF technique are different and have been introduced for different goals in the uplink direction. Memory was introduced to tackle heterogeneity (see [Mish19], [Phil20]), EF to recover convergence for biased compressors (see [2]=[Kar2019]).
>     - **Theoretical Results and analysis**. Horvath and Richtarik, 2020 showed that EF with *unbiased* operators can hurt performance (sect. 3) and as mentioned in Remark 2, EF does not bring improvements on DORE. Theoretical improvement with EF has been observed only for biased contractive operators, but (1) this class is restrictive; (2) convergence does not benefit from the multiplicity of workers [Gorb20, Table 1, no dependence on $n$ on the complexity for GD and Var Reduced methods] (3) Scaling approaches to transform an unbiased compressor into a contractive one degrade the learning rate (thus convergence) and lose the benefit of compressor's independence.
>     - **Experimental Results**. We systematically observe that MCM outperforms DORE.
>
> Remark that those differences are highlighted throughout the paper (esp. table 1,  remark 2,  experiments on DORE, comments on cvge on DORE).
>
> We want to underline that the main novelty of our approach is not to introduce the downlink memory process but **to not degrade the central model**. This is a completely new paradigm compared to existing algorithms performing bi-compression that allows to asymptotically recover the performance of algorithms performing uni-directional compression. In order to make our algorithm converge we then _**have to**_ introduce the downlink memory process (see discussion lines 108 or 223 and fig1). Thus, this mechanism is the mandatory following of our new approach.
>
> 3. **Novelty.** We would also like to summarize the novel insights brought by our paper, a clarification suggested by reviewers.
>    * This is the first algorithm for double compression to focus on a preserved central model. Our approach is also the first to demonstrate, both theoretically and in practice, the same asymptotic convergence rate for simple and double compression, which is a major improvement.
>    * We use a memory process (and _Not Error Feedback,_ which has not been shown to be useful in theory for unbiased compressors or double compression [DORE], and does not bring any substantial difference numerically) on the downlink communication. While the idea of memory is not new, we highlight that:
>         1. **Different goal.** In previous work, memory was introduced to tackle heterogeneity, while we use it here to avoid divergence of the local model’s variance.
>         2. **Different impact on the convergence** In the uplink direction, memory helps to handle heterogeneity, but *does not prevent the increase of the limit variance* by a factor proportional to  the compression constant $\omega_{up}$ (thus the  convergence rate is still degraded, with or without memory). On the other hand, we show that on the downlink direction, memory _improves convergence speed_: the variance resulting from downlink compression (thus constant $\omega_{dwn}$ only appears on lower order terms, with an extra 1/T factor).
>    * Our approach is one of the first to allow for worker dependent model, to theoretically quantify the impact of using worker dependent compressions on the downlink direction, and to naturally adapt to worker dependent compression levels. Also, our approach is also one of the first to result in a practical solution for PP with double compression.
>
> We will add a reference to [1], but note that it was released **after** the submission of our manuscript.
>
> 4. **Experiments.** First, our main contribution is theoretical, and the goal and spirit of our set of  experiments is primarily to support the theory and highlight the importance of each factor. We believe simple experiments are sometimes more insightful: e.g., fig 2b shows that in the interpolation regime we obtain a linear convergence, and other figures highlight bias variance tradeoffs for a constant step size.
>
> Secondly, we would like to point out that we also have experiments on CIFAR10 (on LeNet).
>
> However, we agree that more experiments can be added. We have performed complementary experiments on FashionMnist and heterogeneous EMNIST. The shape of convergence curves matches the one in the paper, but as it is not possible to upload images on openreview, we report the scores after 300 epochs below.
>
> Note that as in the paper, those experiments are performed **without any tuning of the algorithms**, (e.g., with the **same** learning rate for all algorithms and without ever reducing it).
>     **The goal is *not* to reach the baseline accuracy for MCM, but to show that our method is able to achieve a performance close to the one of single compression framework (DIANA), while performing a huge downlink compression  (16x and more, depending on the model size) (potentially worker dependent).**
>
> |Nonconvex framework | MNIST (CNN, d=2e4, 2 bits-quantization with norm 2) | Fashion MNIST (FashionSimpleNet, d=4e5, 2 bits-quantization with norm 2) | Heterogeneous EMNIST (CNN, d=2e4, 2 bits-quantization with norm 2) | CIFAR-10 (LeNet, d=62e3, 2-bits-quantization with norm inf) |
> |---| ---:| ---:| ---:| ---:|
> | Baseline accuracy for the selected network [Ref] |  | 92.3% [Link2]|| 67.52% [Link1] |
> | Accuracy after 300 epochs   | SGD: 99.0%| SGD: 92.4% | SGD: 99.0%|SGD: 69.1%    |
> |                             | Diana: 98.9% | Diana: 92.4%| Diana: 98.9%|Diana: 64.0%  |
> |                             | MCM: 98.8%| MCM: 90.6%| MCM: 98.9%|MCM: 63.5%    |
> |                             | Artemis: 97.9%| Artemis: 86.7%| Artemis: 98.3%|Artemis: 54.8%|
> |                             | Dore: 97.9% | Dore: 87.9%| Dore: 98.5% |Dore: 56.3%   |
> | Train loss after 300 epochs | SGD:0.025| SGD: 0.093| SGD: 0.026|SGD: 0.909    |
> |                             | Diana: 0.034| Diana: 0.141| Diana: 0.031|Diana: 1.047  |
> |                             | MCM: 0.033| MCM: 0.209| MCM: 0.030|MCM: 1.096   |
> |                             | Artemis:0.075| Artemis: 0.332| Artemis:  0.052|Artemis: 1.342|
> |                             | Dore: 0.072| Dore: 0.300| Dore: 0.048|Dore: 1.292   |
> |_________________ |_________________ |_________________ |_________________ |_________________ |_________________ |
>
> We can also provide experiments on CIFAR with a larger model (Resnet20 and VGG16) if the reviewer believes such experiments would add something to the paper.
>
> 5. **Slow convergence in DL.** Indeed S11(a) exhibits a slower convergence than unidirectional compression. This is due to an error in how we had set the initial value of $H_0$, which led to a very large value of $V_0$ (in eq 4, 6). With the correct setting convergence is much faster. Details on that mistake are given in the general comment to all reviewers above.
>
> Note that (even with that mistake) we still outperformed naive double compression, even with EF (Artemis, Dore).
>
> 6. **Wall clock time.** The compression of $w_k-H_{k-1}$ on the central server is typically a marginal overhead w.r.t. gradient computation (backpropagation in DL), gradient compression, and communication. Here, as we perform experiments in a **simulated environment** (there is *actually* no communication cost), we do “gain time” thanks to the compression, but we ensure that double compression does not cost time. We here report the computation time when training on Heterogeneous EMNIST.
>
>
> | | Computation Time for 150 epoch with batch size b=128 | Communication time (architecture and network dependent) |
> |---| ---:| ---:|
> |SGD | 15421s | No compression |
> |SGD | 16773s, ratio: 1.08 | Compression on uplink |
> |SGD | 16769s, ratio: 1.08 | Compression on up and downlink (>50x) |
>
> We hope this clarifies your concerns and that you will consider raising your score accordingly. We would be happy to provide more elements if there remain any unresolved questions.
>
> ---
>
> - [1], [2] from reviewer’s comment
> - [Mish19]: Distributed Learning with Compressed Gradient Differences, Konstantin Mishchenko, Eduard Gorbunov, Martin Takáč, Peter Richtárik
> - [Phil20]: Bidirectional compression in heterogeneous settings for distributed or federated learning with partial participation: tight convergence guarantees, Constantin Philippenko, Aymeric Dieuleveut
> - [Kar2019] = [2]: Error Feedback Fixes SignSGD and other Gradient Compression Schemes, Sai Praneeth Karimireddy, Quentin Rebjock, Sebastian U. Stich, Martin Jaggi
> - [Gorb2020]: Linearly Converging Error Compensated SGD, Eduard Gorbunov, Dmitry Kovalev, Dmitry Makarenko, Peter Richtárik
> - [Lin2013]: Network In Network, Min Lin, Qiang Chen, Shuicheng Yan.

---

> > ### Comment · Reviewer_9w9P · 2021-09-02
> > **Thanks for answering my concerns**
> >
> > Thanks for being meticulous about addressing the concerns of all the reviewers.
> >
> > However, the reason I am not inclined to accept this paper is the following:
> > The whole downlink compression with memory idea is not surprising and I dont feel adds any value to optimization community.
> > To be honest, doing SGD with compressed iterates [Stich Sparsified SGD with memory paper] was novel and nice; But after that re-purposing the general idea of compensating for error and achieving linear convergence for uplink, downlink etc etc for distributed , federated and all that is mostly too incremental.
> >
> > But nonetheless, I am happy to raise my score to 5.
> >
> > [Stich] https://proceedings.neurips.cc/paper/2018/file/b440509a0106086a67bc2ea9df0a1dab-Paper.pdf

---

### Author Response · Authors · 2021-08-10
**General comment**

We thank the reviewers for the time they spent on the review, their careful readings, and their encouraging comments. The paper will be revised accordingly; the current modifications also contain a series of smaller improvements to style and clarity.

We would like to first clarify a couple of points regarding experiments and the novelties brought by our approach.

### Substantial improvement in Deep Learning Experiments.
In the submitted version of the paper, for the deep learning experiments, we set by mistake the initial value of the downlink memory $H_0$ to $0$, together with a standard pytorch initialization technique for the weights $w_0$. Algortihm still converges but to minimize the value of the "initial condition" $V_0$ in the convergence guarantees (eq. (4) and (6)) and to ensure that $V_0 = ||w_0-w_*||^2)$, **we should instead set $H_0 = w_0$** (the initial model being transmitted, at the very first iteration, without compression). The mistake came from the fact that in the convex case, we use $w_0=0$, thus $H_0=0$.

This has degraded the behavior of all convergence curves in DL, esp Figs. 2c, 2d, S10, S11, as the value of $V_0$ was much higher for MCM than competing algorithms, and explained why the difference between MCM and DORE was not as strong as expected, and MCM appeared less efficient than DIANA.

### Complementary experiments and accuracy values.
Reviewers rightly pointed that more experiments could be helpful, and that accuracy figures were missing. We performed more experiments, by running the script  provided as SM on other datasets:
- In the convex setting: on a9a, covtype, madelon, phishing, w8a.
- In the non-convex setting: on FashionMNIST, FEMNIST,
We report excess loss values (for the convex setting) and test loss and accuracies (for the non convex setting) below.

As in the paper, those experiments are performed **without any tuning of the algorithms**, (e.g., with the **same** learning rate for all algorithms and without ever reducing it).
    **The goal is *not* to reach the baseline accuracy for MCM, but to show that our method is able to achieve a performance close to the one of single compression framework (DIANA), while performing a huge downlink compression  (16x and more, depending on the model size) (potentially worker dependent).**


We could improve the score of MCM by reducing the step size after a certain number of epochs.

#### A. Convex new Results

We  also provide complementary experiments on (a9a, phishing, w8a), on which the ordering is even clearer after a few epochs. For example:

|Excess loss after 450 epochs | SGD  | DIANA  | MCM  | DORE |
|---| ---:| ---: | ---:| ---:|
| a9a b=50       | -3.5 | -2.7 | -2.7 | -1.8 |
| Phishing b=50  | -3.7 | -3.5 | -3.4 | -2.7 |
| w8a b=8        | -3.5 | -3.0 | -2.5 | -1.75|
| Compression    | no   | uni-dir | bi-dir | bi-dir |


#### B. Non-Convex new results

|Nonconvex framework | MNIST (CNN, d=2e4, 2 bits-quantization with norm 2) | Fashion MNIST (FashionSimpleNet, d=4e5, 2 bits-quantization with norm 2) | Heterogeneous EMNIST (CNN, d=2e4, 2 bits-quantization with norm 2) | CIFAR-10 (LeNet, d=62e3, 2-bits-quantization with norm inf) |
|---| ---:| ---:| ---:| ---:|
| Baseline accuracy for the selected network [Ref] |  | 92.3% [Link2]|| 67.52% [Link1] |
| Accuracy after 300 epochs   | SGD: 99.0%| SGD: 92.4% | SGD: 99.0%|SGD: 69.1%    |
|                             | Diana: 98.9% | Diana: 92.4%| Diana: 98.9%|Diana: 64.0%  |
|                             | MCM: 98.8%| MCM: 90.6%| MCM: 98.9%|MCM: 63.5%    |
|                             | Artemis: 97.9%| Artemis: 86.7%| Artemis: 98.3%|Artemis: 54.8%|
|                             | Dore: 97.9% | Dore: 87.9%| Dore: 98.5% |Dore: 56.3%   |
| Train loss after 300 epochs | SGD:0.025| SGD: 0.093| SGD: 0.026|SGD: 0.909    |
|                             | Diana: 0.034| Diana: 0.141| Diana: 0.031|Diana: 1.047  |
|                             | MCM: 0.033| MCM: 0.209| MCM: 0.030|MCM: 1.096   |
|                             | Artemis:0.075| Artemis: 0.332| Artemis:  0.052|Artemis: 1.342|
|                             | Dore: 0.072| Dore: 0.300| Dore: 0.048|Dore: 1.292   |
|_________________ |_________________ |_________________ |_________________ |_________________ |_________________ |


Link1: https://github.com/icpm/pytorch-cifar10

Link2: https://github.com/kefth/fashion-mnist

If the paper is accepted, we will also provide more experiments on large scale models (CIFAR-10 + ResNet-20 and VGG16).




### Novelty.
We would also like to summarize the novel insights brought by our paper, a clarification suggested by reviewers.
   * This is the first algorithm for double compression to focus on a preserved central model. Our approach is also the first to demonstrate, both theoretically and in practice, the same asymptotic convergence rate for simple and double compression, which is a major improvement.
   * We use a memory process (and _Not Error Feedback,_ which has not been shown to be useful in theory for unbiased compressors or double compression [DORE], and does not bring any substantial difference numerically) on the downlink communication. While the idea of memory is not new, we highlight that:
        1. **Different goal.** In previous work, memory was introduced to tackle heterogeneity, while we use it here to avoid divergence of the local model’s variance.
        2. **Different impact on the convergence** In the uplink direction, memory helps to handle heterogeneity, but *does not prevent the increase of the limit variance* by a factor proportional to  the compression constant $\omega_{up}$ (thus the  convergence rate is still degraded, with or without memory). On the other hand, we show that on the downlink direction, memory _improves convergence speed_: the variance resulting from downlink compression (thus constant $\omega_{dwn}$ only appears on lower order terms, with an extra 1/T factor).
   * Our approach is one of the first to allow for worker dependent model, to theoretically quantify the impact of using worker dependent compressions on the downlink direction, and to naturally adapt to worker dependent compression levels. Also, our approach is also one of the first to result in a practical solution for PP with double compression.


We thank the reviewers again. We provide detailed answers to the reviewers concerns below and would be happy to provide more elements if there remain any unresolved questions.

---

### Decision · Program_Chairs · 2021-09-27

**Decision:**

Accept (Poster)

**Comment:**

This work proposes MCM, a method that performs bidirectional (i.e., both uplink and downlink) compression in distributed learning. A benefit of the approach is that it is able to match convergence rates of methods that use compression in only a single direction. The reviewers were generally in agreement that the work was well-written and motivated. However, many of the reviewers were unclear about the novelty of the approach, and there were a significant number of additional experiments included as part of the discussion period (e.g., relating to test metrics, new datasets). I believe the paper will be significantly strengthened if the authors carefully incorporate the discussed changes (specifically, more clearly outlining the main contributions/novelty and including the additional experiments that have been run since the time of submission). Although one reviewer finds the results unsurprising, this view was not shared by others, and I agree with the remainder of the reviewers that this work provides a solid contribution in communication-efficient learning.